# Single molecule spectrum dynamics imaging with 3D target-locking tracking

Hao Sha [1,2,6], Yu Wu [2,3,6], Yongbing Zhang [1,4,6] ✉, Ran Liu [2,3], Xiaochen Feng [1], Haoyang Li [2,5], Zhong Wang [2], Xiufeng Zhang [2,5] & Shangguo Hou [2] ✉

Fluorescence spectra offer rich physicochemical insights into molecular environments and interactions. However, imaging the dynamic fluorescence spectrum of rapidly moving biomolecules, along with their positional dynamics, remains a significant challenge. Here, we report a three-dimensional target-locking-based single-molecule fluorescence Spectrum Dynamics Imaging Microscopy (3D-SpecDIM), a method capable of simultaneously capturing both rapid 3D positional dynamics and physicochemical parameter changing dynamics of the biomolecules with enhanced spectral accuracy, high spectral acquisition speed, single-molecule sensitivity, and high 3D spatiotemporal localization precision. As a demonstration, 3D-SpecDIM is applied to real-time spectral imaging of the mitophagy process, highlighting its enhanced ratiometric fluorescence imaging capability. Additionally, 3D-SpecDIM is used for multi-resolution imaging, providing valuable contextual information on the mitophagy process. Furthermore, we demonstrated the quantitative imaging capability of 3D-SpecDIM by imaging the cellular blebbing process. By continuously monitoring the physicochemical parameter dynamics of biomolecular environments through spectral information, coupled with 3D positional dynamics imaging, 3D-SpecDIM offers a versatile platform for concurrently acquiring multiparameter dynamics, providing comprehensive insights unattainable through conventional imaging techniques. This work represents a substantial advancement in single-molecule spectral dynamics imaging techniques.

Single-molecule fluorescence spectroscopy (SMFS) has become a prevalent tool for deciphering biomolecular interactions, conformation dynamics, and molecular compositions[1–5]. The spatially or temporally separated molecular detection of SMFS enables measuring the non-equilibrium molecular dynamics that would otherwise be buried in ensemble-averaged measurements. To date, the advancement of SMFS has paved the way for significant discoveries in various fields, including biochemistry, biophysics, and molecular biology. For instance, SMFS has elucidated the intricate dynamics of enzyme catalysis, protein folding, molecular tautomerization, and nucleic acid interactions with unprecedented precision[6–16].

In addition to leveraging fluorescence intensity information to investigate the positional dynamics of biomolecules, the fluorescence spectrum provides rich physicochemical insights into the molecular environment and interactions[2,17–26]. The spectral characteristics, including peak shifts, bandwidth, and emission profiles, reveal crucial

---

[1]School of Computer Science and Technology, Harbin Institute of Technology (Shenzhen), Shenzhen, China. [2]Institute of Systems and Physical Biology, Shenzhen Bay Laboratory, Shenzhen, China. [3]School of Life Science and Technology, Harbin Institute of Technology, Harbin, China. [4]Pengcheng Laboratory, Shenzhen, China. [5]Center for Ultrafast Science and Technology, School of Chemistry and Chemical Engineering, Shanghai Jiao Tong University, Shanghai, China. [6]These authors contributed equally: Hao Sha, Yu Wu, Yongbing Zhang. ✉e-mail: ybzhang08@hit.edu.cn; shangguo.hou@szbl.ac.cn

details about the local microenvironment, binding events, and conformational states of the biomolecules under investigation. Interrogating the fluorescence spectrum dynamics can provide critical physicochemical parameter changing information of the studied biomolecules.

In recent years, various spectrally resolved single-molecule fluorescence imaging techniques have been developed to capture the spatial distribution of biomolecular fluorescence spectra within cells[21,22,27–29]. To thoroughly investigate biological processes, it is essential to continuously monitor the dynamics of the fluorescence spectrum, along with the biomolecules' location changing dynamics. However, despite advancements in spectral dynamics detection[28,30,31], imaging the fluorescence spectrum dynamics of rapidly moving biomolecules with high speed and long observation time remains a significant challenge. This challenge arises primarily from multiple factors: (i) the rapidly moving target biomolecule may traverse outside of the excitation focal plane during the imaging period, which leads to insufficient spectrum dynamics data acquisition; (ii) the swift motion of target biomolecule during camera's exposure time can cause blurring in the fluorescence spectral image; and (iii) the limited number of fluorescence photons collected during short exposure times restricts the spatiotemporal localization precision of the biomolecules.

The primary challenge lies in maintaining the molecule within the excitation focal volume while ensuring optical conjugation between its location and the spectral imaging plane during motion. Target-locking three-dimensional single-molecule tracking (TL-3D-SMT) offers an effective solution by continuously repositioning the target molecule at the center of the excitation volume through active feedback control[32–46]. In this work, we developed a single-molecule fluorescence spectral dynamics imaging method by integrating TL-3D-SMT with rapid fluorescence spectral detection, termed 3D-SpecDIM (3D target-locking-based single-molecule fluorescence Spectral Dynamics Imaging Microscopy). In 3D-SpecDIM, motion-induced blur during spectral acquisition is significantly reduced through target-locking tracking, which keeps the target molecule relatively stationary within

the excitation volume (Supplementary Fig. 1). By combining TL-3D-SMT with high-frame-rate spectral imaging, the system achieves high spectral precision and extended observation durations, even for rapidly moving targets. Moreover, this strategy effectively prevents the loss of fast-moving target molecules during imaging, ensuring continuous data acquisition.

3D-SpecDIM enables the simultaneous capture of rapid 3D positional dynamics and fluorescence spectral dynamics of biomolecules with enhanced spectral accuracy, high spectral acquisition speed, single-molecule sensitivity, and high 3D spatiotemporal localization precision. We demonstrate its capabilities by observing freely diffusing single fluorescent microspheres and single fluorescent dyes in solutions. The spectral precision, temporal resolution, and sensitivity of spectral imaging, along with the spatiotemporal precision of 3D positional localization of 3D-SpecDIM have been thoroughly characterized. Furthermore, 3D-SpecDIM has been applied to real-time spectral imaging of the mitophagy process, demonstrating its enhanced ratiometric fluorescence imaging capability. Additionally, 3D-SpecDIM enables multi-resolution imaging, offering valuable contextual information on the mitophagy process. Finally, 3D-SpecDIM has been employed to monitor the dynamics of cell membrane polarity changes during membrane blebbing, illustrating its potential for multiparameter quantitative imaging.

## Results
### 3D-SpecDIM overview
The general outline of 3D-SpecDIM is shown in Fig. 1. Here, we utilized the 3D single molecule active real-time tracking (3D-SMART) method to perform target-locking 3D single molecule tracking, which offers single-molecule sensitivity, high tracking speed, large imaging depth, and remarkable long tracking duration time[43]. In our TL-3D-SMT configuration, a pair of electro-optic deflectors and a tunable acoustic gradient index (TAG) lens were utilized to drive the laser focus scanning on 3D (Supplementary Figs. 2 and 3). Fluorescence or scattering photons were collected by high-speed single-photon avalanche diodes

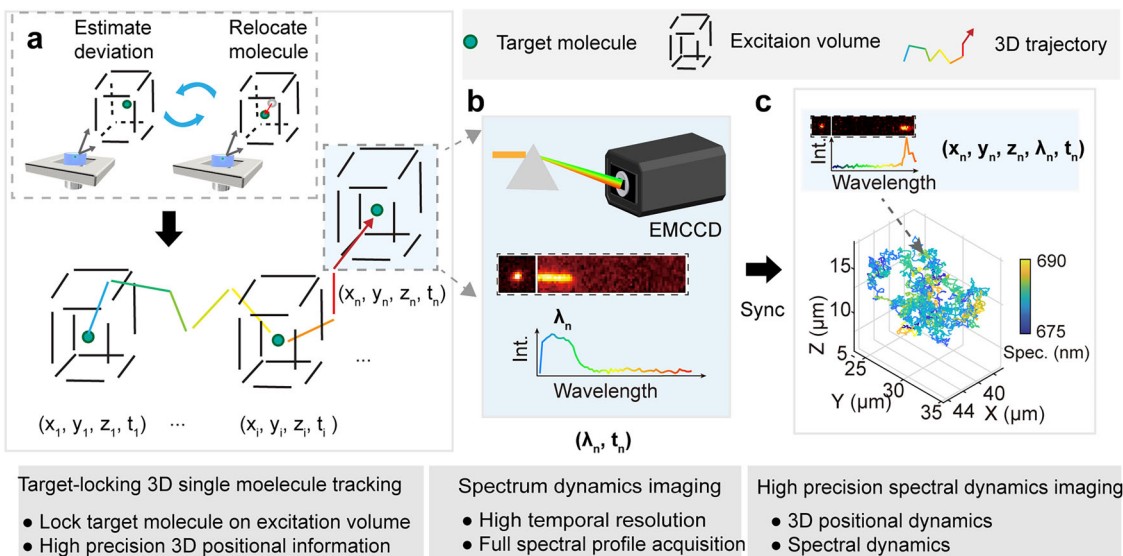

**Fig. 1 | Schematic of 3D-SpecDIM. a** The focused laser spot is rapidly scanned within a small volume after the objective lens. The molecule's deviation from the center of the illumination volume is estimated in real time using an FPGA, which processes photon arrival time information. Based on this deviation, a feedback control voltage is applied to the piezo stage to reposition the molecule at the center of the illumination volume. This process continuously locks the target molecule within the small excitation volume, enabling high-spatiotemporal-precision 3D single-molecule dynamics observation. **b** To simultaneously acquire the single-molecule fluorescence spectrum, a prism-based spectral imaging system was integrated into the detection path. The fluorescence was dispersed by a prism and projected onto an EMCCD to capture the spectral profile. The target-locking imaging strategy enables spectral acquisition with high imaging speed, high spectral precision, and large imaging depth. **c** The synchronization of 3D positional dynamics and spectral dynamics allows for multiparameter dynamic data acquisition.

and the photon arrival times were processed in real-time by a field-programmable gate array (FPGA) to calculate the deviation of the target molecule from the center of excitation volume. Subsequently, the feedback control voltages were applied to the piezo stages to relocate the molecule in the center of the excitation volume while simultaneously recording its 3D trajectory (Supplementary Fig. 4). This TL-3D-SMT can capture single fluorophore dynamics with high speed and high spatiotemporal precision in 3D, effectively preventing the loss of rapidly moving target molecules during data acquisition[41–43]. Additionally, the point-excitation-point-detection configuration of TL-3D-SMT facilitates observations at large imaging depths.

To simultaneously acquire single-molecule fluorescence spectra, a prism-based spectral imaging system was integrated into the detection path[29,47]. The fluorescence was divided into a reference channel and a spectral channel using a beam splitter. The fluorescence in the spectral channel was dispersed by a prism and projected onto one side of multiplying charge-coupled device (EMCCD), while the fluorescence in the reference channel was directly projected onto the other side. The pixel shifts between the images in the reference and spectral channels on the EMCCD were utilized to register the spectral distribution. It is noteworthy that 3D-SpecDIM requires capturing of only a single spectral stripe image at a time, allowing for the use of a smaller EMCCD detection area and achieving a higher frame rate. This capability effectively enhances spectral acquisition speed, with a theoretical maximum frame rate of 644 fps (Supplementary Table 1). After extracting spectral information from the spectral image, the data is synchronized with the 3D position of molecules with high temporal precision.

## Characterization of the performance of 3D-SpecDIM

The performance of 3D-SpecDIM was first characterized by spectrally tracking freely diffusing 200 nm fluorescent microspheres in aqueous solution. The 3D positional dynamics of the microspheres, along with their fluorescence spectral dynamics were simultaneously captured and synchronized (Fig. 2a, b). To further improve the spectral localization precision, we developed an optimized spectral detection method based on the Vision Transformer (ViT) model with a domain adaptation strategy (ViT_d)[48,49]. Unlike conventional fitting methods, which rely on curve fitting for spectral peak emission wavelength localization, the ViT_d model leverages positional encoding to capture spatial relationships within spectral images, thereby facilitating precise spectral peak emission wavelength identification. Compared to the conventional normal Gaussian fitting method, the ViT_d-based approach improved spectral localization precision from 1.63 nm to 1.11 nm, representing a 32% enhancement (Fig. 2c, Supplementary Figs. 5 and 6). Additionally, the performance of the ViT_d model was compared with that of a convolutional neural network (CNN) in terms of spectral localization precision (Fig. 2d). The ViT_d model shows improved performance over both the CNN and the normal fitting method. It should be noted that the ViT_d-based spectral detection method is more effective in high photon counts situations than in low photon counts conditions due to the influence of the signal-to-noise ratio. This improvement highlights the advantage of the positional encoding capability of ViT_d in spectral feature recognition.

Subsequently, we characterized the spectral localization precision, spectral imaging temporal resolution, and the sensitivity under various photon counts and exposure time situations with 200 nm fixed microspheres (Fig. 2d–f). The precision of fluorescence spectral localization was assessed through the standard deviation of the localized spectral peak emission wavelength position, providing a measure of the system's capability to distinguish spectral shifts in a continuous spectral measurement. The spectral localization precision was evaluated across different photon count conditions using the normal fitting method, the CNN model, and the ViT_d model. Among these approaches, the ViT_d model demonstrated superior performance,

achieving a precision of 0.3 nm with 100k photon counts for this sample (Fig. 2d).

The spectral imaging temporal resolution was evaluated by analyzing the spectral localization precision at varying camera exposure times while tracking a 200 nm fluorescent bead (Fig. 2e). The maximum achievable temporal resolution is fundamentally limited by hardware performance factors, such as shutter transfer time and readout time (Supplementary Fig. 7). However, the practical temporal resolution depends on the photon emission rate of the sample. For instance, when tracking a 200 nm fluorescent bead, an acceptable spectral localization precision can be achieved with a camera exposure time of 1 ms. Given the system's minimum acquisition time of 1.55 ms (Supplementary Table 1), the corresponding spectral imaging temporal resolution is 2.55 ms in this scenario.

To assess the sensitivity of spectral tracking, varying exposure times and excitation intensities were applied (Fig. 2f). The fluorescence photon count rate measured by the APD was used to quantify the fluorescence emission rate. As shown in Fig. 2f and Supplementary Fig. 8, increasing the exposure time enhances spectral localization precision by facilitating the collection of more photons per frame in the EMCCD spectral images. The minimum APD signal required for successfully performing simultaneous 3D single molecule tracking and spectral imaging is 5 kHz (Fig. 2f), comparable to the typical fluorescence signal of a single fluorophore. Meanwhile, the spatiotemporal resolution of 3D positional localization can reach 1 ms and several nanometers (Supplementary Fig. 9). As shown in Fig. 2d–f, an inherent trade-off exists between spectral localization precision and spectral imaging temporal resolution, both of which are influenced by the fluorescence photon count rate. These relationship plots serve as a valuable reference for optimizing imaging parameters, such as excitation intensity and EMCCD exposure time, based on the fluorescence emission rate of the sample and the specific requirements of the experiment.

## Single-molecule spectral dynamics tracking

Next, the single-molecule spectral dynamics imaging capability of 3D-SpecDIM was first demonstrated by tracking a freely diffusing SeTau 647 fluorophore in 90% glycerol solution (Fig. 2g–i). Despite the low emission rate of the single fluorophore, we successfully performed continuous and simultaneous monitoring of both its fluorescence spectrum and 3D positional dynamics for 6 s (Fig. 2g, h). To enhance the signal-to-background ratio, the EMCCD exposure time was set to 100 ms. The hydrated diameter of the fluorophore, calculated from the mean square displacement, was determined to be 0.9 nm, consistent with the expected size of a single molecule (Fig. 2i). To further demonstrate the single-molecule spectral tracking capability of 3D-SpecDIM, we conducted experiments tracking three distinct fluorescent dyes in a 90% glycerol solution, including Atto 665, SeTau 647, and Atto 565, with approximately 30 trajectories recorded for each dye (Fig. 2j–m, Supplementary Fig. 10). The hydrodynamic diameter of the tracked molecules, calculated using the Einstein-Stokes equation, was 1.28 ± 0.55 nm (Fig. 2), with a tracking duration of 5.47 ± 4.51 s (Fig. 2l). Notably, the measured hydrated diameter aligns with previously reported single-molecule measurements[43], further confirming the single-molecule tracking capability of 3D-SpecDIM. The spectral peak emission wavelength (Fig. 2k) and fluorescence intensities (Fig. 2m) were also statistically analyzed. To assess spectral accuracy, we compared the spectral peak emission wavelength measured by 3D-SpecDIM with those obtained from a commercial spectro-fluorometer (Supplementary Fig. 20 and Supplementary Table 7). It is worth noting that 3D-SpecDIM is primarily designed to capture spectral changing dynamics that arise from environmental or biological fluctuations. In this context, the precision of measuring relative spectral shift—rather than the absolute peak emission wavelength position—provides the most meaningful information.

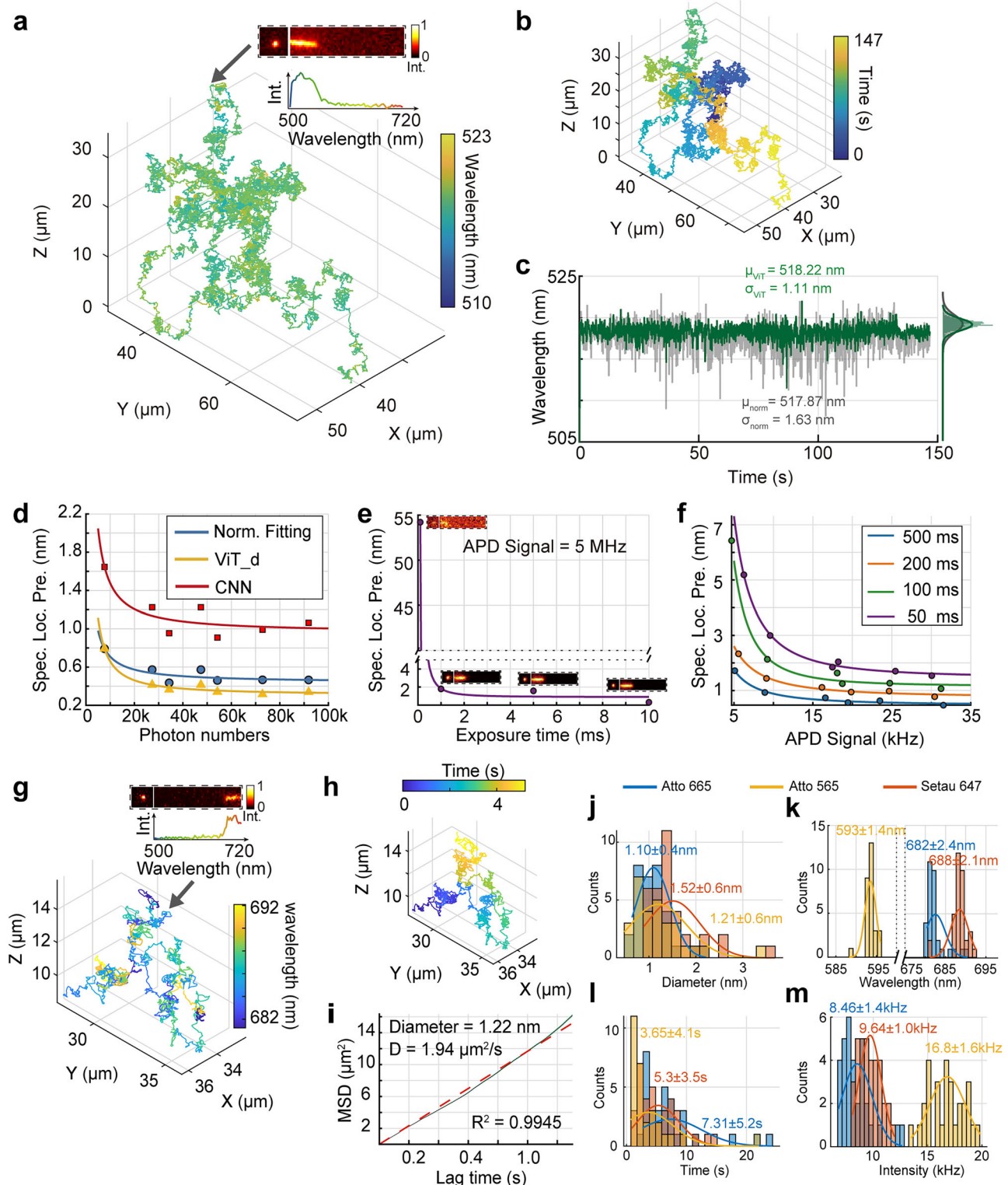

Furthermore, the single-molecule tracking capability was further validated through the observation of photoblinking. In this experiment, BSA protein labeled with two Atto 565 fluorescent dyes was tracked using 3D-SpecDIM, during which photoblinking of the dyes was observed (Supplementary Fig. 11). The fluorescence intensity alternated between single-dye and two-dye intensity levels. To minimize photobleaching, the excitation laser power was slightly reduced in the experiment. These results collectively validate the single-molecule spectral tracking capability of 3D-SpecDIM.

Notably, 3D-SpecDIM facilitates the acquisition of fluorescence spectra of freely diffusing molecules at picomolar concentration, representing a substantially sensitivity enhancement over conventional fluorescence spectrometers. Additionally, this single-molecule spectral imaging approach also can greatly reduce the sample size required for analysis based on spectral change information.

### Identify particle switching with 3D-SpecDIM

In the realm of real-time single-particle tracking, a critical question often arises regarding how to differentiate between two particles with

**Fig. 2 | Characterization of the performance of 3D-SpecDIM. a, b** 3D moving trajectory of a 200 nm fluorescent bead diffusing in water solution. The color indicates spectrum (**a**) or time (**b**). Insert in (**a**) shows the fluorescence spectral image in EMCCD and the spectral profile for one time point. All heat maps use the same color coding, representing normalized intensity. **c** Comparison precision of normal (Gaussian distribution) fitting spectral localization method (gray line) and ViT_d spectral localization method (green line). Bin time: 29 ms. **d** Spectral localization error as a function of photon numbers collected on the EMCCD in a single frame exposure time. The blue line indicates the precision obtained with the normal fitting method, while the orange line shows the precision obtained with ViT method, and the red line represents the CNN method. Exposure time of EMCCD: 100 ms. **e** Spectral localization error as a function of exposure time. The APD signal of fluorescent bead (the photon numbers captured by the APD per second) is 5 MHz. **f** Spectral localization precision as a function of APD signal with different

exposure time. **g, h** 3D moving trajectory of a single SeTau 647 fluorescent molecule diffusing in 90 wt% glycerol. The trajectory is color-coded to represent either the spectrum (**g**) or time (**h**). The inset (**g**) displays the fluorescence spectral image captured by the EMCCD and the corresponding spectral profile at the current time point. Exposure time: 100 ms. **i** Mean square displacement (MSD) as a function of lag time for the trajectory shown in (**g**). The red dashed line represents a linear fit to the MSD, yielding a diffusion coefficient of 2.67 $\mu m^2/s$. The calculated hydrodynamic diameter of the molecule is 0.89 nm, which aligns with the expected size of a single fluorophore. Statistics of the hydrodynamic diameter (**j**), spectral peak emission wavelength (**k**), tracking durations (**l**), and emission rates (**m**) of three different fluorophores: Atto 665 ($N = 33$ trajectories), SeTau 647 ($N = 35$ trajectories), and Atto 565 ($N = 26$ trajectories). The mean and standard deviation values are labeled on the histogram figure. Exposure time of EMCCD: 100 ms.

similar fluorescence emission rates within a continuous tracking trajectory. We propose that this challenge can be addressed by analyzing differences in their fluorescence spectra. To validate this concept, we tracked a mixture of microspheres with two distinct fluorescence spectra in solution using 3D-SpecDIM (Supplementary Fig. 12 and Supplementary Movie 1). Supplementary Fig. 12a shows a trajectory where the tracked object transitions from a yellow fluorescent bead to a green fluorescent bead. Both fluorescent beads exhibit similar fluorescence intensities, although the green bead displays greater intensity fluctuations. The switching event could not be distinguished from the trajectory alone, as there were no observable hopping movements (Supplementary Fig. 12d) and both particles exhibited similar diffusion speed (Supplementary Fig. 12c). However, the particle switching is easily identifiable through their spectral information. These results demonstrate that spectrum dynamics imaging can enhance the accuracy of real-time single particle tracking in complex environments.

## Improved ratiometric fluorescence imaging

Ratiometric fluorescence imaging is widely used to monitor local environment changes[50]. In traditional ratiometric fluorescence imaging, such as wavelength-split detection, two fluorescence channels are configured to collect signals across different wavelength ranges. The ratio between the two channels is utilized as a means of spectral characterization. However, the accuracy of ratiometric imaging can be compromised by emission spectrum crosstalk, which reduces sensitivity and introduces potential artifacts[51]. The capability of 3D-SpecDIM to acquire detailed spectral profiles enables the implementation of spectral unmixing[52], which substantially improves the sensitivity of ratiometric fluorescence imaging, as demonstrated by simulation results (Supplementary Fig. 13).

To demonstrate the improved ratiometric fluorescence imaging capability of 3D-SpecDIM, we applied it to monitor the mitophagy process (Fig. 3 and Supplementary Fig. 14). Mitophagy is an essential cellular mechanism that selectively degrades damaged or redundant mitochondria, thereby preserving cellular homeostasis and maintaining energy equilibrium[53]. Owing to the dynamic movement of autophagosomes and lysosomes during mitophagy, continuously observing this mitophagy process within a single focal plane presents significant challenges. The 3D target-locking tracking capability of 3D-specDIM overcomes this limitation by keeping the autophagosomes within the illumination volume, enabling continuous observation and providing unambiguous information on interaction dynamics.

During mitophagy, autophagosomes undergo translocation from the cytoplasm to the lysosome, accompanied by a decrease in pH (Supplementary Fig. 15). To monitor this translocation, we engineered a pH-sensitive fluorescent mitochondrial probe (Fig. 3a, Supplementary Figs. 16 and 17). This probe integrates mGold fluorescent protein, whose fluorescence intensity decreases as pH levels decline, alongside HaloTag-JF549, whose fluorescence remains relatively stable across

varying pH conditions (Supplementary Fig. 17). Consequently, the intensity ratio of mGold to JF549, referred to as the pH ratio, serves as an indicator of the local pH environment and is used to determine the position of autophagosomes. Additionally, lysosomes are labeled with deep red LysoTracker, enabling simultaneous monitoring of lysosome.

We employed rapamycin (Rapa) to induce autophagy pathways by inhibiting the mammalian target of rapamycin (mTOR). The autophagosome was subsequently tracked in 3D, while the spectra of mGold, JF549, and deep red LysoTracker were simultaneously monitored (Fig. 3b, c). The intensity ratio, derived from spectral unmixing using 3D-SpecDIM, demonstrated a remarkably improvements in sensitivity and accuracy compared to wavelength-split detection methods, which rely directly on intensity measurements from individual fluorescence channels (Supplementary Figs. 13 and 18). 3D-SpecDIM yielded a calibrated lysosomal pH of 4.5, consistent with previously reported values, whereas the wavelength-split detection method produced a pH of 6, showing a substantial deviation. In rapamycin-treated cells, there was a notable increase in the number of trajectories exhibiting a decreased pH ratio compared to the control group (17 vs. 1 per hour). These trajectories also displayed higher diffusion coefficients (Fig. 3d–f and Supplementary Fig. 19).

Additionally, 3D-SpecDIM offers the capability to acquire the 3D volumetric image of lysosome while simultaneously performing spectral tracking of mitochondria, a method referred to as "multi-resolution imaging" (Supplementary Notes 4). By registering the fluorescence photons with the corresponding laser focus positions in a separate detection channel, the 3D volumetric images of lysosome can be reconstructed. Multi-resolution imaging is achieved by aligning the 3D spectral trajectory with the 3D volume image, thereby providing valuable contextual information on the mitophagy process (Fig. 3g, h and Supplementary Movie 2).

## Multiparameter quantitative imaging

A distinctive feature of 3D-SpecDIM is its ability to continuously monitor the fluorescence spectral profile changing dynamics, while simultaneously acquiring 3D positional dynamics. This capability facilitates the quantitative analysis of the kinetics and dynamics of biological interactions. In this study, we employed 3D-SpecDIM for quantitative imaging of the cellular blebbing process. Cellular blebbing, characterized by transient protrusions of the cell membrane, plays a crucial role in cell apoptosis and migration[54]. We observed that cellular blebbing can be induced by Nile Red staining combined with high-intensity blue light illumination (Supplementary Table 2). Imaging the rapidly changing dynamics of cellular blebbing is challenging with conventional grating-based spectral scanning microscopes due to their limited temporal resolution and spectral localization precision (Supplementary Table 6). In contrast, 3D-SpecDIM offers high temporal resolution for spectral imaging, enhanced spectral localization precision, and simultaneous real-time 3D positional tracking, making it a powerful tool for studying fast dynamic processes.

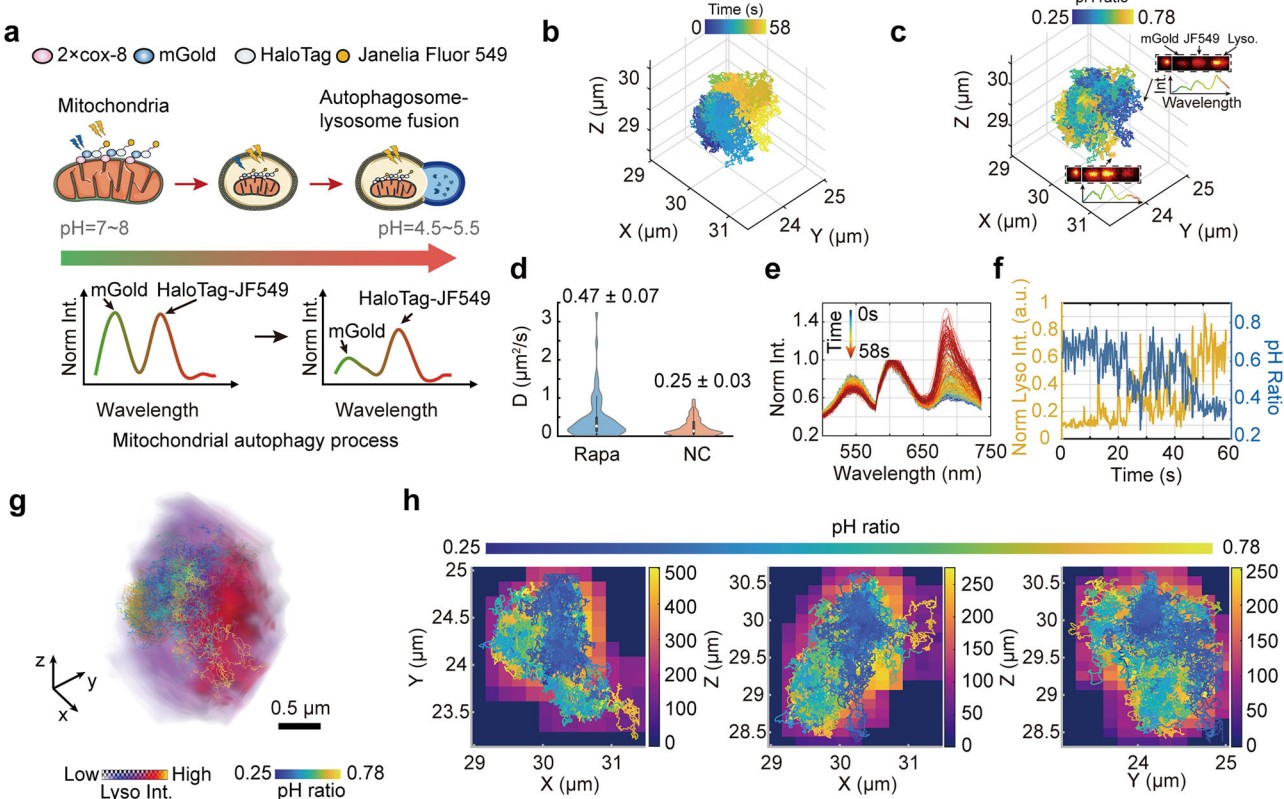

**Fig. 3 | Enhanced ratiometric fluorescence imaging with 3D-SpecDIM in live cells. a** Schematic of mitophagy. The mitochondrion was labeled with a pH-sensitive fluorescent probe mGold-HaloTag-JF549. The intensity ratio of mGold to JF549 serves as an indicator of the local environmental pH. As the pH of the local environment decreases, the intensity ratio correspondingly decreases. **b, c** 3D trajectory of a mitochondrion during mitophagy. The color indicates time (**b**) or intensity ratio (**c**). Inserts in (**c**) show the spectral images in two different time points indicated by arrows, which have different pH ratio values. **d** Comparison of mitochondrial moving speed in rapamycin-treated cell (0.47 ± 0.07 μm²/s, Mean ± SD) and untreated cell (0.25 ± 0.03 μm²/s, Mean ± SD). For each condition, $n = 66$ trajectory segments, extracted from mitochondria in multiple independent cells from the same culture. Center line in the violin plot indicates the median; box bounds represent the 25th and 75th percentiles; whiskers extend to the most extreme data points within 1.5 × the interquartile range from the box. **e** The fluorescence spectral profile changes during mitophagy process. The three spectral peaks from left to right correspond to mGold, JF549, and LysoTracker, respectively. **f** The pH ratio of pH-sensitive fluorescent probe (blue line) or the intensity of lysosome (yellow line) as a function of time. Note that after 40 s, pH decreased when lysosome strength increased. **g** Multi-resolution imaging of mitophagy process. The 3D volume image of lysosome is overlaid with spectrally encoded 3D moving trajectory of mitochondrion. The blue-to-yellow color-coded trajectory represents the 3D movement of the mitochondrion, with the color encoding the pH ratio. Meanwhile, the 3D image of the lysosome is color-coded from violet to red, indicating fluorescence intensity. The 3D lysosome volume image was reconstructed by registering the fluorescence photons to laser focus positions in a separated detection channel. Detailed reconstruction information is provided in Supplementary Notes 4. The checkerboard pattern represents transparency, while the alternating color blocks indicate different intensity levels. **h** The sum intensity projection images of (**g**) in xy plane (left), xz plane (middle), and yz plane (right).

To monitor this process, we employed 3D-SpecDIM to track the 3D positional dynamics of 100 nm silver nanoparticles (AgNPs) attached to the cell surface[55], while simultaneously monitoring the fluorescence spectral dynamics of the surrounding Nile Red molecules (Supplementary Fig. 20). Nile Red is a solvatochromic, lipophilic fluorescent dye whose fluorescence emission spectra vary in response to the polarity of its environment, making it an effective indicator of cell membrane polarity[25] (Fig. 4a–c, Supplementary Figs. 21–23). The plasmonic enhancement of the electromagnetic field near the surface of AgNPs leads to an increase in local light density[56], which is a critical factor in initiating of cellular blebbing (Supplementary Table 2). As a result, cellular blebbing frequently initiates at sites where these nanoparticles are located (Supplementary Movie 3). It is noteworthy that, although AgNPs are effective photocatalysts, no phototoxicity-induced blebbing attributable to AgNPs alone was observed (Supplementary Table 2).

The results show that the half-accumulative probability time for bleb appearance is 5.26 min (Fig. 4d and Supplementary Fig. 20). Furthermore, blebbing occurred more frequently at the cell periphery than on the inner side, with a frequency 1.75 times higher (Supplementary Fig. 23d). Intriguingly, during blebbing events, AgNPs exhibited pronounced vertical movement, with a speed of 0.45 ± 0.08 μm per minute and a displacement of 3.2 ± 0.95 μm, while the translational movement was slower, with a speed of 0.27 ± 0.02 μm per minute and a displacement of −0.02 ± 0.02 μm (Fig. 4e and Supplementary Fig. 24). This observation suggests that the AgNP was elevated by the expanding bleb, a dynamic process that would be difficult to capture in 2D spectral imaging using conventional spectral scanning microscopy.

Additionally, during the cellular blebbing process, the membrane bleb size increased concomitantly with a decrease in the spectral peak position, indicating a decrease in membrane polarity, with a decreasing rate of 0.65 ± 0.32 nm per minute (Fig. 4c). In contrast, non-blebbing cells exhibited considerably smaller changes in both the movement distance of AgNPs and spectral dynamics (Supplementary Table 3).

## Discussion

In summary, we developed 3D-SpecDIM, a real-time single-molecule spectral dynamics imaging method utilizing 3D target-locking

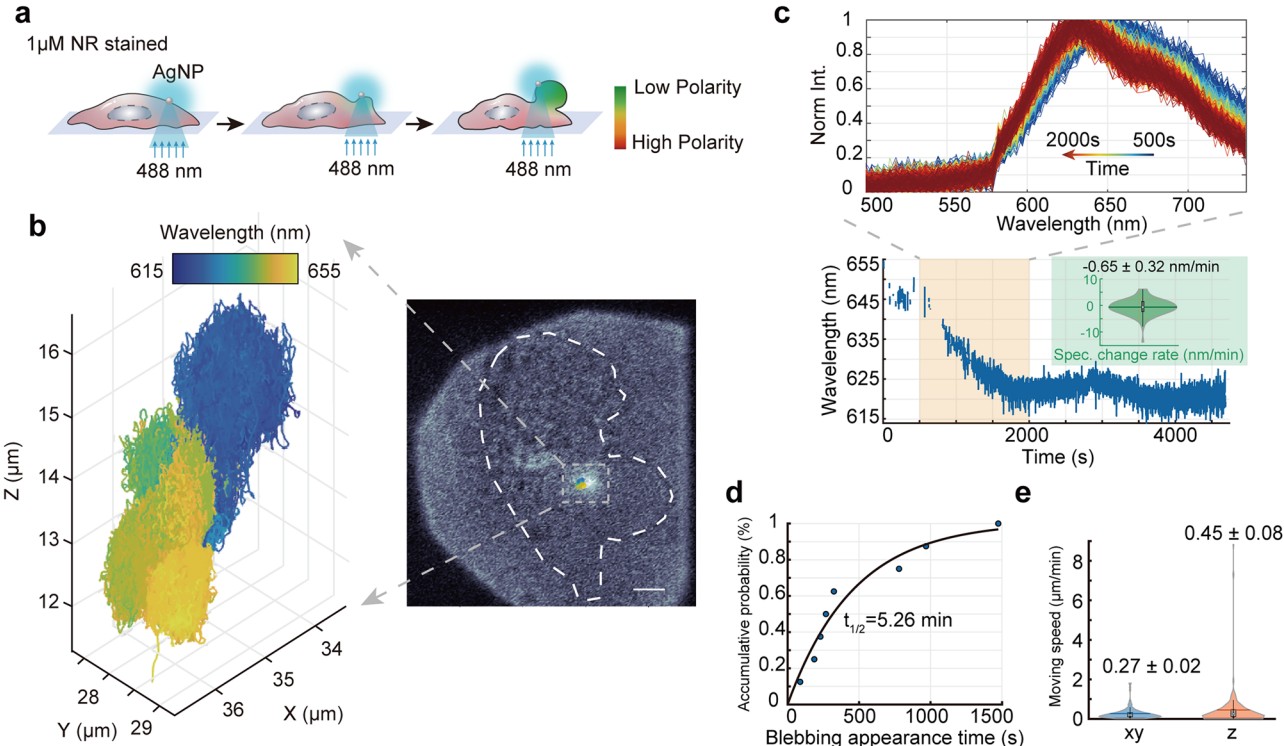

**Fig. 4 | Quantitative spectral imaging with 3D-SpecDIM in live cells. a** Schematic of cellular blebbing. The cell membrane was stained with Nile Red. With blue laser illumination, the cellular blebbing can be induced. AgNP was tracked in 3D and the fluorescence spectral dynamics of Nile Red was monitored by 3D-SpecDIM. **b** 3D trajectory of a single AgNP during cellular blebbing event. The color in trajectory indicates the fluorescence peak emission wavelength of Nile Red. Insert: The overlayed bright field image with 3D trajectory of AgNP. The white dash line shows the profile of cell. Scale bar: 5 μm. Data are representative of multiple independent experiments, with the number of independent tracks for each condition summarized in Supplementary Table 3. **c** Upper panel: the changing fluorescence spectral profiles of Nile Red from 500 to 2000 s during the blebbing process; Lower panel: fluorescence spectral peak emission wavelength of Nile Red as a function of time during the cellular blebbing. Only successfully fitted spectral data points are presented. At some time points, the signal-to-noise ratio was too low to obtain a valid spectral peak emission wavelength estimation. Insert: The spectral change rate of Nile Red during cell blebbing was calculated in 60-s intervals (Mean ± SEM). $n = 148$ segments from 8 trajectories in multiple independent cells. **d** The accumulative probability of cell blebbing appearance as a function of time. The points data were curve fitted with the function $y = a(1 - e^{-b \cdot x})$. When $y = 0.5$, the corresponding $x$ value is 315.93 s. **e** The comparison between vertical and translational moving speed of AgNPs during cell blebbing (Mean ± SEM). $n = 148$ segments (60-s intervals) from 8 trajectories in multiple independent cells. For the violin plots in (**c**, **e**), the center line indicates the median; box bounds represent the 25th and 75th percentiles; whiskers extend to the most extreme data points within 1.5× the interquartile range from the box.

tracking. This method enables the simultaneous capture of the 3D positional dynamics and fluorescence spectral dynamics of biomolecules with enhanced spectral accuracy, high spectral acquisition speed, single-molecule sensitivity, and high 3D spatiotemporal localization precision. The ability to detect multiparameter dynamics allows for the interrogation of subtle physicochemical changes within the microenvironment during rapid biological events. Furthermore, its capacity for spectral profile acquisition facilitates the use of spectral unmixing, achieving higher sensitivity and accuracy in ratiometric fluorescence dynamics imaging. Additionally, the continuous detection of spectral changes, coupled with 3D positional dynamics imaging, provides the potential for quantitative analysis of both the kinetics and dynamics of biological systems. These advantages make 3D-SpecDIM a powerful tool for comprehensive analysis in both biochemical and biological systems.

While 3D-SpecDIM was demonstrated with fluorescence spectral dynamics, it is also applicable to scattering spectral dynamics imaging, making it suitable for a broad range of single nanoparticle spectral analyses in an untethered state, such as monitoring catalytic and chemical reaction processes[57]. Additionally, 3D-SpecDIM can be extended to Raman imaging, allowing the simultaneous acquisition of 3D particle positional dynamics and Raman spectral dynamics[58]. Together, 3D-SpecDIM provides a versatile platform for the concurrent acquisition of multiparameter dynamics, delivering comprehensive insights unattainable through conventional imaging techniques.

## Methods

### The optical setup of 3D-SpecDIM

The schematic diagram of 3D-SpecDIM is shown in Supplementary Fig. 2. Three lasers with wavelengths of 488, 561, and 638 nm (LBX-488-60-CSB-PPA, LCX-561L-50-CSB-PPA, and LBX-638-100-CSB-PPA, OXXIUS) were used for excitation. Three pairs of achromatic doublets lenses and three pinholes were utilized for spatial filtering for the three lasers. Two dichroic mirrors (T510lpxru and T588lpxr, Chroma) were used for combining laTsers. Then the laser polarization was tuned by a half waveplate (AHWP05M-580, Tholabs) and cleaned by Glan-Thompson polarizer (GLP10-A, Lbtek) before entering a pair of EODs (Model 310 A, Conoptics Inc.). The laser beam was then expended by a pair of lenses (AC254-075-A-ML and AC254-300-A-ML, Thorlabs). A tunable acoustic gradient lens (TAGLENS-T1, Mitutoyo) was used to modulate the axial focus position. Then a pair of lenses (AC254-250-A-ML and AC254-150-A-ML, Thorlabs) were used to relay the laser to objective lens (HC PL APO 100x/1.40 OIL CS2, Leica). A dichroic mirror (ZT405/488/561/640 rpcv2, Chroma) was utilized for separating the excitation and emission light.

Subsequently, emission fluorescence was filtered by band-pass filter (refer to Supplementary Table 4 for detailed filter configuration)

and split by a beam splitter and directed into tracking and spectral imaging paths. In the tracking path, emission light was focused onto an APD using an achromatic lens (L11, AC254-050-A-ML, Thorlabs). In the spectral path, the fluorescence was further divided by a 30/70 splitter (BSS10R, Thorlabs). A CaF2 prism (PS863, Thorlabs) was used to disperse the fluorescent spectral signal. Finally, the beams were reflected by a right-angle prism mirror (MRA25L-E02, Thorlabs) and focused with two achromatic lenses (L12, L13; AC254-150-A-ML, Thorlabs). An EMCCD (iXon Ultra 897, Andor) was used to record the spectral distribution. A two-axis piezo nanopositioner (Nano-PDQ275, Mad City Lab) and a z axis piezo nanopositioner (Nano-OPQ65, Mad City Lab) were used to move the sample and objective, respectively. An FPGA (PCIe-7858, National Instruments Corp.) was utilized to count the fluorescent signal, calculate the molecular position, control the nanopositioners, and record the 3D trajectory of molecules.

Data acquisition was performed using Andor Solis (v2.90) and LabVIEW (NI LabVIEW 2023). The 3D visualization of the multi-resolution image was performed with Avizo Amira software (Avizo 3D v2022.2). Image processing and spectral analysis were performed in MATLAB R2024b (MathWorks) with custom scripts and ImageJ v1.53k.

If two or more targets are present within the excitation volume, the tracking system localizes to the peak emission wavelength of their combined fluorescence emissions. As a result, the tracking system tends to lock onto the target with the higher fluorescence emission rate. Practically, the density of molecules or particles should be maintained at a low level to minimize the presence of multiple targets in the excitation volume. The synchronization of 3D single molecule tracking and spectral imaging was realized by instant triggering EMCCD with the tracking signal or by aligning the start time point of tracking and spectral signal. The spectral registration method can be found in Supplementary Figs. 25 and 26. The laser power after the objective lens for each experiment can be found in Supplementary Table 3.

## Spectral peak emission wavelength calculations

The "normal fitting method" refers to fitting spectral curves with a Gaussian distribution. Specifically, this method involves using all recorded intensity-wavelength pairs and applying a curve fitting algorithm to extract the parameters of the Gaussian profile. The spectral peak emission wavelength corresponds to the position of the peak of the fitted Gaussian curve, representing the most likely wavelength of maximum emission.

To improve the robustness of peak emission wavelength estimation under low-photon conditions, we implemented additional data quality control criteria. Specifically, frames were excluded if the average signal intensity within the central 5×5 region of the position image did not exceed a predefined threshold relative to the background estimated from the peripheral corner regions. Furthermore, frames with fitting errors of the position or spectral channels greater than 0.3 were also discarded. These strategies ensure that only frames with sufficient signal quality and reliable localization are included in the final spectral peak emission wavelength analysis.

## Sample preparation and 3D-SpecDIM imaging

**Fluorescent nanoparticle sample.** We used two types of fluorescent spheres (FSDG002 and FSSY002, Bangs Lab) to calibrate the 3D-SpecDIM system. For fixed beads, we diluted the fluorescent bead stock solution into PBS at a ratio of 1:1000. For free-moving beads, the bead stock solution was diluted into pure water at a ratio of 1:4000. The fluorescent bead samples were imaged or tracked according to the configuration in Supplementary Table 3.

**Single dye molecule sample.** SeTau-647-NHS (K9-4149, SETA Biomedicals), Atto-565-NHS (72464-1MG-FO, Merck) and Atto-665-NHS

(AD 665, Atto-tec) were initially dissolved in DMSO at a concentration of 5 mg/ml, then aliquoted into 5 μL vials and stored at −80 °C. The dye was then diluted to 100 pM with distilled water. Subsequently, 1 mL of the dye solution was mixed with 9 g of glycerol (56-81-5, Aladdin) and added to glass-bottom culture dishes (801001, Nest) for 3D-SpecDIM experiments. For single-molecule spectral tracking experiments, the EMCCD exposure time was set to 100 ms, and the excitation laser power at the objective was adjusted to 2 μW to ensure the collection of sufficient photons per frame while minimizing photobleaching.

**Mitophagy tracking.** HeLa cells (CCL-2, ATCC) were cultured in DMEM (PM150223-500, Pricella) supplemented with 10% FBS (C04001-500, VivaCell) and 1% penicillin-streptomycin (PB180120, Procell), maintaining a 37 °C and 5% $CO_2$ environment until reaching 60−80% confluency. Prior to transfection, cells were plated in glass-bottom dishes (801001, Nest) at densities promoting ~70% coverage. The transfections were performed with Lipofectamine 3000 (L3000008, Invitrogen). After post-transfection for 14−18 h, cells expressing 2×COX8-mGold-HaloTag were labeled with JF549 ligand (GA1110, Promega). Then immediately add rapamycin (HY-10219, MedChemExpress) to sample and incubate for 2 h. The COX8, representing Cytochrome C Oxidase Subunit 8, and as a unique pilot targeting mitochondrial peptide (MSVLTPLLLRGLTGSARRLPVPRAK)[59], plays a crucial role in the respiratory chain by encoding cytochrome C oxidase[59]. The mGold, a yellow fluorescent protein, was clone from pCMV-mGold-Actin-C-18 (MiaoLingBio; P50209) (Supplementary Fig. 16). The COX8 and Halo tag was synthesized (by Tsingke Biotech, China) and then fused with mGold by referencing mEmerald-Mito-7 (plasmid 54160, Addgene) using seamless cloning. For three-color mGold-HaloTag-Lysosome tracking, cells were further incubated with 50 nM LysoTracker™ Deep Red (L12492, Thermo Fisher Scientific) working solution for 20 min at 37 °C to enable precise visualization. The subsequent co-localization imaging (Supplementary Figs. 27 and 28) or tracking of mGold, JF549, and LysoTracker Deep Red was performed via confocal fluorescence microscopy (LSM 980, Zeiss; Dragonfly CR-DFLY-202-40, and Dragonfly CR-DFLY-202-2540, Nikon) or 3D-SpecDIM. The presented trajectory in Fig. 3b, c was part of a long trajectory. The alignment procedure of the tracking trajectory on EMCCD image can be found in Supplementary Fig. 29. All kymograms and intensity changes for the relevant data in the main text are shown in Supplementary Figs. 30 and 31.

## Spectral unmixing

While conventional wavelength-split detection methods susceptible to the spectral crosstalk, the full spectral profile acquisition capability of 3D-SpecDIM enable us to perform spectral unmixing and improve the sensitivity and accuracy of ratiometric fluorescence imaging. The process involves solving the equation $M = S \cdot C + E$, where $M$ is the measured signal, $S$ is the known fluorescence spectrum measured by fluorescence spectrophotometer, $C$ is the coefficient matrix representing the contribution of each source, and $E$ is the error. To solve for $C$, the least squares method is applied, resulting in $C = (S^T \cdot S)^{-1} \cdot S^T \cdot M$.

The performance of 3D-SpecDIM with spectral unmixing was demonstrated through ratiometric fluorescence imaging of mGold and JF549 (Supplementary Fig. 18). The calculated intensity ratio was converted to pH values using a calibration curve. Spectral unmixing yielded a calibrated lysosomal pH of 4.5, consistent with previously reported values, whereas the conventional wavelength-split detection method produced a pH of 6, indicating a large deviation. These results highlight the improved accuracy of 3D-SpecDIM with spectral unmixing compared to traditional wavelength-split detection

## Quantitative imaging of the cellular blebbing

HeLa cells were seeded in glass bottom culture dishes (801001, Nest) and incubated for 18–20 h at 37 °C in 5% $CO_2$. For staining, cells were treated with 1 µM Nile Red in the culture medium and incubated for 25 min. Simultaneously, AgNPs (103716, 1 mg/mL, Xfnano) were prepared at a 20 µg/mL concentration in PBS, with ultrasonication for 5 min before adding them to cell sample. Following the Nile Red staining, cells were washed with cold PBS twice and cooled on ice for 5 min before adding the 20 µg/mL AgNPs solution to achieve a final in-cell concentration of 10 µg/mL. Then incubate cell on ice for 10 min. Finally, cells were washed with PBS and ready for imaging. During data collection, the brightfield light source was turned on for a short period to assess the cell's blebbing status, which temporary interrupted the collection of fluorescence spectral data due to the poor signal-to-noise ratio. The spectral information during this period was filled with the spectral values of the previous point. For all live cell imaging with 3D-SpecDIM, the sample temperature was maintained to 37 °C with a heating system (TC-1-100, Bioscience Tools).

## Reporting summary

Further information on research design is available in the Nature Portfolio Reporting Summary linked to this article.

## Data availability

The spectral imaging and 3D trajectory data generated in this study have been deposited in the Zenodo database under accession code 10.5281/zenodo.15367615. The dataset includes raw and processed spectral imaging data, 3D trajectory data, and associated analysis files sufficient to verify the results reported in this article.

## Code availability

The SpecViT and data analysis code are publicly available at: https://github.com/houlab/3D-SpecDIM and archived at Zenodo (https://doi.org/10.5281/zenodo.16742927)[60].

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

## Acknowledgements

We would like to acknowledge support from the Shenzhen Medical Research Fund (B2301003 to S.H.), the National Natural Science Foundation of China (22204106 to S.H., 62031023 and 62331011 to Y.Z.), the Shenzhen Science and Technology Project (GXWD20220818170353009 to Y.Z.), the Evident & Shenzhen Bay Laboratory Joint Optical Microscopic Imaging Technology Development Program (S234602004-4 to S.H.), and the Guangdong Provincial Pearl River Talents Program (2021QN02Z631 to S.H.). We wish to thank Prof. Yan Zhao in Sun Yat-sen University for her assistance on fluorescence spectrophotometer. Some elements of Fig. 3 were adapted from Servier Medical Art (https://smart.servier.com), licensed under a Creative Commons Attribution 4.0 International License (CC BY 4.0).

## Author contributions

S.H. initiated the project. S.H., Y.Z., and H.S. conceived of the project. H.S., Y.W., and R.L. conducted the experiments. H.S., Y.W., Y.Z., and H.S. analyzed the data. X.F., H.L., and Z.W. contributed to building the microscope and data analysis. X.Z. contributed to sample preparation. S.H., H.S., Y.W., and Y.Z. wrote the manuscript. S.H. and Y.Z. supervised the project.

## Competing interests

The authors declare no competing interests.
