## [Transparent Peer Review file · Nature Communications]

Single molecule spectrum dynamics imaging with 3D target-locking tracking

Corresponding Author: Professor Shangguo Hou

Version 0:

Reviewer comments:

Reviewer #1

(Remarks to the Author)

Sha et al. present 3D-SpecDIM, a novel single-molecule imaging technique enabling simultaneous 3D positional tracking and spectral dynamics monitoring of biomolecules, which allows for real-time, high-resolution capture of molecular behavior and physicochemical changes. By employing an active 3D target-locking mechanism, the authors effectively address key challenges in single-molecule fluorescence spectroscopy, maintaining focus on rapidly moving molecules to ensure stable, high-speed spectral acquisition with single-molecule sensitivity. This work's significance lies in its capacity to provide real-time, multiparameter insights, as demonstrated in applications such as mitophagy tracking and cellular blebbing analysis. The integration of 3D tracking with spectral dynamics makes 3D-SpecDIM a valuable tool in the field, and I recommend this paper for publication.

Comments:

1. Choice of AI Model: Why was a transformer model chosen for this application? The primary purpose here seems to be spectral denoising, as spectra from limited photon counts can be noisy, which requires effective denoising algorithms. In microscopy, similar denoising goals are often met with architectures such as U-Net or Autoencoders, which may provide more efficient solutions. Additionally, since the spectral data can effectively be represented as a 1D array (e.g., intensity vs. wavelength), would a simpler neuron network model—rather than a vision model like a transformer or CNN—be feasible?
2. Software Availability: Open availability of the software would greatly facilitate reproducibility and the broader application of the methodology, and the authors are encouraged to consider making it accessible.
3. Switching Between Targets: It is unclear if target switching is actively controlled. Based on the data, it appears that the tracking system may randomly lock onto one target when two or more overlap within the focus. Could the authors clarify how the system handles tracking when multiple targets are present?
4. Spectral Accuracy vs. Defocusing: Could spectral accuracy be affected by defocusing, specifically errors in Z-axis tracking? For example, if dye molecules move out of focus, the spectral image may become spread or blurred, potentially impacting spectral decomposition accuracy.
5. Measurement Uncertainty: The error bars on the diffusion coefficient and the estimated single-molecule diameter are missing.
6. Potential Cytotoxic Effects of Silver Nanoparticles: The authors suggest that silver nanoparticles might enhance local fields, leading to photodamage in cells. Since silver is known to act as a photocatalyst, it may also facilitate chemical reactions that are toxic to cells. Au is more inert. Further elaboration on potential catalytic effects would be beneficial.

Reviewer #2

(Remarks to the Author)

In the manuscript submitted to Nature Communications, Sha et al. present a three-dimensional spectrum dynamics imaging microscopy (3D-SpecDIM), building on their earlier work in real-time 3D tracking (Hou et al., Nature Communications, 11, 3607, 2020). This system, which utilizes a pair of electro-optic deflectors and a tunable acoustic gradient index lens, is applied to monitor pH changes during the mitophagy process in a cell and nanoparticle movement during laser-induced cellular blebbing. While the concept of simultaneous tracking of fluorescent spectra and 3D positioning of fluorophores is interesting, the system characterization and results do not support the authors' claims. The manuscript describes the technique as 'single-molecule' spectrum dynamics imaging, yet has problems to demonstrate in true single-molecule context and resolution. Furthermore, the advantages presented in the application sections by the proposed technology are not convincingly shown to be inherent to the technology itself but rather seem to derive from conventional work. Additionally, the claimed 3D tracking capability does not appear to be useful in discussion of the application demonstration. Detailed concerns and comments are outlined below.

Major comments

1. First and foremost, the manuscript only demonstrates what is described as 'single-molecule' imaging in the experiment shown in Fig. 2g-h. All other experiments, including the biological applications, do not focus on individual single fluorescent molecules but rather on ensembles of fluorophores or nanoparticles. Additionally, the only image provided in the inset of Fig. 2g, which is meant to support the single-molecule claim, does not convincingly show a single molecule. The image suggests an aggregation of molecules, as indicated by its brightness and the duration of tracking (6 seconds). Given these observations, the manuscript should not claim 'single-molecule' capabilities in its title without unequivocal evidence of single-molecule tracking and spectrum acquisition.
2. The manuscript reports that the authors' system can achieve a frame rate of up to 644 fps, corresponding to an exposure time of 1.55 ms. However, I am skeptical that the system can collect sufficient photons from single molecules within this brief exposure time to accurately determine 3D positions and spectral data. The description of the system's hardware capabilities does not convincingly support the performance claims necessary for single-molecule studies.
3. On page 5, line 125, the authors state that their spectral detection method shows 'a significant improvement'. This assertion seems to be based on achieving a 0.52 nm smaller standard deviation using their Vision Transformer method, as shown in Fig. 2c. It is debatable whether this difference is statistically significant.
4. In Fig. 2d-f, the authors use 'precision (nm)' on the y-axes. If this is equivalent to the sigma mentioned in Fig. 2c, the purpose of these plots needs further clarification. The photon counts reported exceed those from fluorescent beads used in experiments Fig. 2a-c, which raises questions about the method's compatibility with single-molecule signals. Moreover, a 100 ms exposure time is used here, which significantly contradicts their claimed operational time of 1.55 ms.
5. Regarding Fig. 2f, it is unclear how this figure supports the authors' claim of a minimum 7 kHz emission rate, particularly since the exposure time used significantly exceeds the 1.55 ms discussed in the main text. The origin of the 7 kHz rate is also not clearly elucidated.
6. I find it difficult to understand the support provided by Supplementary Table 1 for the claimed 1.55 ms temporal precision. I am skeptical that such a short exposure time can capture sufficient photons to accurately calculate spectral data. The limiting factor here is not merely the camera's acquisition speed, but the photon budget from the fluorescent molecules in the sample. The authors should provide an analysis of the photon yield from typical fluorophores used in their applications to substantiate their claims.
7. The authors should include data showing the trajectory of captured photon counts over time in all their 3D tracking experiments, particularly in the single-molecule demonstration presented in Fig. 2g-h. This information is critical for assessing the robustness and reliability of their tracking methodology.
8. Regarding Fig. 2g-h, it remains unclear how the authors distinguish between two closely located single molecules. The authors should provide multiple raw movies of the observed molecules to demonstrate the capability of their system in resolving individual molecules in close proximity without system confusion.
9. The manuscript lacks a description of how the system tracks targeting fluorophores in 3D and manages the movement of the sample stage. It is unclear how the authors differentiate between positively and negatively defocused images to accurately control system movement. While this method may be detailed in their previous publication (Hou et al., Nature Communications, 11, 3607, 2020), clarity and completeness in the current manuscript are essential. The authors should explicitly describe the tracking method and algorithms used, ensuring the manuscript stands as an independent work.
0. I am unconvinced that the 'improved ratiometric fluorescence imaging' demonstrated in the mitophagy application showcases the benefits of their 3D-SpecDIM. The improved ratiometric characterization largely results from spectral unmixing, a technique already well-established in prior publications. Additionally, the tracking of fluorescent molecule-labeled mitochondria does not add any interpretive value to the claims made by the authors in this context.
1. The authors should also provide a clearer interpretation of Fig. 3g. They also need to clarify the meaning of the checkerboard pattern in the Lyso Int. colorbar.
2. The duration for which the authors tracked AgNP and Nile red molecules in their cellular blebbing experiment should be specified. The binding and dissociation kinetics of Nile red on and off the cell membrane are known to be rapid. Based on

the descriptions provided, it seems that the method resolved ensembles of Nile red molecules rather than individual molecules. If this is the case, tracking multiple molecules does not provide additional insights that could not be achieved with standard fluorescence imaging techniques, which can already map the changes in cell membrane blebbing and capture polarity shifts using a grating combined with standard imaging. Therefore, the tracking trajectories shown in Fig. 4b appear to be superfluous.

13. In the conclusion, the authors claim that their method 'significantly improves the quantitative analysis of both the kinetics and dynamics of biological systems.' However, the authors fail to demonstrate or discuss molecular kinetics, as the data presented appear to be from molecular ensembles rather than single fluorescent molecules. This discrepancy between claims and demonstrated results undermines the validity of their conclusions.

Minor comments

1. The manuscript does not define 'spec. cent.' despite its use. A description of what constitutes the centroid of the spectrum is necessary for understanding its relevance and how it is calculated within the context of the study.

2. The caption for Fig. 3e is misplaced, appearing after that of panel f. This should be corrected to maintain the logical order and readability of the figure captions.

Reviewer #3

(Remarks to the Author)

In this report, Sha et al. have added spectral detection to a single-particle tracking method, 3D-SMART. They call their new version "3D-SpecDIM". The achievement of adding spectral imaging to 3D-SMART is a worthy one, and they have clearly made good progress in this direction. I would like to support publication of their results, but the present version of the manuscript is not convincing regarding the accuracy and utility of this exciting method, and I am concerned that it will not be accessible at all for a non-expert audience.

First, I found this manuscript rather difficult to read, not for the language (although editing will be needed), but rather for the overall organization and clarity of the logical flow. The authors need to more clearly lay out what the challenges are and show definitively that their method overcomes them. For example, it seems to me that movement of the particle within the field of view, motion blur, and the confounding effects of knowing the stage's position precisely are key technical challenges that should be more clearly explained and demonstrated. The latter two challenges – blur and stage movement effects on spectral measurements – are not addressed at all, as far as I can tell.

The central aspect of the paper, details of how spectra are measured and the data analyzed, is remarkably difficult to parse from the writing and organization of the manuscript. Just as one minor example of the challenges in reading this paper – I am still not sure what, exactly, their neural net spits out in terms of the spectra. Are they plotting a peak? A centroid? I think it is a centroid, but I honestly cannot tell because they abbreviate to "Spect. Cent." on plots and I cannot find an explicit statement in the text, captions, or SI. There are many such issues; I have listed a few in the minor comments section below.

Another major problem I see throughout is that the authors appear to conflate making a precise measurement with making an accurate measurement. They should compare the spectra they measure to published spectra for these emitters, to illustrate that their measured spectra are accurate. I also suggest that the authors make available kymograms showing the full spectral timecourse, rather than just plotting the track in a color scale representing the (again, centroid?) that was estimated by the neural net, and showing a single snapshot of one (representative?) spectrum.

Yet another major issue for me is the set of biological applications that the authors have selected. Both mitochondria and cell blebs are microscale, not nanoscale, objects. No nanoscale position information is relevant in either case, and they move incredibly slowly. Therefore, I fundamentally do not see why 3D-SMART is needed to track these objects – why not just image them in two color channels, perhaps with some sort of autofocus? I do not think that these example applications adequately demonstrate the power of having spectral information available when tracking a nanoparticle. Note that for the blebbing application, 100nm silver nanoparticles are also tracked on the membrane – but it is not clear why the movement of these objects is relevant. The changes in their position and speed, and changes in pH, seem simply consistent with being present on a membrane bleb.

MINOR COMMENTS: This is an incomplete list, but reflects the general attention to detail that will be necessary to improve the manuscript.

- The method is touted as being able to track single fluorophores, but only one track of a single SeTau 647 (in 90% glycerol) is shown. More examples should be given in the SI to show that this is not a one-off measurement. And, further analysis is needed to illustrate that the spectra are correct.

- How does the movement of the particle in the tracking spot influence the spectrum? Is motion blur accounted for? - Need details of how the prism's nonlinear dispersive characteristics were calibrated / spectrum was calibrated. A mapping function between pixels and wavelength should be shown in the SI. Is the EMCCD detection efficiency accounted for? - Validation of spectral measurements by comparison with published spectra for different emitters should be shown in the SI. - In the main paper, no references or explanation are given for the vision transformer / domain adaptation method used to fit spectra. (A couple of references are given in the SI, however.) The comparison shown in Fig. 2c does not convincingly show that the "ViT" method is more accurate, simply that it produces a less noisy result. This analytical choice, and the accuracy (rather than precision) of the method, need to be more clearly and quantitatively justified.

- I am unclear as to what is meant by "normal fitting method" for spectra. Does this refer to fitting spectra as a sum of "normal distributions", which is indeed a common method?
- Figure 1 seems unnecessarily large and detailed given that this system is basically 3D-SMART with a prism / camera added to the detection path. I suggest re-focusing on panels d) and e), which are unique to this work. In particular, the detail in panel e) is the main point of the paper, and therefore should be shown more prominently.
- Figure 2a repeats the data from Figure 1e; this should only be shown once.
- Fig. 3 – presentation of g) and h) needs to be re-thought. Very difficult to interpret.
- What data are shown in 2c? Is this a "Spectral centroid"? Is it the peak of the spectrum?
- SI Fig. 6: Why is the image of the bead oblong? Astigmatism in the system would presumably influence the measured spectra in a focus-dependent manner. Scale bars should be shown on the images. The images also appear to have position-dependent illumination; beads are dim at right and no beads are observed at far right.
- Throughout: None of the heat maps (spectra) have colorbars or color scaling indicated
- Use of acronyms seems excessive; why is a new acronym needed for the technique when this is just 3D-SMART with spectra?
- Manuscript should be thoroughly edited for typos and grammar

Version 1:

Reviewer comments:

Reviewer #1

(Remarks to the Author)

The authors have addressed all of our concerns very well. The revised version includes new materials comparing different AI models (CNN vs transformer), effectively demonstrating the strength of their approach. However, although they mention that the code is published, the provided address (<https://github.com/houlab/3D-SpecDIM>) appears to be incorrect or unavailable. Could the authors please supply the correct address for accessing the SpecViT code? Overall, I recommend publication pending this minor correction.

(Remarks on code availability)

The code is not available for download.

Reviewer #2

(Remarks to the Author)

In the revised manuscript, Sha et al. have carefully and effectively addressed nearly all the comments and concerns raised in the previous review round. The authors have included additional experimental results to substantiate their system's capability for single-molecule analysis, which was one of the primary concerns noted in earlier feedback. They have also clarified and expanded upon sections that were previously unclear or logically difficult to follow. For example, they detailed how their 3D-SpecDIM operates to achieve tracking of particles in 3D and obtain ratiometric fluorescence imaging. I now can support the publication of this work in Nature Communications. However, there are still a few comments that the authors need to address before the manuscript can be fully accepted. These points are detailed below.

Comments

1. In their response to the previous review round, the authors noted that "In single-molecule spectral tracking experiments, the EMCCD exposure time is set to 100 ms to ensure the collection of sufficient photons in each spectral image frame." While this detail can be inferred from Figure 2 and its caption, it should be explicitly stated in the main manuscript to prevent any potential misunderstandings about their single-molecule operation by future readers.
2. The authors are advised to specify the methodology used for the dual-channel ratiometric detection discussed in the new Supplementary Figure 18. Although they provided several paragraphs explaining how their new method is better than the conventional one in response to the 10th comment of Reviewer #2, there is no significant addition to the main text apart from Supplementary Figure 18 itself. The authors should offer more detailed guidance on interpreting their improved ratiometric characterization and clarify the rationale behind their method's enhanced accuracy. Specifically, they need to explain why the pH value of 4.5, extracted using their method, represents an improvement over the conventional method's estimated pH of 6.
3. In response to Comment 12 of Reviewer #2, the authors mentioned that their Nile red measurement in their cellular blebbing experiment "did not involve a single Nile Red molecule but rather an ensemble of molecules." However, this clarification is not explicitly stated in the revised manuscript. The authors should make this point clear.
4. The caption of Supplementary Figure 20 states that "The solid lines represent Gaussian fits...", yet no such lines appear in the figure. The authors need to review this figure carefully and correct any discrepancies.

(Remarks on code availability)

Reviewer #3

(Remarks to the Author)

I appreciate the many careful revisions the authors have completed, especially the addition of methodological detail, the revisions to main paper figures and text, and the new SI figures to help the reader understand this interesting work.

My major concern remains the issue of spectral accuracy in the data presented. It is very clear that the authors have achieved excellent precision in their chosen output parameter, the spectral centroid, but throughout the work I note many instances where it seems that the accuracy of the centroid – and by extension, the measured spectrum itself – is very much in question. These concerns appear as a repeated theme in the detailed notes and questions below; I strongly encourage the authors to work to address this issue prior to publication if the editorial team concurs.

The other major concern may be easily fixed as I believe it to be due to an oversight: The authors have not, in fact, made the code available on their GitHub (this may be due to a misunderstanding of privacy settings on the repository, or similar). In any case, at the time of this review, it is not available at the cited link (<https://github.com/houlab/3D-SpecDIM>), nor is there any indication that it exists on the publicly available version of the houlab.github.io page. It is also not included in the .zip file for reviewers. While the improved description of the code and the comparisons to other methods are decent improvements from the previous version of the manuscript, I was not able to evaluate the code.

Finally, I note that the other reviewers shared my concerns that the example applications do not fully highlight the future utility of this technique. I do appreciate the expanded single-molecule data and analysis; I think this has helped somewhat. I also appreciate the wording changes that more carefully contextualize the present demonstrations relative to claims of single-molecule level applications that are not demonstrated here; in short, here I leave the determination of suitability to the editorial team.

Detailed comments / questions: Most importantly, the characterization of the system's accuracy remains in my opinion inadequate. The new SI Fig. 25 does not inspire confidence that their calibrations correctly reproduce spectra; systematic issues are clearly evident there (see specific comments below). Moreover, it is imperative that their reported parameter of choice, the "spectral centroid" be accurate, and not just precise. It seems to me that the centroids reported in this work are not accurate even though they clearly achieve excellent precision. This can be easily checked by comparing to predicted and measured centroids of bulk spectra; again, I give some comparisons and examples in the specific comments below.

- Reviewer 1 asked whether targets moving out of focus could compromise the accuracy of spectral estimation: Based on the new data in SI Fig 1b, it is clear that the accuracy is indeed compromised; systematic drifts in the centroid by more than 15 nm are evident in the first third of the trace alone. While some of this drift might be attributable to X-Y movement within the detection region, most of it should be calibrated out, right? Or, if it is due to X-Y movement, then this calibration needs to be revisited to eliminate spatial dependence. Also, the authors' responses here really only discuss the "precision" with Z-defocus, and barely mention accuracy.

- Notably, the new SI Fig. 1 also raises some new questions about the analysis software; it is surprising that in much of 1b "the signal-to-noise ratio was too low to obtain a valid spectral centroid estimation"... yet this is for a bright bead; surely a single-molecule signal would have far worse SNR. Moreover, the SNR looks fine by eye and a centroid could be computed analytically (without a NN).

- What is the accuracy of the calculated spectral centroids for each of the single dyes? Specifically: In Fig. 2k, the average spectral centroids of the individual dyes are reported as 593 nm (ATTO 565), 682 nm (ATTO 665), and 688 nm (Setau 647). A quick inspection of the published spectra for these dyes illustrates that none of the measured centroids reported in this paper are accurate: ATTO 565's centroid should be around 612 nm (see emission spectrum at <https://www.atto-tec.com/ATTO-565.html>); ATTO 665's centroid should be around 702 nm (spectrum: <https://www.atto-tec.com/ATTO-665.html>), and Setau647's centroid should be around 693 nm (<https://app.fluorofinder.com/dyes/1531-setau-647-nhs-ex-max-649-nm-em-max-695-nm>). The centroids reported in the paper disagree with the published spectra by up to -20 nm; this absolutely must be addressed. An additional note on this: If, as is often the case, the emission filters are set to eliminate the bluer side of the emission spectrum, then one would expect the experimentally determined centroids to skew red, not blue. Again, this highlights the major issue of accuracy vs. precision in this work.

- I am not clear on what SI Fig. 8 is showing. Is the blue line simply an inverse relationship between time and # Setau molecules to reach a 5nm tracking precision (seems most likely, but unclear)? Is it some sort of a fit to all of the data? Is this figure meant to illustrate that most experimental conditions were more favorable for tracking than the theoretical minimum? (If so, do they each produce better than 5 nm precision, as predicted?)

- SI Fig. 10 should show the kymograms of the spectra / localization for each example molecule.

- SI Fig. 11 is a nice demonstration of single-fluorophore blinking, but shows surprisingly low brightness per fluorophore. According to Fig. 2m, Atto 565 should have ~17kHz brightness (assuming the same excitation parameters). Yet in SI Fig 11 its brightness is more like 10 kHz. In the response to Reviewer #2 the authors state that the power was reduced; this should be documented properly in the SI / methods to explain this discrepancy.

- SI Fig. 11 should also show the raw kymogram of the spectrum, and a trace of the centroid, to illustrate that it does not change when one molecule blinks.

- SI Fig. 25: It is still not clear whether the authors account for bin stretching (more wavelengths are contained in the redder bins, translating to apparent higher counts in red bins in the raw data. Based on the consistent errors with higher measured red shoulder counts evident in Fig. SI 25d, it appears to me that the authors are not accounting for this stretch when transforming their spectra to wavelength space. This issue exacerbates inaccuracies in spectral measurement.

- On a related note, I continue to be concerned that not accounting for spectrally-dependent camera detection efficiencies harms the accuracy of the measured spectra.

- In SI Fig. 25d, the yellow fluorescent bead's measured spectrum does not match the bulk measurement, showing two distinct peaks rather than a shoulder. Why?

- SI Fig. 30: Why should the intensity fluctuations (often up to a factor of 2 or 3x) be attributed to changes in photon collection efficiency? Are they correlated to Z position? If so, this point needs to be thoroughly explained, especially in light of the "example" traces shown for tracking single dyes in SI Figs. 10 and 11, which do not fluctuate in this way despite changing Z position. In particular, SI Fig. 30b is troubling, as this looks like blinking of a multi-dye cluster and does not appear to correlate to positional changes seen in the main figure localization trace.

- o Additionally, SI Fig. 30b seems to show 6 frames per second rather than the stated 100ms (10 frames per second) exposure for single molecule experiments.

- o The 6 frames per second issue is also evident in SI movie #5.

- SI Fig. 31: These kymograms appear to be peak-normalized; they would be far better presented with the original scaling to allow the reader to more readily evaluate changes in signal-to-noise and brightness from the raw data.

- o In particular, because of the peak-normalization, it is difficult to understand why certain frames of SI Fig. 31b fail while others that appear to have worse SNR produce a result.

- o Can the authors comment on why the pH ratio appears to change / fluctuate between high and low so rapidly during the transition? In the kymogram shown in SI Fig. 31c, these "switch" events are even more evident than they originally appear in Fig. 3f. It almost looks like blinking of one or both channels, but this doesn't make sense because this is not close to single-molecule level...

- SI movies should include the kymograms (not peak-normalized).

(Remarks on code availability)

Code was not available at the stated link.

Version 2:

Reviewer comments:

Reviewer #3

(Remarks to the Author)

I appreciate the authors' publication of the code, as well as many of their manuscript modifications.

However: I continue to be very concerned that this approach yields inaccurate centroid values. In my previous comments, I provided a few centroid values that were calculated directly from reference spectra from the websites listed. The authors' response instead compared their measured centroid values to the PEAK values from these reference spectra. As I'm sure the authors are aware, the centroid of a distribution is not necessarily the same as the location of its peak, and therefore it is not a valid comparison to compare centroid values to peak values. In the case of these reference emission spectra, the centroids are as much as 20 nm away from the peak values. Therefore, the new stated error of " ± 3.48 nm" and the new SI Table 7 are completely incorrect and need to be revised to compare to reference spectra centroids, NOT peak values.

So, my previous major concern stands: the reported centroids are not close to matching reference spectra centroids, so this method appears to be rather inaccurate despite its admirable precision.

(Remarks on code availability)

Verified availability but no time for full review.

Reviewer #1 (Remarks to the Author):

Sha et al. present 3D-SpecDIM, a novel single-molecule imaging technique enabling simultaneous 3D positional tracking and spectral dynamics monitoring of biomolecules, which allows for real-time, high-resolution capture of molecular behavior and physicochemical changes. By employing an active 3D target-locking mechanism, the authors effectively address key challenges in single-molecule fluorescence spectroscopy, maintaining focus on rapidly moving molecules to ensure stable, high-speed spectral acquisition with single-molecule sensitivity. This work's significance lies in its capacity to provide real-time, multiparameter insights, as demonstrated in applications such as mitophagy tracking and cellular blebbing analysis. The integration of 3D tracking with spectral dynamics makes 3D-SpecDIM a valuable tool in the field, and I recommend this paper for publication.

Response: Thank you very much for your professional review and recommendation.

Comments:

1. Choice of AI Model: Why was a transformer model chosen for this application? The primary purpose here seems to be spectral denoising, as spectra from limited photon counts can be noisy, which requires effective denoising algorithms. In microscopy, similar denoising goals are often met with architectures such as U-Net or Autoencoders, which may provide more efficient solutions. Additionally, since the spectral data can effectively be represented as a 1D array (e.g., intensity vs. wavelength), would a simpler neuron network model—rather than a vision model like a transformer or CNN—be feasible?

Response: Thank you for your insightful comment. The Vision Transformer (ViT) model utilized in this study is not designed for spectral denoising. Instead,

its primary purpose is to identify spectral features, with a specific emphasis on determining the spectral centroid. Specifically, the model processes a spectral image as input and generates the position of the spectral centroid as its output.

The choice of the Transformer model was motivated by its incorporation of positional encoding, which enhances its capacity to capture spatial relationships between pixels in the spectral image. This capability is especially advantageous for spectral identification tasks, as the positional information among pixels is crucial for accurately determining the spectral centroid.

We have conducted a comparative analysis of the performance of the ViT_d model and convolutional neural networks (CNNs) in the revised manuscript. The results demonstrate that ViT outperforms CNNs in spectral localization precision, particularly under high signal-to-noise ratio conditions (Fig. 2d).

We sincerely appreciate your attention to this important aspect, as it has provided us with an opportunity to further elaborate on the advantages of utilizing the Vision Transformer in our study.

Revision: In page 6, Line 6, the following sentences have been added:

Unlike conventional fitting methods, which rely on curve fitting for spectral centroid localization, the ViT_d model leverages positional encoding to capture spatial relationships within spectral images, thereby facilitating precise spectral centroid identification. Compared to the conventional normal Gaussian fitting method, the ViT_d-based approach improved spectral localization precision from 1.63 nm to 1.11 nm, representing a 32% enhancement (Fig. 2c, Supplementary Fig. 5 and Supplementary Fig. 6). Additionally, the performance of the ViT_d model was compared with that of a convolutional neural network (CNN) in terms of spectral localization precision (Fig. 2d). The ViT_d model shows improved performance over both the CNN and the normal fitting method. It should be noted that the ViT_d-based spectral detection method is more effective in high photon counts situations than in low photon

counts conditions due to the influence of the signal-to-noise ratio. This improvement highlights the advantage of the positional encoding capability of ViT_d in spectral feature recognition.

The Fig. 2d has been modified accordingly:

2. *Software Availability: Open availability of the software would greatly facilitate reproducibility and the broader application of the methodology, and the authors are encouraged to consider making it accessible.*

Response: Thank you for your comment. We think this is a great suggestion. The SpecViT code now has been made publicly available on GitHub, along with comprehensive documentation. The GitHub repository can be accessed at: <https://github.com/houlab/3D-SpecDIM>. Additionally, the workflow of the 3D-SpecDIM system, illustrating the logic of the spectral tracking program, is presented in **Supplementary Figure 4**. A detailed, step-by-step description of the workflow, including the data acquisition and preprocessing pipeline, is provided in **Supplementary Note 3**.

Revision: **Supplementary Figure 4** and **Supplementary Note 3** have been included in the revised manuscript. Additionally, the SpecViT code has been made publicly available on GitHub.

3. *Switching Between Targets: It is unclear if target switching is actively controlled. Based on the data, it appears that the tracking system may randomly lock onto one target when two or more overlap within the focus. Could the*

authors clarify how the system handles tracking when multiple targets are present?

Response: Thank you for your comment. This is an excellent question. During the tracking process, the system determines the target's position based on the arrival times of fluorescence photons that are detected by APD detector, whether there is one or multiple targets within the excitation volume. When two or more targets are present, the system localizes to the centroid of their combined fluorescence emissions. As a result, the tracking system tends to lock onto the target with the higher fluorescence emission rate. Practically, the density of molecules or particles is maintained at a low level to minimize the presence of multiple targets in the excitation volume. We agree that this point requires clarification, and we have added the relevant details to the methods section.

Revision: In Page 14, Line 1, the following sentences have been added:

“If two or more targets are present within the excitation volume, the tracking system localizes to the centroid of their combined fluorescence emissions. As a result, the tracking system tends to lock onto the target with the higher fluorescence emission rate. Practically, the density of molecules or particles should be maintained at a low level to minimize the presence of multiple targets in the excitation volume.”

4. Spectral Accuracy vs. Defocusing: Could spectral accuracy be affected by defocusing, specifically errors in Z-axis tracking? For example, if dye molecules move out of focus, the spectral image may become spread or blurred, potentially impacting spectral decomposition accuracy.

Response: We appreciate your insightful comment. We think this is a great question. Defocusing can significantly reduce spectral precision, which in turn

impacts spectral accuracy. To evaluate the effect of defocusing on spectral precision, we conducted two assessments.

First, with 3D target-locking tracking, we investigated how axial positional tracking precision influences spectral precision. To do this, we deliberately adjusted the feedback control parameters of the axial piezo stage to alter the Z-axis localization precision during tracking and assessed the corresponding spectral localization precision. As shown in **Supplementary Fig. 1a**, when the Z-axis tracking precision degraded from 50 nm to 450 nm, the spectral localization precision declined from 0.57 nm to 1.39 nm, demonstrating the dependence of spectral precision on Z-axis tracking performance.

Second, in the absence of 3D target-locking tracking, we evaluated how spectral localization precision varies as fluorescent beads move out of focus (**Supplementary Fig. 1b-c**). The results indicate that as the Z-axis displacement increases from 0 to ± 1.2 μm , the spectral localization precision decreases from 0.6 nm to approximately 2.6 nm at a Z defocus distance of -1.2 μm and 2.1 nm at a Z defocus distance of +1.2 μm .

It is important to emphasize that during spectral dynamics imaging with 3D-SpecDIM, the target particle is actively maintained within the focal plane through target-locking tracking. This approach effectively minimizes Z-axis defocusing during tracking, thereby ensuring high spectral accuracy.

Thank you for highlighting this point, as it provides an opportunity to further demonstrate the advantages of 3D-SpecDIM.

Revision: Supplementary Figure 1 has been added:

Supplementary Figure 1. The impact of defocusing on spectral precision. (a) Z-axis localization precision as a function of spectral localization precision. Fixed fluorescent beads were tracked while systematically adjusting the feedback control parameters of the axial piezo stage to modify localization precision. Localization precisions were calculated with 1 second window. (b) Upper panel: kymograms of spectral images of a freely diffusing fluorescent bead in an aqueous solution under conditions without target-locking tracking. Lower panel: the spectral centroid as a function of frame number, with only successfully fitted data points shown. At certain time points, the signal-to-noise ratio was too low to obtain a valid spectral centroid estimation. (c) Defocusing distance as a function of spectral localization precision. A fixed fluorescent particle was imaged at various axial positions. As the Z defocusing distance increased from 0 to $\pm 1.2 \mu\text{m}$, the spectral localization precision decreased from 0.6 nm to approximately 2.6 nm at a Z defocusing distance of $-1.2 \mu\text{m}$ and 2.1 nm at $1.2 \mu\text{m}$. (d) Impact of motion blur on subpixel positional precision of

EMCCD image with target-locking tracking. Data represents the trajectories shown in Fig. 2a-c. Errors were calculated by subtracting the average x- and y-pixel localization values across all frames from the x- and y-localization values in each individual EMCCD frame.

5. *Measurement Uncertainty: The error bars on the diffusion coefficient and the estimated single-molecule diameter are missing.*

Response: Thank you for your comment. In the mean square displacement (MSD) plot of a single trajectory, each data point includes an associated error bar. However, due to the high time resolution of 1 ms during trajectory data collection, the large number of data points caused the error bars to overlap, obscuring the clarity of the MSD plot. To address this issue, we have replaced the error bars with an error envelope in the revised manuscript, providing a clearer and more visually accessible representation of the fitting results.

The diffusion coefficient is derived by fitting the MSD curve to the following equation:

$$MSD = 2 n D t,$$

where n is the dimensionality of diffusion (in our case, $n=3$), t is the time interval, and D is the diffusion coefficient. According to the Einstein-Stokes equation:

$$D =$$

$$\frac{k_B T}{6 \pi \eta r},$$

where k_B is the Boltzmann constant, T is the absolute temperature, η is the dynamic viscosity of the medium, and r is the hydrodynamic radius of the molecule, we can calculate the hydrodynamic radius of the molecule. To evaluate the fitting error of diffusion coefficient, we have calculated the coefficient of determination (R^2) between the trajectory data and the standard Brownian motion model. The calculated R^2 value is 0.9838, demonstrating excellent agreement with the model.

Furthermore, we conducted a statistical analysis of the hydrodynamic diameter, spectral centroid, tracking durations, and emission rates of three different fluorophores: Atto 665 ($N = 33$), SeTau 647 ($N = 35$), and Atto 565 ($N = 26$), as shown in Fig. 2j-m and the error bars have been provided.

Revision: Fig. 2i has been modified and Fig. 2j-m has been added to the revised manuscript:

(g, h) 3D Moving Trajectory of a single SeTau 647 fluorescent molecule diffusing in 90 wt.% glycerol. The trajectory is color-coded to represent either the spectrum (g) or time (h). The inset in (g) displays the fluorescence spectral image captured by the EMCCD and the corresponding spectral profile at the current time point. (i) Mean square displacement (MSD) as a function of lag time for the trajectory shown in (g). The red dashed line represents a linear fit to the MSD, yielding a diffusion coefficient of $2.67 \mu\text{m}^2/\text{s}$. The calculated

hydrodynamic diameter of the molecule is 0.89 nm, which aligns with the expected size of a single fluorophore. **(j-m)** Statistical analyses of the hydrodynamic diameter (**j**), spectral centroid (**k**), tracking durations (**l**), and emission rates (**m**) of three different fluorophores: Atto 665 (N=33), SeTau 647 (N=35), and Atto 565 (N=26).

In Supplementary Information file, Supplementary Notes 5 has been added:

"Supplementary Notes 5. Calculation of diffusion coefficient.

The diffusion coefficient is derived by fitting the MSD curve to the following relationship:

$$MSD = 2nD t,$$

where n is the dimensionality of diffusion (in our case, $n=3$), t is the time interval, and D is the diffusion coefficient. According to the Einstein-Stokes equation:

$$D = \frac{k_B T}{6\pi\eta r},$$

where k_B is the Boltzmann constant, T is the absolute temperature, η is the dynamic viscosity of the medium, and r is the hydrodynamic radius of the molecule, we can calculate the hydrodynamic radius of the molecule."

6. Potential Cytotoxic Effects of Silver Nanoparticles: The authors suggest that silver nanoparticles might enhance local fields, leading to photodamage in cells. Since silver is known to act as a photocatalyst, it may also facilitate chemical reactions that are toxic to cells. Au is more inert. Further elaboration on potential catalytic effects would be beneficial.

Response: Thank you for your comment. We think this is a great question. As indicated in Supplementary Table 2, our results demonstrate that cellular

blebbing in this study is predominantly induced by Nile Red staining in combination with high-intensity blue laser illumination.

In the comparative experiment, three conditions were examined: (1) cells treated with both AgNPs and Nile Red, (2) cells treated with Nile Red alone, and (3) cells treated with AgNPs alone. Cellular blebbing was observed in conditions (1) and (2), but not in (3), indicating that AgNPs alone do not induce blebbing.

Moreover, in an additional experiment, cells treated with Nile Red under low intensity blue laser illumination did not exhibit blebbing, highlighting that high intensity blue laser illumination is essential for blebbing induction. The role of AgNPs in this context is to enhance the local light density near their surfaces through the plasmonic enhancement of the electromagnetic field, which facilitates cellular blebbing at the sites where AgNPs are located.

We agree with you that silver nanoparticles are effective photocatalysts. Notably, in our experiments, only a few AgNPs were located on the cell membrane, and we did not observe phototoxicity induced blebbing. However, we agree that further clarification would be beneficial. We have revised the manuscript accordingly. Thank you for your valuable suggestion.

Revision: In page 11 Line 15, the following sentences were added:

“The plasmonic enhancement of the electromagnetic field near the surface of AgNPs leads to an increase in local light density⁵⁸, which is a critical factor in initiating of cellular blebbing (**Supplementary Table 2**). As a result, cellular blebbing frequently initiates at sites where these nanoparticles are located (**Supplementary Movie 3**). It is noteworthy that, although AgNPs are effective photocatalysts, no phototoxicity-induced blebbing attributable to AgNPs alone was observed (**Supplementary Table 2**).”

Reviewer #2 (Remarks to the Author):

In the manuscript submitted to Nature Communications, Sha et al. present a three-dimensional spectrum dynamics imaging microscopy (3D-SpecDIM), building on their earlier work in real-time 3D tracking (Hou et al., Nature Communications, 11, 3607, 2020). This system, which utilizes a pair of electro-optic deflectors and a tunable acoustic gradient index lens, is applied to monitor pH changes during the mitophagy process in a cell and nanoparticle movement during laser-induced cellular blebbing. While the concept of simultaneous tracking of fluorescent spectra and 3D positioning of fluorophores is interesting, the system characterization and results do not support the authors' claims. The manuscript describes the technique as 'single-molecule' spectrum dynamics imaging, yet has problems to demonstrate in true single-molecule context and resolution. Furthermore, the advantages presented in the application sections by the proposed technology are not convincingly shown to be inherent to the technology itself but rather seem to derive from conventional work. Additionally, the claimed 3D tracking capability does not appear to be useful in discussion of the application demonstration. Detailed concerns and comments are outlined below.

Response: We are grateful for the time and effort that you devoted to providing feedback on our manuscript, and we sincerely appreciate the insightful comments and valuable improvements that were suggested.

We fully agree that additional data are required to further validate the single-molecule spectral tracking capability. We have incorporated additional single-molecule spectral tracking data into the revised manuscript. These include spectral tracking of three different types of fluorescent dyes, as well as spectral tracking of a two-dye-labeled BSA protein exhibiting photoblinking. These additions further demonstrate the single-molecule spectral tracking capability.

In the applications section, the advantages of 3D-SpecDIM are as follows:

- 1) 3D-SpecDIM enables high spatiotemporal precision in detecting spectral dynamics alongside 3D positional dynamics. Compared to conventional spectral imaging methods available in commercial microscopes, such as the Zeiss 980, 3D-SpecDIM achieves a temporal resolution for spectral detection that is enhanced by at least two orders of magnitude when imaging cellular events (Supplementary Table 6). The limited temporal resolution of spectral imaging in the Zeiss 980 renders it unsuitable for capturing such dynamic processes.
- 2) The target-locking tracking capability of 3D-SpecDIM maintains the object within the focal plane throughout the entire imaging period. This effectively eliminates out-of-focus issues caused by target movement, which are common with conventional methods.
- 3) 3D-SpecDIM enables multi-resolution imaging, allowing the simultaneous acquisition of 3D volumetric images during spectral imaging. This provides additional contextual information, enhancing the interpretation of the studied events.

In the mitophagy tracking and cellular blebbing-induced membrane polarity change experiments, the 3D target-locking tracking capability of 3D-SpecDIM effectively eliminates out-of-focus issues caused by target movement, thereby extending the observation time. The 3D target-locking tracking also enables multi-resolution imaging, which allows for the simultaneous acquisition of 3D positional and spectral dynamics of mitochondria, along with the volumetric imaging of lysosomes, marking, to the best of our knowledge, the first demonstration of such a capability. With 3D tracking, the dynamic changes in the 3D spatial location and the diffusion coefficient of the target object can be accurately determined. When combined with fluorescence spectral dynamics

detection, this capability facilitates a quantitative investigation of the cellular blebbing process.

We agree that the advantages of 3D-SpecDIM in these applications require further clarification. Accordingly, we have included additional data and analyses in the revised manuscript to address this point. Thank you for highlighting this point, as it has provided us with an opportunity to further elaborate on the advantages of 3D-SpecDIM.

Major comments

1. First and foremost, the manuscript only demonstrates what is described as 'single-molecule' imaging in the experiment shown in Fig. 2g-h. All other experiments, including the biological applications, do not focus on individual single fluorescent molecules but rather on ensembles of fluorophores or nanoparticles. Additionally, the only image provided in the inset of Fig. 2g, which is meant to support the single-molecule claim, does not convincingly show a single molecule. The image suggests an aggregation of molecules, as indicated by its brightness and the duration of tracking (6 seconds). Given these observations, the manuscript should not claim 'single-molecule' capabilities in its title without unequivocal evidence of single-molecule tracking and spectrum acquisition.

Response: Thank you for your insightful comment. We fully agree that additional data are necessary to validate the single-molecule spectral tracking capability.

To address this, we conducted experiments tracking three distinct fluorescent dyes in a 90% glycerol solution, including Atto 665, SeTau 647, and Atto 565, with approximately 30 trajectories recorded for each dye. We conducted statistical analysis of these trajectories and their corresponding spectral data

(**Fig. 2j–m, Supplementary Figure 10**). The hydrodynamic diameter of the tracked molecules, calculated using the Einstein-Stokes equation, was 1.28 ± 0.55 nm, with an average tracking duration of 5.47 ± 4.51 seconds. Notably, the measured hydrated diameter is consistent with previously reported single-molecule measurements [Nat Commun 11, 3607 (2020)], further confirming the single-molecule tracking capability of 3D-SpecDIM.

Furthermore, the single-molecule tracking capability was further validated through the observation of photoblinking. In this experiment, BSA protein was labeled with two Atto 565 fluorescent dyes and tracked using 3D-SpecDIM, where photoblinking of the dyes was observed (**Supplementary Figure 11**). The fluorescence intensity alternated between single-dye and two-dye intensity levels. To minimize photobleaching, the excitation laser power was slightly reduced in the experiment.

These results collectively validate the single-molecule spectral tracking capability of 3D-SpecDIM. We appreciate your comment, as it has provided an opportunity to further highlight and substantiate this capability.

Revision: In page 7 Line 28, the following sentences were added to the revised manuscript:

“To further demonstrate the single-molecule spectral tracking capability of 3D-SpecDIM, we conducted experiments tracking three distinct fluorescent dyes in a 90% glycerol solution, including Atto 665, SeTau 647, and Atto 565, with approximately 30 trajectories recorded for each dye (**Fig. 2j–m, Supplementary Fig. 10**). The hydrodynamic diameter of the tracked molecules, calculated using the Einstein-Stokes equation, was 1.28 ± 0.55 nm (**Fig. 2**), with a tracking duration of 5.47 ± 4.51 seconds (**Fig. 2l**). The spectral centroids (**Fig. 2k**) and fluorescence intensities (**Fig. 2m**) were also statistically analyzed. Notably, the measured hydrated diameter aligns with previously reported

single-molecule measurements [Nat Commun 11, 3607 (2020)], further confirming the single-molecule tracking capability of 3D-SpecDIM.

Furthermore, the single-molecule tracking capability was further validated through the observation of photoblinking. In this experiment, BSA protein labeled with two Atto 565 fluorescent dyes was tracked using 3D-SpecDIM, during which photoblinking of the dyes was observed (Supplementary Fig. 11). The fluorescence intensity alternated between single-dye and two-dye intensity levels. To minimize photobleaching, the excitation laser power was slightly reduced in the experiment. These results collectively validate the single-molecule spectral tracking capability of 3D-SpecDIM.”.

Fig 2j-m has been added to Fig. 2:

(j-m) Statistics of the hydrodynamic diameter (j), spectral centroid (k), tracking durations (l), and emission rates (m) of three different fluorophores: Atto 665

(N=33), SeTau 647 (N=35), and Atto 565 (N=26). The mean and standard deviation values are labeled on the histogram figure.

In the Supplementary Information file, Supplementary Figure 10 and Supplementary Figure 11 have been added:

Supplementary Figure 10. Single molecule spectral tracking of three distinct molecules. (a) Left: 3D moving trajectory of a single Atto 665 fluorescent molecule diffusing in 90 wt% glycerol. Right: spectral centroid (up panel) and intensity (bottom panel) as a function of time. (b) Left: 3D moving trajectory of a single Atto 565 fluorescent molecule diffusing in 90 wt% glycerol. Right: spectral centroid (up panel) and intensity (bottom panel) as a function of time. (c) Left: 3D moving trajectory of a single SeTau 647 fluorescent molecule diffusing in 90 wt% glycerol. Right: spectral centroid (up panel) and intensity (bottom panel) as a function of time. (d) Mean square

displacement (MSD) of Atto 665 (top), Atto 565 (middle), and Setau 647 (bottom) as a function of lag time for trajectory (a-c).

Supplementary Figure 11. Single molecule demonstration with photoblinking. (a) 3D trajectory of a two-Atto 565-labeled BSA protein diffusing in a 90 wt.% glycerol solution, with the trajectory color-coded to represent time. (b) Fluorescence intensity as a function of time for the trajectory shown in (a), demonstrating alternation between single-dye and two-dye intensity levels. (c) Mean square displacement as a function of lag time for the trajectory in (a). (d) Intensity distribution histogram for the trajectory shown in (a). The histogram displays two distinct peaks: the higher peak corresponds to both dyes being in the emissive state, while the lower peak reflects one dye transitioning to a dark state

2. The manuscript reports that the authors' system can achieve a frame rate of up to 644 fps, corresponding to an exposure time of 1.55 ms. However, I am skeptical that the system can collect sufficient photons from single molecules within this brief exposure time to accurately determine 3D positions and spectral data. The description of the system's hardware capabilities does not convincingly support the performance claims necessary for single-molecule studies.

Response: Thank you for your comment.

The reported spectral imaging temporal resolution of 1.55 ms refers to the theoretical maximum temporal resolution of the system, determined primarily by hardware performance factors such as shutter transfer time and readout time (**Supplementary Figure 7**), and does not account for photon limitations. When the exposure time is reduced to as low as 10 μ s, the system still requires 1.55 ms to complete data readout, defining the intrinsic time resolution limit under conditions with sufficient photon flux, such as in the tracking of larger particles.

We acknowledge that detecting weak single-molecule fluorescence spectra within a 1.55 ms exposure time is not feasible. Therefore, in our single-molecule spectral imaging experiments, the EMCCD exposure time is set to 100 ms to ensure adequate photon collection.

We apologize for any confusion caused by the description in our manuscript. In response to your insightful comment, we have revised and clarified this point in the updated manuscript. Furthermore, we have included a detailed analysis of the relationship between the number of fluorophores and exposure time in response to a related question. We sincerely thank you for highlighting this important issue.

Revision: In Page 5 Line 24, the following sentences have been revised:

“This capability effectively enhances spectral acquisition speed, with a theoretical maximum frame rate of 644 fps.”.

In Page 6 Line 28, the following sentences have been added:

“The spectral imaging temporal resolution was evaluated by analyzing the spectral localization precision at varying camera exposure times while tracking a 200 nm fluorescent bead (**Fig. 2e**). The maximum achievable temporal resolution is fundamentally limited by hardware performance factors, such as shutter transfer time and readout time (**Supplementary Fig. 7**). However, the practical temporal resolution depends on the photon emission rate of the sample. For instance, when tracking a 200 nm fluorescent bead, an acceptable spectral localization precision can be achieved with a camera exposure time of 1 ms. Given the system’s minimum acquisition time of 1.55 ms (**Supplementary Table 1**), the corresponding spectral imaging temporal resolution is 2.55 ms in this scenario.

To assess the sensitivity of spectral tracking, varying exposure times and excitation intensities were applied (**Fig. 2f**). The fluorescence photon count rate measured by the APD was used to quantify the fluorescence emission rate. As shown in **Fig. 2f** and **Supplementary Fig. 8**, increasing the exposure time enhances spectral localization precision by facilitating the collection of more photons per frame in the EMCCD spectral images.”.

3. On page 5, line 125, the authors state that their spectral detection method shows ‘a significant improvement’. This assertion seems to be based on achieving a 0.52 nm smaller standard deviation using their Vision Transformer method, as shown in Fig. 2c. It is debatable whether this difference is statistically significant.

Response: Thank you for your valuable comment. We have revised the description to enhance the objectivity.

Revision: In Page 6 Line 9, the following sentences have been revised:

“Compared to the conventional normal Gaussian fitting method, the ViT_d-based approach improved spectral localization precision from 1.63 nm to 1.11 nm, representing a 32% enhancement (Fig. 2c, Supplementary Fig. 5 and Supplementary Fig. 6).”.

4. In Fig. 2d-f, the authors use p 'recision (nm)' on the y-axes. If this is equivalent to the sigma mentioned in Fig. 2c, the purpose of these plots needs further clarification. The photon counts reported exceed those from fluorescent beads used in experiments Fig. 2a-c, which raises questions about the method's compatibility with single-molecule signals. Moreover, a 100 ms exposure time is used here, which significantly contradicts their claimed operational time of 1.55 ms.

Response: Thank you for your insightful comments. We apologize for any confusion caused by the lack of sufficient detail in the manuscript.

The precision shown in Fig. 2d–f is equivalent to the sigma presented in Fig. 2c, obtained by calculating the standard deviation of the localized spectral centroid positions. It is used to quantify how much the spectral shifts can be resolved in a continuous spectral measurement.

The purpose of Fig. 2d-f is to characterize the system's performance, including its spectral localization precision, spectral imaging temporal resolution and sensitivity. These relationship plots serve as a reference for optimizing imaging parameters, such as excitation intensity and EMCCD exposure time, according to the fluorescence emission rate of the sample and the specific requirements of the experiment. Thank you for pointing out this. We have further clarified it in the revised manuscript.

The single-molecule fluorescence signal is indeed significantly weaker than that of fluorescent beads, as shown in Fig. 2m. However, the single-molecule fluorescence intensity remains above the detection sensitivity of the system (Fig. 2f). The spectral tracking of three distinct single fluorescent dyes, along with the spectral tracking of a two-dye-labeled BSA protein exhibiting photoblinking, further validates the system's single-molecule spectral tracking capability (Fig. 2j–m, Supplementary Figure 10, and Supplementary Figure 11).

In single-molecule spectral tracking experiments, the EMCCD exposure time is set to 100 ms to ensure the collection of sufficient photons in each spectral image frame. The 1.55 ms value represents the theoretical maximum temporal resolution of the system, primarily constrained by hardware performance factors such as shutter transfer time and readout time (Supplementary Figure 7), independent of photon count limitations. We apologize again for any confusion caused by the lack of sufficient detail in the manuscript and have clarified this point in the revised version.

Revision: In Page 6 Line 21, the following sentences have been revised:

“The precision of fluorescence spectral localization was assessed through the standard deviation of the localized spectral centroid position, providing a measure of the system's capability to distinguish spectral shifts in a continuous spectral measurement. The spectral localization precision was evaluated across different photon count conditions using the normal fitting method, the CNN model, and the ViT_d model. Among these approaches, the ViT_d model demonstrated superior performance, achieving a precision of 0.3 nm with 30,000 photon counts for this sample (**Fig. 2d**).”.

In Page 7 Line 15, the following sentences have been revised:

“As shown in **Fig. 2d-f**, an inherent trade-off exists between spectral localization precision and spectral imaging temporal resolution, both of which are influenced by the fluorescence photon count rate. These relationship plots serve as a valuable reference for optimizing imaging parameters, such as excitation intensity and EMCCD exposure time, based on the fluorescence emission rate of the sample and the specific requirements of the experiment.”

In Page 6 Line 28, the following sentences have been added:

“The spectral imaging temporal resolution was evaluated by analyzing the spectral localization precision at varying camera exposure times while tracking a 200 nm fluorescent bead (**Fig. 2e**). The maximum achievable temporal resolution is fundamentally limited by hardware performance factors, such as shutter transfer time and readout time (**Supplementary Fig. 7**). However, the practical temporal resolution depends on the photon emission rate of the sample. For instance, when tracking a 200 nm fluorescent bead, an acceptable spectral localization precision can be achieved with a camera exposure time of 1 ms. Given the system’s minimum acquisition time of 1.55 ms (**Supplementary Table 1**), the corresponding spectral imaging temporal resolution is 2.55 ms in this scenario.”.

5. Regarding Fig. 2f, it is unclear how this figure supports the authors’ claim of a minimum 7 kHz emission rate, particularly since the exposure time used significantly exceeds the 1.55 ms discussed in the main text. The origin of the 7 kHz rate is also not clearly elucidated.

Response: Thank you for your comment. We sincerely apologize for any confusion caused by the description of **Fig. 2f**.

In the fluorescence detection pathway, a 50:50 non-polarized beamsplitter was used to equally divide the emitted light between the APD and EMCCD. In the previous version of Fig. 2f, the emission rate was defined as:

$$Emission\ rate = c\ d_e,$$

where C represents the number of photons detected by the APD per second, and d_e denotes the APD's detection efficiency. The origin of the 7 kHz rate corresponds to the data point with the smallest x-axis position in Fig. 2f. This value indicates that spectral tracking can be achieved when a minimum of 7 kHz fluorescence signal goes to the APD.

We acknowledge that the use of "emission rate" may have caused confusion. Since the fluorescence reaching the APD is equivalent to that reaching the EMCCD, we have replaced it with "APD signal" in the revised manuscript to more clearly represent the fluorescence intensity of the target.

As shown in Fig. 2f, under similar APD signal conditions, increasing the exposure time improves spectral localization precision. As previously mentioned, the 1.55 ms value represents the theoretical maximum temporal resolution of the system and does not correspond to the temporal resolution in the case of Fig. 2f.

Revision: In Page 7 Line 7, the following sentences have been added:

“To assess the sensitivity of spectral tracking, varying exposure times and excitation intensities were applied (**Fig. 2f**). The fluorescence photon count rate measured by the APD was used to quantify the fluorescence emission rate. As shown in **Fig. 2f** and **Supplementary Fig. 8**, increasing the exposure time enhances spectral localization precision by facilitating the collection of more photons per frame in the EMCCD spectral images. The minimum APD signal required for successfully performing simultaneous 3D single molecule tracking

and spectral imaging is 5 kHz (**Fig. 2f**), comparable to the typical fluorescence signal of a single fluorophore.”

The caption of **Fig. 2e-f** has been modified:

“(e) Spectral localization error as a function of exposure time. The APD signal of fluorescent bead (the photon numbers captured by the APD per second) is 5 MHz. (f) Spectral localization precision as a function of APD signal with different exposure time.”.

6. I find it difficult to understand the support provided by Supplementary Table 1 for the claimed 1.55 ms temporal precision. I am skeptical that such a short exposure time can capture sufficient photons to accurately calculate spectral data. The limiting factor here is not merely the camera's acquisition speed, but the photon budget from the fluorescent molecules in the sample. The authors should provide an analysis of the photon yield from typical fluorophores used in their applications to substantiate their claims.

Response: Thank you for your comment and kind suggestion.

As previously mentioned, the 1.55 ms value represents the theoretical maximum temporal resolution of the system, determined by the camera's acquisition speed under the assumption of sufficient photon availability. For single-molecule spectral tracking, the temporal resolution is constrained by the low fluorescence photon emission rate, requiring an exposure time of 100 ms in this context. We tracked three distinct groups of dyes and conducted a statistical analysis of their fluorescence intensities (**Fig. 2m**). As shown in **Fig. 2g-m**, **Supplementary Figure 10**, and **Supplementary Figure 11**, single-molecule spectral information can be successfully acquired under this condition.

Additionally, we analyzed the relationship between exposure time and the number of SeTau 647 fluorophores required to achieve a minimum spectral localization error of 5 nm (Supplementary Figure 8). We also incorporated experimental configurations, including exposure time and fluorescence intensity, for tracking various fluorescent dyes and fluorescent beads in this manuscript into this plot. Here, fluorescence intensity was converted to the equivalent number of SeTau 647 molecules.

Revision: In the Supplementary Information file, Supplementary Figure 25 has been added:

Supplementary Figure 8. The relationship between exposure time and the number of SeTau 647 fluorophores required to achieve a minimum spectral localization error of 5 nm. The symbols represent the experimental configurations, including exposure time and fluorescence intensity, for tracking various fluorescent dyes and fluorescent beads in this paper. Here, fluorescence intensity was converted to the equivalent number of SeTau 647 molecules.

7. The authors should include data showing the trajectory of captured photon counts over time in all their 3D tracking experiments, particularly in the single-molecule demonstration presented in Fig. 2g-h. This information is critical for assessing the robustness and reliability of their tracking methodology.

Response: Thank you for your kind suggestion. We have provided the photon count changes over time for all tracking experiments presented in the main text, including the single-molecule demonstration shown in Fig. 2g-h.

Revision: In the Supplementary Information file, a Supplementary Figure 30 has been added:

Supplementary Figure 30. The APD signal as a function of time for Fig. 2a (a), Fig. 2 g-h (b), Fig. 3b-c (c) and Fig. 4b. It is worth noting that the intensity

fluctuations should be attributed to changes in the photon collection efficiency of the objective lens caused by variations in axial position.

8. Regarding Fig. 2g-h, it remains unclear how the authors distinguish between two closely located single molecules. The authors should provide multiple raw movies of the observed molecules to demonstrate the capability of their system in resolving individual molecules in close proximity without system confusion.

Response: Thank you for your comment. This is a good question. During the tracking process, the system determines the target's position based on the arrival times of fluorescence photons that detected by APD detector, whether there is one or multiple targets within the excitation volume. When two or more targets are present, the system localizes to the centroid of their combined fluorescence emissions. As a result, the tracking system tends to lock onto the target with the higher fluorescence emission rate. Practically, the density of molecules or particles is maintained at a low level to minimize the presence of multiple targets in the excitation volume.

We agree that this point requires further clarification, and we have added the relevant details to the methods section. Additionally, two supplementary movies have been provided.

Revision: In Page 14 Line 1, the following sentences have been added:

“If two or more targets are present within the excitation volume, the tracking system localizes to the centroid of their combined fluorescence emissions. As a result, the tracking system tends to lock onto the target with the higher fluorescence emission rate. Practically, the density of molecules or particles should be maintained at a low level to minimize the presence of multiple targets in the excitation volume.”

The following supplementary videos have been added:

Supplementary Video 4. Spectral tracking of single 200 nm fluorescent bead in water solution.

Supplementary Video 5. Spectral tracking of single SeTau 647 dye molecule in 90% glycerol solution.

9. The manuscript lacks a description of how the system tracks targeting fluorophores in 3D and manages the movement of the sample stage. It is unclear how the authors differentiate between positively and negatively defocused images to accurately control system movement. While this method may be detailed in their previous publication (Hou et al., Nature Communications, 11, 3607, 2020), clarity and completeness in the current manuscript are essential. The authors should explicitly describe the tracking method and algorithms used, ensuring the manuscript stands as an independent work.

Response: Thank you for your comment and we think this is a great suggestion.

The axial position of the target molecule is determined by correlating the current laser focus axial position with the fluorescence photon arrival times detected by the APD. Specifically, a TAG lens is employed to drive the laser focus in a sinusoidal motion along the axial direction. The laser focus position at any given time can be precisely calculated based on the output phase of the TAG lens. Upon detection of a fluorescence photon by the APD, its arrival time is used to correlate with the laser focus position, allowing for the estimation of the target molecule's axial position.

We have now provided a detailed description of the 3D tracking process and the algorithms used in the study in the Supplementary Information. This

includes schematics of the target-locking 3D single-molecule tracking mechanism, the workflow of the 3D-SpecDIM system, and a description of pixel-wavelength relationship calibration, single-molecule spectral tracking data acquisition, and description for custom-build 3D-SpecDIM LabVIEW program.

Revision: In the Supplementary Information file, **Supplementary Figure 3**, **Supplementary Figure 4** and **Supplementary Notes 3** have been added:

Supplementary Figure 3. Schematics of target-locking 3D single-molecule tracking. The laser focus is deflected in three dimensions by the EOD and TAG lens. The arrival times of fluorescence photons detected by the APD are utilized for real-time position estimation using an FPGA. Subsequently, an active feedback control signal is applied to the piezo stage to maintain the target fluorophore within the excitation volume.

Supplementary Figure 4. The workflow of the 3D-SpecDIM system. (a) Spectral data acquisition program. Real-time EMCCD images are displayed via a custom LabVIEW program, which automatically saves spectral images throughout the tracking session. **(b)** Spectral tracking feedback loop. The system tracks particles by updating positional estimates, converting them to voltage signals to control the piezo stage, and recording the piezo stage positions and spectral images.

Supplementary Notes 3. Workflow and details of 3D-SpecDIM tracking and spectral data acquisition

1. Pixel-wavelength relationship calibration

Fixed four-color fluorescent beads (Thermo Fisher Scientific, TetraSpeck™, T7279) were placed on the microscope, and the 3D-SpecDIM system was utilized to track a fluorescent bead immobilized on the coverslip, capturing its spectral information using the EMCCD. The resulting image was stitched into a 16×80 matrix, with the 16×16 region representing the non-dispersed spatial

channel, where the bead center was located at pixel coordinates (8, 8), and the remaining 16×64 region corresponding to the dispersed spectral channel for intensity analysis along pixel coordinates (**Supplementary Fig. 25a**). Using different filters (514/30 nm, 577/20 nm, 591/6 nm) and measuring the spectral channel positions of lasers (490.2 nm, 561.7 nm, 636.8 nm), six paired datasets of pixel shifts and spectral wavelengths were obtained (**Supplementary Fig 16b**). A quadratic polynomial fitting was then applied to derive the calibration function for converting pixel shifts into spectral wavelengths (**Supplementary Fig 16c**), allowing accurate conversion of intensity variations along pixel shifts into the true wavelength distribution.

To demonstrate the accuracy of spectral detection, we compared the spectra acquired using the 3D-SpecDIM system with those obtained from a commercial spectrofluorometer (FS5, Edinburgh Instruments). For the commercial spectrometer measurements, emission spectra were recorded for four different fluorophores: four-color fluorescent beads (Thermo Fisher Scientific, TetraSpeckTM, T7279), Atto 565 dyes, yellow fluorescent beads (FSSY002, Bangs Laboratories), and red fluorescent beads (FSFR002, Bangs Lab). As shown in Supplementary Figure 16d, the spectral data obtained with the 3D-SpecDIM system (circles) closely align with those measured by the commercial spectrometer (solid lines), demonstrating strong agreement. This consistency confirms that the 3D-SpecDIM system is able to accurately measure fluorescence spectra while providing the added advantage of high spatiotemporal resolution for single-molecule dynamics studies.

2. Single molecule spectral tracking data acquisition

The real-time 3D single molecule tracking relies on real-time position estimation and active feedback control to maintain the target fluorophore within the excitation volume (**Supplementary Figure. 3**). Specifically, A pair of electro-optic deflectors (EODs) were used to deflect the laser focus in the XY

plane, and a tunable acoustic gradient (TAG) lens was used to scan the laser focus in the Z axis. Fluorescence photons are collected by an avalanche photodiode (APD). The EODs were controlled with analogue voltage and the XY position of laser focus can be calculated by converting the voltage to distance. The TAG lens is employed to drive the laser focus in a sinusoidal motion along the axial direction. The laser focus position at any given time can be precisely calculated based on the output phase of the TAG lens. Upon detection of a fluorescence photon by the APD, its arrival time is used to correlate with the laser focus position, allowing for the estimation of the target molecule's position \mathbf{p}_k with a field-programmable gate array (FPGA).

The position \mathbf{p}_k is updated using a Kalman filter-based calculation, which integrates the prior position \mathbf{p}_{k-1} , the photon counts n_k , and the variance of the previous estimate σ_{k-1}

2. The formula for the updated position $\mathbf{p}_{k|k}$ is as follows:

$$\mathbf{p}_{k|k} = \frac{\mathbf{p}_{k|k-1} \cdot w^2 + c_k \cdot n_k \cdot \sigma_{k|k-1}}{w^2 + n_k \cdot \sigma_{k|k-1}} \quad (7)$$

where w is the laser beam covariance, and c_k is the current laser position. The variance update is calculated by:

$$\sigma_{k|k} = \frac{\sigma_{k|k-1}^2}{w^2 + n_k \cdot \sigma_{k|k-1}^2} \quad (8)$$

The following equations provide predictions for the position and variance, respectively:

$$\begin{aligned} \mathbf{p}_{k|k} &= \mathbf{p}_{k-1|k-1}, \sigma_{k|k-1} \\ \sigma_{k|k} &= \sigma_{k-1|k-1}^2 + 2D\tau, \end{aligned} \quad (9)$$

Where D is the expected diffusion coefficient and τ is the time between calculations (the bin time).

The workflow of the 3D-SpecDIM system is shown in **Supplementary Figure. 4**. For spectral acquisition, the camera settings must first be configured, including selecting the imaging region based on the tracked position, adjusting the cooling temperature, exposure time, and camera gain. Real-time EMCCD images are displayed through a custom-built LabVIEW program. Once tracking begins, the program automatically records and saves the spectral images until the tracking session concludes.

For the tracking process, the system initially drives the piezo to locate particles with fluorescence photon counts exceeding a predefined threshold. Upon capturing a single particle, the system updates the positional estimates (x_k , y_k , z_k) and converts these estimates into voltage signals to drive the piezo stage, ensuring the particle's position estimate is close to zero. It worth to note that the positional estimates (x_k , y_k , z_k) indicate the distances of molecule to the center of excitation volume in each direction. If the photon count falls below a certain threshold, tracking is interrupted. This threshold is typically set at twice the background photon count to maintain tracking stability.

If the photon count falls below a certain threshold, tracking is interrupted. This threshold is typically set at twice the background photon count to maintain tracking stability.

3. Description for custom-build 3D-SpecDIM LabVIEW program.

The 3D-SpecDIM LabVIEW program was developed based on the 3D-SMART program previously established by Prof. Kevin Welsher's lab. The primary modifications include configuring the camera parameters and synchronizing the EMCCD camera with the tracking system. The workflow of the program is illustrated in **Supplementary Figure. 4**

10. I am unconvinced that the ‘improved ratiometric fluorescence imaging’ demonstrated in the mitophagy application showcases the benefits of their 3D-SpecDIM. The improved ratiometric characterization largely results from spectral unmixing, a technique already well-established in prior publications. Additionally, the tracking of fluorescent molecule-labeled mitochondria does not add any interpretive value to the claims made by the authors in this context.

Response: We appreciate your comment and the opportunity to clarify the benefits of 3D-SpecDIM in ratiometric fluorescence imaging.

We would like to clarify that our work does not introduce a new spectral unmixing method. Rather, the mitophagy application is intended to demonstrate that 3D-SpecDIM can simultaneously capture both the three-dimensional positional dynamics and spectral dynamics of individual mitochondria.

Conventional ratiometric fluorescence imaging methods utilize two different band-pass filters to collect fluorescence signals across distinct wavelength ranges. The ratio between these two channels is then used for spectral characterization. However, due to emission spectrum crosstalk, the accuracy and sensitivity of these methods are compromised. In contrast, 3D-SpecDIM acquires the full fluorescence spectral profile, enabling spectral unmixing and thereby providing more accurate and sensitive measurements.

To further illustrate this point, we conducted a comparative analysis using the data from Fig. 3. In the dual-channel ratiometric detection method, pixel intensities on the EMCCD image were summed across two distinct wavelength ranges and then divided to calculate the intensity ratio. Compared to this conventional approach, 3D-SpecDIM-enabled spectral unmixing offers enhanced accuracy (**Supplementary Figure 18**). Specifically, the pH in lysosomes determined through spectral unmixing with 3D-SpecDIM is approximately 4.5, whereas the conventional dual-channel ratiometric detection method yields a value of 6.

The 3D tracking capability of 3D-SpecDIM offers two advantages in ratiometric detection of mitophagy. First, it maintains the mitochondrion within the excitation focal plane throughout the measurement, thereby extending the observation duration. Second, target-locking tracking ensures that only the target mitochondrion is imaged, effectively reducing spectral imaging time compared to a commercial spectral imaging microscope (**Supplementary Table 6**).

Revision: In the supplementary Information file, the following figure has been modified:

Supplementary Figure 18. Comparison of the 3D-SpecDIM-enabled spectral unmixing ratiometric fluorescence detection and the conventional dual-channel ratiometric detection. (a) The intensity ratio comparison of dual-channel ratiometric detection (yellow line) and 3D-SpecDIM-enabled spectral unmixing detection (blue line). **(b)** Similar comparison as in (a), with the corresponding mGold/JF549 intensity ratio converted to a calibrated pH. The pH range is scaled to start at 6.5.

11. The authors should also provide a clearer interpretation of Fig. 3g. They also need to clarify the meaning of the checkerboard pattern in the Lyso Int. colorbar.

Response: Thank you for your comment. We apologize for any lack of clarity in the original manuscript.

In Fig. 3g, the spectrally encoded 3D trajectory of the mitochondrion is overlaid on the 3D volumetric image of the lysosome. The blue-to-yellow color-coded trajectory represents the 3D movement of the mitochondrion, with the color encoding the pH ratio. Meanwhile, the 3D image of the lysosome is color-coded from violet to red, indicating fluorescence intensity. The 3D lysosome volume image was reconstructed by registering the fluorescence photons to laser focus positions in a separated detection channel. The detailed reconstructed information can be found in Supplementary Notes 4.

The checkerboard pattern represents transparency, while the alternating color blocks indicate different intensity levels. This visualization improves clarity and the intensity variations within the 3D image.

Revision: In caption of Fig. 3, the following sentence has been revised and added:

“The blue-to-yellow color-coded trajectory represents the 3D movement of the mitochondrion, with the color encoding the pH ratio. Meanwhile, the 3D image of the lysosome is color-coded from violet to red, indicating fluorescence intensity. The 3D lysosome volume image was reconstructed by registering the fluorescence photons to laser focus positions in a separated detection channel. Detailed reconstruction information is provided in **Supplementary Notes 4**. The checkerboard pattern represents transparency, while the alternating color blocks indicate different intensity levels.”.

12. The duration for which the authors tracked AgNP and Nile red molecules in their cellular blebbing experiment should be specified. The binding and dissociation kinetics of Nile red on and off the cell membrane are known to be

rapid. Based on the descriptions provided, it seems that the method resolved ensembles of Nile red molecules rather than individual molecules. If this is the case, tracking multiple molecules does not provide additional insights that could not be achieved with standard fluorescence imaging techniques, which can already map the changes in cell membrane blebbing and capture polarity shifts using a grating combined with standard imaging. Therefore, the tracking trajectories shown in Fig. 4b appear to be superfluous.

Response: Thank you for your comment and kind suggestions. We have now included a statistical analysis of the AgNPs tracking duration in the revised manuscript (**Supplementary Figure 20**). In this experiment, only AgNPs were tracked, while the fluorescence spectrum of Nile Red in proximity to the AgNPs was simultaneously monitored.

We focused on the membrane polarity in the vicinity of AgNPs, which can be inferred from the fluorescence spectrum of Nile Red. In this case, the detection did not involve a single Nile Red molecule but rather an ensemble of molecules within the $0.9 \times 0.85 \mu\text{m}$ excitation region that near the tracked AgNP.

Compared to traditional grating-based spectral imaging techniques, 3D-SpecDIM offers several advantages. The first is improved temporal resolution. In traditional grating-based spectral imaging, capturing cellular blebbing events requires imaging the entire cell. Acquiring a 2D spectral scanning image typically takes tens of seconds to several minutes (**Supplementary Table 6**), which may be insufficient to study the dynamic changes in blebbing. In contrast, 3D-SpecDIM employs AgNPs as probes to enhance the excitation field and induce cellular blebbing, greatly reduced the required imaging area. Compared to conventional grating-based spectral imaging methods, 3D-SpecDIM achieves several orders of magnitude improvement in the temporal resolution of spectral detection (**Supplementary Table 6**), enabling more precise measurements of blebbing dynamics.

The second advantage of 3D-SpecDIM is its enhanced spectral localization precision. In traditional grating-based spectral imaging techniques, spectral localization precision is constrained by the spectral scanning step size, which, for example, has a minimum of 3 nm in the Zeiss LSM 980 microscope. In contrast, 3D-SpecDIM achieved a spectral localization precision of 0.5 nm in this experiment, enabling the detection of more subtle spectral changes.

Additionally, with target-locking tracking, 3D-SpecDIM can maintain the observed bleb within the focal plane throughout the imaging process, enabling long-term continuous data acquisition. Since cellular blebbing is a highly dynamic process, traditional spectral imaging methods may struggle to capture sufficient data, as the blebbing site can move out of the imaging focal plane.

Furthermore, the 3D tracking capability of 3D-SpecDIM provides additional dimensional information, such as 3D positional dynamics. Specifically, in the cellular blebbing experiment, the 3D trajectory of the AgNP particle on the bleb offers further insights into the blebbing process. The AgNP exhibits greater displacement and velocity in the axial direction compared to the lateral direction, suggesting that it was elevated by the blebbing. Additionally, the AgNP's displacement correlates with fluorescence spectral centroid shifts (**Supplementary Figure 24f**), further highlighting the dynamic nature of the process.

In summary, the higher temporal resolution of spectral imaging, enhanced spectral localization precision, and simultaneous real-time 3D positional tracking capability enable 3D-SpecDIM to capture more subtle and rapid physicochemical and positional dynamics. This capability provides a more comprehensive understanding of the biological events under investigation.

Revision: In Page 11, Line 4, the following sentences have been modified:

Imaging the rapidly changing dynamics of cellular blebbing is challenging with conventional grating-based spectral scanning microscopes due to their limited temporal resolution and spectral localization precision (**Supplementary Table 6**). In contrast, 3D-SpecDIM offers high temporal resolution for spectral imaging, enhanced spectral localization precision, and simultaneous real-time 3D positional tracking, making it a powerful tool for studying fast dynamic processes.

In Page 11, Line 27, the following sentences have been modified:

This observation suggests that the AgNP was elevated by the expanding bleb, a dynamic process that would be difficult to capture in 2D spectral imaging using conventional spectral scanning microscopy.

In the Supplementary Information file, the figure and table have been added:

Supplementary Figure 20. The histogram distribution of laser irradiation time for blebbing (blue, n = 15) and no blebbing (orange, n = 8) events. The solid lines represent Gaussian fits to the distributions, with mean irradiation

times of $\mu=539.74$ s ($\sigma=486.65$ s) for blebbing events and $\mu=489.01$ s ($\sigma=449.72$ s) for no blebbing events.

Supplementary Table 6. Spectral detection performance comparison between 3D-SpecDIM and Zeiss LSM 980 spectral scanning confocal microscope.

	Spectral scanning step size/precision	Imaging size	Time resolution	Measuring dimensions
3D-SpecDIM	~0.5 nm	-	~0.1 s	3D
Zeiss-Commercial Microscope LSM980	3 nm (from 568 nm to 740 nm)	16.1×16.1 μm^2 (64×64 pix)	117 s	2D
		84.9×84.9 μm^2 (1024×1024 pix)	634 s	2D
	5 nm (from 568 nm to 740 nm)	16.1×16.1 μm^2 (64×64 pix)	46 s	2D
		84.9×84.9 μm^2 (1024×1024 pix)	202 s	2D

13. In the conclusion, the authors claim that their method ‘significantly improves the quantitative analysis of both the kinetics and dynamics of biological systems.’ However, the authors fail to demonstrate or discuss molecular kinetics, as the data presented appear to be from molecular

ensembles rather than single fluorescent molecules. This discrepancy between claims and demonstrated results undermines the validity of their conclusions.

Response: Thank you for your comment. We acknowledge the absence of single-molecule kinetics analysis in this manuscript. The biological applications presented here primarily focus on single-particle tracking, providing information on moving speed, displacement, and spectral change rates. However, the single-molecule tracking demonstration in this study highlights the potential of this method for single-molecule kinetics analysis when applied to appropriate biological systems. To address this discrepancy, we have revised the statement to: “provides the potential for quantitative analysis of both the kinetics and dynamics of biological systems”.

Revision: In Page 12, Line 16, the following sentence has been revised:

“Additionally, the continuous detection of spectral changes, coupled with 3D positional dynamics imaging, provides the potential for quantitative analysis of both the kinetics and dynamics of biological systems.”.

Minor comments

1. The manuscript does not define 'spec. cent.' despite its use. A description of what constitutes the centroid of the spectrum is necessary for understanding its relevance and how it is calculated within the context of the study.

Response: Thank you for pointing out this oversight. “Spec. cent.” refers to the spectral centroid, which is the wavelength corresponding to the maximum value of the fluorescence emission spectrum. We have now included a detailed explanation of this term in the manuscript.

Revision: In the caption of Fig. 2a, the following explanation has been added:

“Spec. cent.: Spectral centroid.”.

In the Methods section of main text, the following sentences have been added:

“Spectral centroid calculations

The "normal fitting method" refers to fitting spectral curves with a Gaussian distribution. Specifically, this method involves using all recorded intensity-wavelength pairs and applying a curve fitting algorithm to extract the parameters of the Gaussian profile. The spectral centroid corresponds to the position of the peak of the fitted Gaussian curve, representing the most likely wavelength of maximum emission.”.

2. The caption for Fig. 3e is misplaced, appearing after that of panel f. This should be corrected to maintain the logical order and readability of the figure captions.

Response: Thank you for pointing out this issue. We have carefully reviewed all figure captions in the manuscript and have corrected the misplaced caption for Fig. 3e to ensure logical order and improve readability.

Revision: In the caption of Fig. 3d and 3e, the following sentences have been modified:

(e) The fluorescence spectral profile changes during mitophagy process. The three spectral peaks correspond to mGold, JF549, and LysoTracker, respectively. (f) The pH ratio of pH-sensitive fluorescent probe (blue line) or the intensity of lysosome (yellow line) as a function of time. Note that after 40 s, pH decreased when lysosome strength increased.

Reviewer #3 (Remarks to the Author):

In this report, Sha et al. have added spectral detection to a single-particle tracking method, 3D-SMART. They call their new version “3D-SpecDIM”. The achievement of adding spectral imaging to 3D-SMART is a worthy one, and they have clearly made good progress in this direction. I would like to support publication of their results, but the present version of the manuscript is not convincing regarding the accuracy and utility of this exciting method, and I am concerned that it will not be accessible at all for a non-expert audience.

Response: We are grateful for the time and effort that you devoted to providing feedback on our manuscript, and we sincerely appreciate your support for the publication and your insightful comments.

1. First, I found this manuscript rather difficult to read, not for the language (although editing will be needed), but rather for the overall organization and clarity of the logical flow. The authors need to more clearly lay out what the challenges are and show definitively that their method overcomes them. For example, it seems to me that movement of the particle within the field of view, motion blur, and the confounding effects of knowing the stage’s position precisely are key technical challenges that should be more clearly explained and demonstrated. The latter two challenges – blur and stage movement effects on spectral measurements – are not addressed at all, as far as I can tell.

Response: Thank you for your comment. We apologize for the lack of clarity in the previous version of the manuscript and have reorganized the manuscript to improve clarity and readability.

We agree that the advantage of spectral imaging with 3D target-locking tracking requires further demonstration. In 3D-SpecDIM, the target is locked

to the excitation volume center via active feedback tracking. The role of the stage here is continuously repositioning the moving target to the center of excitation volume. In this way, the target keeps relatively stationary to the excitation volume and reduces the motion blur of the spectral image on EMCCD. The 3D positional tracking precision determines how well the target is locked to the excitation volume, which can affect the spectral localization precision. We have conducted an experiment to evaluate the influence of the axial tracking precision on spectral localization precision (**Supplementary Figure 1a**). The result shows that the spectral localization precision decreases with the axial tracking precision decrease. However, compared with spectral imaging without 3D target-locking tracking, the blur caused by the movement of the particle during the spectral acquisition time can be effectively reduced (**Supplementary Figure 1b**).

We also evaluated the defocusing caused spectral localization precision deterioration (**Supplementary Figure 1c**), further highlight the advantage of 3D target-locking tracking on improve spectral localization precision deterioration.

Revision: In Page 4, Line 1, the following sentences have been modified:

“The primary challenge lies in maintaining the molecule within the excitation focal volume while ensuring optical conjugation between its location and the spectral imaging plane during motion. Target-locking three-dimensional single-molecule tracking (TL-3D-SMT) offers an effective solution by continuously repositioning the target molecule at the center of the excitation volume through active feedback control³²⁻⁴⁶. In this work, we developed a single-molecule fluorescence spectral dynamics imaging method by integrating TL-3D-SMT with rapid fluorescence spectral detection, termed 3D-SpecDIM (**3D** target-locking-based single-molecule fluorescence **Spectral Dynamics**

Imaging Microscopy). In 3D-SpecDIM, motion-induced blur during spectral acquisition is significantly reduced through target-locking tracking, which keeps the target molecule relatively stationary within the excitation volume (**Supplementary Fig. 1**). By combining TL-3D-SMT with high-frame-rate spectral imaging, the system achieves high spectral precision and extended observation durations, even for rapidly moving targets. Moreover, this strategy effectively prevents the loss of fast-moving target molecules during imaging, ensuring continuous data acquisition.”.

In the Supplementary Information file, **Supplementary Figure 1** has been added:

Supplementary Figure 1. The impact of defocusing on spectral precision. (a) Z-axis localization precision as a function of spectral localization precision. Fixed fluorescent beads were tracked while systematically adjusting the feedback control parameters of the axial piezo stage to modify localization

precision. Localization precisions were calculated with 1 second window. **(b)** Upper panel: kymograms of spectral images of a freely diffusing fluorescent bead in an aqueous solution under conditions without target-locking tracking. Lower panel: the spectral centroid as a function of frame number, with only successfully fitted data points shown. At certain time points, the signal-to-noise ratio was too low to obtain a valid spectral centroid estimation. **(c)** Defocusing distance as a function of spectral localization precision. A fixed fluorescent particle was imaged at various axial positions. As the Z defocusing distance increased from 0 to $\pm 1.2 \mu\text{m}$, the spectral localization precision decreased from 0.6 nm to approximately 2.6 nm at a Z defocusing distance of $-1.2 \mu\text{m}$ and 2.1 nm at $1.2 \mu\text{m}$. **(d)** Impact of motion blur on subpixel positional precision of EMCCD image with target-locking tracking. Data represents the trajectories shown in Fig. 2a-c. Errors were calculated by subtracting the average x- and y-pixel localization values across all frames from the x- and y-localization values in each individual EMCCD frame.

2. The central aspect of the paper, details of how spectra are measured and the data analyzed, is remarkably difficult to parse from the writing and organization of the manuscript. Just as one minor example of the challenges in reading this paper – I am still not sure what, exactly, their neural net spits out in terms of the spectra. Are they plotting a peak? A centroid? I think it is a centroid, but I honestly cannot tell because they abbreviate to “Spect. Cent.” on plots and I cannot find an explicit statement in the text, captions, or SI. There are many such issues; I have listed a few in the minor comments section below.

Response: Thank you for your comment.

To clarify the spectral measurement and data analysis methods, we have added a detailed description in **Supplementary Figures 25** and **Supplementary Figures 3-4**, as well as **Supplementary Note 3** in the revised manuscript. This

includes information on hardware configuration, data acquisition workflow, spectral calibration, and the principles underlying spectral analysis.

We apologize for any confusion caused by the abbreviations and descriptions in the manuscript. To clarify, “Spec. Cent.” refers to the centroid of the fluorescence spectrum. We have now provided a clear definition of this term in the manuscript.

Regarding the neural network outputs, it indeed provides the spectral centroid directly after processing the raw spectral data. This allows for efficient spectral analysis and reduces the computational burden associated with fitting peak profiles. We have added a more detailed description of this process in **Supplementary Note 2**.

Revision: In the Supplementary Information file, **Supplementary Figure 25, Supplementary Figure 3-4, and Supplementary Notes 3** have been added to illustrate the spectral measurement and the data analysis pipeline:

Supplementary Figure 25. Spectral registration. **(a)** Top panel: the reference image (left) and spectral image (right) of four-colors fluorescent bead (Thermo Fisher Scientific, TetraSpeck™, T7279) on EMCCD. The spectral image was generated by horizontally flipping the raw spectral image. Bottom panel: the fluorescence intensity distribution versus pixel shift for spectral image. **(b)** Spectral calibration with different narrow bandpass filters and lasers. **(c)** Convert the pixel shift between the reference image and the spectral image into spectral wavelengths using quadratic polynomial fitting. **(d)** Comparison of spectra acquired with 3D-SpecDIM and a commercial spectrofluorometer. Four different fluorophores were analyzed: four-color fluorescent bead (Thermo Fisher Scientific, TetraSpeck™, T7279, blue line and circle), Atto 565 dye (yellow line and circle), yellow fluorescent bead (FSSY002, Bangs Lab, green

line and circle), and red fluorescent bead (FSFR002, Bangs Lab, red line and circle). The root mean square error (RMSE) between the spectral data from the two systems was calculated to be 0.23.

Supplementary Figure 3. Schematics of target-locking 3D single-molecule tracking.

The laser focus is deflected in three dimensions by the EOD and TAG lens. The arrival times of fluorescence photons detected by the APD are utilized for real-time position estimation using an FPGA. Subsequently, an active feedback control signal is applied to the piezo stage to maintain the target fluorophore within the excitation volume.

Supplementary Figure 4. The workflow of the 3D-SpecDIM system. (a) Spectral data acquisition program. Real-time EMCCD images are displayed via a custom LabVIEW program, which automatically saves spectral images throughout the tracking session. **(b)** Spectral tracking feedback loop. The system tracks particles by updating positional estimates, converting them to voltage signals to control the piezo stage, and recording the piezo stage positions and spectral images.

Supplementary Notes 3. Workflow and details of 3D-SpecDIM tracking and spectral data acquisition

1. Pixel-wavelength relationship calibration

Fixed four-color fluorescent beads (Thermo Fisher Scientific, TetraSpeck™, T7279) were placed on the microscope, and the 3D-SpecDIM system was utilized to track a fluorescent bead immobilized on the coverslip, capturing its spectral information using the EMCCD. The resulting image was stitched into a 16×80 matrix, with the 16×16 region representing the non-dispersed spatial

channel, where the bead center was located at pixel coordinates (8, 8), and the remaining 16×64 region corresponding to the dispersed spectral channel for intensity analysis along pixel coordinates (**Supplementary Fig. 25a**). Using different filters (514/30 nm, 577/20 nm, 591/6 nm) and measuring the spectral channel positions of lasers (490.2 nm, 561.7 nm, 636.8 nm), six paired datasets of pixel shifts and spectral wavelengths were obtained (**Supplementary Fig 16b**). A quadratic polynomial fitting was then applied to derive the calibration function for converting pixel shifts into spectral wavelengths (**Supplementary Fig 16c**), allowing accurate conversion of intensity variations along pixel shifts into the true wavelength distribution.

To demonstrate the accuracy of spectral detection, we compared the spectra acquired using the 3D-SpecDIM system with those obtained from a commercial spectrofluorometer (FS5, Edinburgh Instruments). For the commercial spectrometer measurements, emission spectra were recorded for four different fluorophores: four-color fluorescent beads (Thermo Fisher Scientific, TetraSpeck™, T7279), Atto 565 dyes, yellow fluorescent beads (FSSY002, Bangs Laboratories), and red fluorescent beads (FSFR002, Bangs Lab). As shown in Supplementary Figure 16d, the spectral data obtained with the 3D-SpecDIM system (circles) closely align with those measured by the commercial spectrometer (solid lines), demonstrating strong agreement. This consistency confirms that the 3D-SpecDIM system is able to accurately measure fluorescence spectra while providing the added advantage of high spatiotemporal resolution for single-molecule dynamics studies.

2. Single molecule spectral tracking data acquisition

The real-time 3D single molecule tracking relies on real-time position estimation and active feedback control to maintain the target fluorophore within the excitation volume (**Supplementary Figure. 3**). Specifically, A pair of electro-optic deflectors (EODs) were used to deflect the laser focus in the XY

plane, and a tunable acoustic gradient (TAG) lens was used to scan the laser focus in the Z axis. Fluorescence photons are collected by an avalanche photodiode (APD). The EODs were controlled with analogue voltage and the XY position of laser focus can be calculated by converting the voltage to distance. The TAG lens is employed to drive the laser focus in a sinusoidal motion along the axial direction. The laser focus position at any given time can be precisely calculated based on the output phase of the TAG lens. Upon detection of a fluorescence photon by the APD, its arrival time is used to correlate with the laser focus position, allowing for the estimation of the target molecule's position p_k with a field-programmable gate array (FPGA).

The position p_k is updated using a Kalman filter-based calculation, which integrates the prior position p_{k-1} , the photon counts n_k , and the variance of the previous estimate σ_{k-1}

2. The formula for the updated position $p_{k|k}$ is as follows:

$$p_{k|k} = \frac{p_{k|k-1} \cdot w^2 + c_k \cdot n_k \cdot \sigma_{k|k-1}}{w^2 + n_k \cdot \sigma_{k|k-1}} \quad (7)$$

where w is the laser beam covariance, and c_k is the current laser position. The variance update is calculated by:

$$\sigma_{k|k} = \frac{2 \cdot \sigma_{k|k-1}^2}{w^2 + n_k \cdot \sigma_{k|k-1}^2} \quad (8)$$

The following equations provide predictions for the position and variance, respectively:

$$p_{k|k} = p_{k-1|k-1}, \sigma_{k|k-1} \quad (9)$$

$$\sigma_{k|k} = \sigma_{k-1|k-1} + 2D \tau,$$

Where D is the expected diffusion coefficient and τ is the time between calculations (the bin time).

The workflow of the 3D-SpecDIM system is shown in **Supplementary Figure. 4**. For spectral acquisition, the camera settings must first be configured, including selecting the imaging region based on the tracked position, adjusting the cooling temperature, exposure time, and camera gain. Real-time EMCCD images are displayed through a custom-built LabVIEW program. Once tracking begins, the program automatically records and saves the spectral images until the tracking session concludes.

For the tracking process, the system initially drives the piezo to locate particles with fluorescence photon counts exceeding a predefined threshold. Upon capturing a single particle, the system updates the positional estimates (x_k, y_k, z_k) and converts these estimates into voltage signals to drive the piezo stage, ensuring the particle's position estimate is close to zero. It worth to note that the positional estimates (x_k, y_k, z_k) indicate the distances of molecule to the center of excitation volume in each direction. If the photon count falls below a certain threshold, tracking is interrupted. This threshold is typically set at twice the background photon count to maintain tracking stability.

If the photon count falls below a certain threshold, tracking is interrupted. This threshold is typically set at twice the background photon count to maintain tracking stability.

3. Description for custom-build 3D-SpecDIM LabVIEW program.

The 3D-SpecDIM LabVIEW program was developed based on the 3D-SMART program previously established by Prof. Kevin Welsher's lab. The primary modifications include configuring the camera parameters and synchronizing the EMCCD camera with the tracking system. The workflow of the program is illustrated in **Supplementary Figure. 4**.

In the caption of Fig 2a, the following explanation has been added:

“Spec. cent.: Spectral centroid.”.

In the Supplementary Note 2, the following sentences have been revised:

“Learnable positional embeddings are then incorporated into the patch sequence to retain spatial information, forming the input to the transformer encoder. Following the encoding process, the output is passed through a single-layer multilayer perceptron (MLP), which generates the spectral centroid—a single numerical value representing the center of the spectral distribution for the analyzed fluorescence spectrum. This workflow ensures precise spectral feature extraction and robust centroid detection from spectral images.”.

3. Another major problem I see throughout is that the authors appear to conflate making a precise measurement with making an accurate measurement. They should compare the spectra they measure to published spectra for these emitters, to illustrate that their measured spectra are accurate.

Response: Thank you for your comments and kind suggestion.

We have evaluated the system’s precision under varying fluorescence intensities and exposure times. The precision of fluorescence spectral localization was assessed by calculating the standard deviation of the localized spectral centroid position, which quantifies the system’s ability to resolve spectral shifts in continuous spectral measurements.

Spectral measurement accuracy refers to the degree to which the measured spectrum aligns with the ground truth. To evaluate the accuracy of spectral detection, we compared the spectra acquired using the 3D-SpecDIM system with those obtained from a commercial spectrofluorometer (FS5, Edinburgh Instruments). For the commercial spectrometer measurements, emission spectra were recorded for four different fluorophores: four-color fluorescent

beads (Thermo Fisher Scientific, TetraSpeck™, T7279), Atto 565 dyes, yellow fluorescent beads (FSSY002, Bangs Laboratories), and red fluorescent beads (FSFR002, Bangs Lab). As shown in Supplementary Figure 25d, the spectral data obtained with the 3D-SpecDIM system (circles) closely align with those measured by the commercial spectrometer (solid lines), demonstrating strong agreement. This consistency confirms that the 3D-SpecDIM system is able to accurately measure fluorescence spectra while providing the added advantage of high spatiotemporal resolution for single-molecule dynamics studies.

To quantitatively assess the accuracy, we calculated the root mean square error (RMSE) between the spectral data from the two systems, yielding a value of 0.23. This low RMSE further validates the precision of the 3D-SpecDIM system in spectral measurements.

Revision: In Page 6 Line 21, the following sentences have been revised:

“The precision of fluorescence spectral localization was assessed through the standard deviation of the localized spectral centroid position, providing a measure of the system’s capability to distinguish spectral shifts in a continuous spectral measurement. The spectral localization precision was evaluated across different photon count conditions using the normal fitting method, the CNN model, and the ViT_d model. Among these approaches, the ViT_d model demonstrated superior performance, achieving a precision of 0.3 nm with 30,000 photon counts for this sample (Fig. 2d).”

In the Supplementary Information file, **Supplementary Figure 25d** has been added:

Supplementary Figure 25. Spectral registration. (d) Comparison of spectra acquired with 3D-SpecDIM and a commercial spectrofluorometer. Four different fluorophores were analyzed: four-color fluorescent bead (Thermo Fisher Scientific, TetraSpeckTM, T7279, blue line and circle), Atto 565 dye (yellow line and circle), yellow fluorescent bead (FSSY002, Bangs Lab, green line and circle), and red fluorescent bead (FSFR002, Bangs Lab, red line and circle). The root mean square error (RMSE) between the spectral data from the two systems was calculated to be 0.23.

4. I also suggest that the authors make available kymograms showing the full spectral timecourse, rather than just plotting the track in a color scale representing the (again, centroid?) that was estimated by the neural net, and showing a single snapshot of one (representative?) spectrum.

Response: Thank you for your great suggestion. We have now included the kymograms of the spectral data in the revised manuscript.

Revision: In the Supplementary Information file, the **Supplementary Figure 31** has been added:

Supplementary Figure 31. Kymograms for the relevant trajectories in main text. The top panel corresponds to the kymograms of the spectral images, in which the the upper bright line shows the reference position signal while the lower wider line shows the spectral signal. The bottom panel in (a), (b), and (d) show the spectral centroid position as a function of time. The bottom panel in (c) is the same with Fig. 2f. (a) Kymogram for trajectory Fig. 2a-c. (b) Kymogram for trajectory Fig. 2g-i. (c) Kymogram for trajectory Fig. 3b-f. (d) Kymogram for trajectory Fig. 4b-e. Only successfully fitted spectral data points are presented. At some time points, the signal-to-noise ratio was too low to obtain a valid spectral centroid estimation.

5. Yet another major issue for me is the set of biological applications that the authors have selected. Both mitochondria and cell blebs are microscale, not

nanoscale, objects. No nanoscale position information is relevant in either case, and they move incredibly slowly. Therefore, I fundamentally do not see why 3D-SMART is needed to track these objects – why not just image them in two color channels, perhaps with some sort of autofocus? I do not think that these example applications adequately demonstrate the power of having spectral information available when tracking a nanoparticle.

Response: Thank you for your comment.

The size of healthy mitochondria is typically on the microscale. However, mitochondria undergoing mitophagy often exhibit a reduced size, generally in the hundreds of nanometers range [Nature 554, 382–386 (2018)]. To further analyze the size of mitochondria, we performed a detailed assessment using the Mitochondria Analyzer plugin in ImageJ. As shown in **Supplementary Figure 14a-b**, the distribution of form factor values for mitochondria highlights their morphological variability. The form factor is a parameter that reflects mitochondrial shape, where values close to 1 indicate a nearly circular morphology. Most fragmented mitochondria exhibit a perimeter of approximately 1.9 μm . Furthermore, we calculated the longest shortest path of mitochondria, which serves as an approximation of their diameter, yielding an average value of 250 ± 240 nm. Additionally, we reconstructed the 2D image of the tracked mitochondrion in Fig. 3 by mapping fluorescence photons to the laser focus positions at their corresponding time points (**Supplementary Figure 14c**). The full width at half maximum (FWHM) of this mitochondrion was measured to be 0.4 ± 0.07 μm (**Supplementary Figure 14d**). Collectively, these results demonstrate that 3D-SpecDIM is well-suited for studying the mitophagy process.

In the cell blebbing experiments, 100 nm AgNPs were tracked. The AgNP was utilized to enhance the excitation light field and initiate the cellular blebbing.

During the blebbing process, only the polarity of the membrane region surrounding the tracked nanoparticle was monitored.

There are several advantages using 3D target-locking tracking:

- 1. Improves the temporal resolution of spectral detection.** In traditional grating-based spectral imaging, capturing mitophagy or cellular blebbing events requires imaging the entire cell. Acquiring a 2D spectral scanning image typically takes tens of seconds to several minutes (**Supplementary Table 6**), which may be insufficient to study these events. In contrast, 3D-SpecDIM only focuses on the target object, greatly reduced the required imaging area. Compared to conventional grating-based spectral imaging methods, 3D-SpecDIM achieves several orders of magnitude improvement in the temporal resolution of spectral detection (**Supplementary Table 6**), enabling more precise measurements of the spectral changing dynamics.
- 2. Extends the data acquisition efficiency.** With target-locking tracking, 3D-SpecDIM can maintain the observing object within the focal plane throughout the imaging process, enabling long-term continuous data acquisition. Traditional spectral imaging methods may struggle to capture sufficient data, as the target may move out of the focal plane during imaging process.
- 3. Provides additional dimensional information.** The 3D positional information and the moving speed information can be acquired simultaneously with the spectral imaging. For example, in the cellular blebbing experiment, the AgNP exhibits greater displacement and velocity in the axial direction compared to the lateral direction, suggesting that it was elevated by the blebbing. Additionally, the AgNP's displacement correlates with fluorescence spectral centroid shifts (**Supplementary Figure 24f** further highlighting the dynamic nature of the process).

Compared to the two-color channel detection method (referred to as the "wavelength-split detection method" in the manuscript), the full spectral acquisition capability of 3D-SpecDIM enables the application of spectral unmixing techniques. Conventional ratiometric fluorescence imaging methods utilize two different band-pass filters to collect fluorescence signals across distinct wavelength ranges. The ratio between these two channels is then used for spectral characterization. However, due to emission spectrum crosstalk, the accuracy and sensitivity of these methods are compromised. In contrast, 3D-SpecDIM acquires the full fluorescence spectral profile, enabling spectral unmixing and thereby providing more accurate and sensitive measurements.

To further illustrate this point, we conducted a comparative analysis using the data from **Fig. 3**. In the dual-channel ratiometric detection method, pixel intensities on the EMCCD image were summed across two distinct wavelength ranges and then divided to calculate the intensity ratio. Compared to this conventional approach, 3D-SpecDIM-enabled spectral unmixing offers enhanced accuracy (**Supplementary Figure 13d** **Supplementary Figure 18**). Specifically, the pH in lysosomes determined through spectral unmixing with 3D-SpecDIM is approximately 4.5, whereas the conventional dual-channel ratiometric detection method yields a value of 6 (**Supplementary Figure 18**).

Additionally, by capturing full spectral profiles, 3D-SpecDIM enables the simultaneous detection of multiple fluorophores using a single spectral imaging camera. For example, in the mitophagy experiment (**Figures 3c, e**), the spectral signatures of three fluorophores were acquired concurrently.

Revision: in the Supplementary Information file, **Supplementary Figure 14**, **Supplementary Table 6**, and **Supplementary Figure 18** have been added:

Supplementary Figure 14. Mitochondrial size analysis during mitophagy.

(a) The heatmap of perimeters of mitophagy versus form factor, where form factor indicates the mitochondrial shape (the values close to 1 indicate a nearly circular morphology). **(b)** The histogram distribution of mitochondrial longest shortest path. Only the form factor lower than 1.5 were analyzed. The longest shortest path approximates the diameter of mitochondrial. **(c)** The 2D image of the tracked mitochondrion in Fig. 3 by mapping fluorescence photons to the laser focus positions at their corresponding time points. Bin time = 1 s. **(d)** The full width at half maximum (FWHM) of the mitochondrial image in (c) as a function of time during tracking.

Supplementary Table 6. Spectral detection performance comparison between 3D-SpecDIM and Zeiss LSM 980 spectral scanning confocal microscope.

	Spectral scanning step size/precision	Imaging size	Time resolution	Measuring dimensions
3D-SpecDIM	~0.5 nm	-	~0.1 s	3D
Zeiss-Commercial Microscope LSM980	3 nm (from 568 nm to 740 nm)	16.1×16.1 μm^2 (64×64 pix)	117 s	2D
		84.9×84.9 μm^2 (1024×1024 pix)	634 s	2D
	5 nm (from 568 nm to 740 nm)	16.1×16.1 μm^2 (64×64 pix)	46 s	2D
		84.9×84.9 μm^2 (1024×1024 pix)	202 s	2D

Supplementary Figure 18. Comparison of the 3D-SpecDIM-enabled spectral unmixing ratiometric fluorescence detection and the conventional dual-channel ratiometric detection. (a) The intensity ratio comparison of dual-channel ratiometric detection (yellow line) and 3D-SpecDIM-enabled spectral unmixing detection (blue line). (b) Similar comparison as in (a), with the corresponding mGold/JF549 intensity ratio converted to a calibrated pH. The pH range is scaled to start at 6.5.

6. Note that for the blebbing application, 100nm silver nanoparticles are also tracked on the membrane – but it is not clear why the movement of these objects is relevant. The changes in their position and speed, and changes in pH, seem simply consistent with being present on a membrane bleb.

Response: Thank you for your comment.

In the cellular blebbing experiment, AgNPs served three important roles. First, since the occurrence of cellular blebbing is dependent on excitation laser intensity, AgNPs were utilized to enhance local light density near their surfaces through plasmonic enhancement of the electromagnetic field, thereby facilitating cellular blebbing at sites where AgNPs were present.

Second, the AgNPs served as a tracking probe, directing the imaging area to continuously focus on the blebbing region. During the imaging process, only the polarity of the membrane region surrounding the tracked AgNP was monitored with high temporal and spectral precision.

Third, the 3D positional dynamics of AgNPs also provide additional insights, as previously discussed. In the cellular blebbing experiment, AgNPs exhibited greater displacement and velocity in the axial direction compared to the lateral direction, suggesting that they were elevated by the blebbing process. Furthermore, AgNP displacement correlated with fluorescence spectral

centroid shifts (**Supplementary Figure 24f**), further emphasizing the dynamic nature of the process.

We agree that this point requires further clarification. We have revised the relevant description in the manuscript accordingly.

Revision: In Page 11, Line 2, the following sentences have been revised:

“We observed that cellular blebbing can be induced by Nile Red staining combined with high-intensity blue light illumination (**Supplementary Table 2**). Imaging the rapidly changing dynamics of cellular blebbing is challenging with conventional grating-based spectral scanning microscopes due to their limited temporal resolution and spectral localization precision (**Supplementary Table 6**). In contrast, 3D-SpecDIM offers high temporal resolution for spectral imaging, enhanced spectral localization precision, and simultaneous real-time 3D positional tracking, making it a powerful tool for studying fast dynamic processes..”

In Page 11, Line 15, the following sentences have been revised:

“The plasmonic enhancement of the electromagnetic field near the surface of AgNPs leads to an increase in local light density⁵⁸, which is a critical factor in initiating of cellular blebbing (**Supplementary Table 2**). As a result, cellular blebbing frequently initiates at sites where these nanoparticles are located (**Supplementary Movie 3**)..”

In Page 11, Line 27, the following sentences have been revised:

“This observation suggests that the AgNP was elevated by the expanding bleb, a dynamic process that would be difficult to capture in 2D spectral imaging using conventional spectral scanning microscopy.”

MINOR COMMENTS: This is an incomplete list, but reflects the general attention to detail that will be necessary to improve the manuscript.

1. The method is touted as being able to track single fluorophores, but only one track of a single SeTau 647 (in 90% glycerol) is shown. More examples should be given in the SI to show that this is not a one-off measurement. And, further analysis is needed to illustrate that the spectra are correct.

Response: Thank you for your insightful comment.

a) **Single molecule demonstration.** We fully agree that additional data are necessary to validate the single-molecule spectral tracking capability. To address this, we conducted experiments tracking three distinct fluorescent dyes in a 90% glycerol solution, including Atto 665, SeTau 647, and Atto 565, with approximately 30 trajectories recorded for each dye. We conducted statistical analysis of these trajectories and their corresponding spectral data (**Fig. 2j–m, Supplementary Figure 10**). The hydrodynamic diameter of the tracked molecules, calculated using the Einstein-Stokes equation, was 1.28 ± 0.55 nm, with an average tracking duration of 5.47 ± 4.51 seconds. Notably, the measured hydrated diameter is consistent with previously reported single-molecule measurements [Nat Commun 11, 3607 (2020)], further confirming the single-molecule tracking capability of 3D-SpecDIM.

Furthermore, the single-molecule tracking capability was further validated through the observation of photoblinking. In this experiment, BSA protein was labeled with two Atto 565 fluorescent dyes and tracked using 3D-

SpecDIM, where photoblinking of the dyes was observed (**Supplementary Figure 11**). The fluorescence intensity alternated between single-dye and two-dye intensity levels. To minimize photobleaching, the excitation laser power was slightly reduced in the experiment.

These results collectively validate the single-molecule spectral tracking capability of 3D-SpecDIM.

b) **Spectral accuracy evaluation.** Spectral measurement accuracy refers to the degree to which the measured spectrum aligns with the ground truth. To evaluate the accuracy of spectral detection, we compared the spectra acquired using the 3D-SpecDIM system with those obtained from a commercial spectrofluorometer (FS5, Edinburgh Instruments). For the commercial spectrometer measurements, emission spectra were recorded for four different fluorophores: four-color fluorescent beads (Thermo Fisher Scientific, TetraSpeckTM, T7279), Atto 565 dyes, yellow fluorescent beads (FSSY002, Bangs Laboratories), and red fluorescent beads (FSFR002, Bangs Lab). As shown in **Supplementary Figure 25d**, the spectral data obtained with the 3D-SpecDIM system (circles) closely align with those measured by the commercial spectrometer (solid lines), demonstrating strong agreement. This consistency confirms that the 3D-SpecDIM system is able to accurately measure fluorescence spectra while providing the added advantage of high spatiotemporal resolution for single-molecule dynamics studies.

To quantitatively assess the accuracy, we calculated the root mean square error (RMSE) between the spectral data from the two systems, yielding a value of 0.23. This low RMSE further validates the precision of the 3D-SpecDIM system in spectral measurements.

Revision: In page 7 Line 28, the following sentences were added to the revised manuscript:

“To further demonstrate the single-molecule spectral tracking capability of 3D-SpecDIM, we conducted experiments tracking three distinct fluorescent dyes in a 90% glycerol solution, including Atto 665, SeTau 647, and Atto 565, with approximately 30 trajectories recorded for each dye (**Fig. 2j-m, Supplementary Fig. 10**). The hydrodynamic diameter of the tracked molecules, calculated using the Einstein-Stokes equation, was 1.28 ± 0.55 nm (**Fig. 2**), with a tracking duration of 5.47 ± 4.51 seconds (**Fig. 2l**). The spectral centroids (**Fig. 2k**) and fluorescence intensities (**Fig. 2m**) were also statistically analyzed. Notably, the measured hydrated diameter aligns with previously reported single-molecule measurements [Nat Commun 11, 3607 (2020)], , further confirming the single-molecule tracking capability of 3D-SpecDIM.

Furthermore, the single-molecule tracking capability was further validated through the observation of photoblinking. In this experiment, BSA protein labeled with two Atto 565 fluorescent dyes was tracked using 3D-SpecDIM, during which photoblinking of the dyes was observed (**Supplementary Fig. 11**). The fluorescence intensity alternated between single-dye and two-dye intensity levels. To minimize photobleaching, the excitation laser power was slightly reduced in the experiment. These results collectively validate the single-molecule spectral tracking capability of 3D-SpecDIM”.

Fig 2j-m has been added to Fig. 2:

(j-m) Statistics of the hydrodynamic diameter (j), spectral centroid (k), tracking durations (l), and emission rates (m) of three different fluorophores: Atto 665 (N=33), SeTau 647 (N=35), and Atto 565 (N=26). The mean and standard deviation values are labeled on the histogram figure.

In the Supplementary Information file, Supplementary Figure 10 and Supplementary Figure 11 have been added:

Supplementary Figure 10. Single molecule spectral tracking of three distinct molecules. (a) Left: 3D moving trajectory of a single Atto 665 fluorescent molecule diffusing in 90 wt% glycerol. Right: spectral centroid (up panel) and intensity (bottom panel) as a function of time. (b) Left: 3D moving trajectory of a single Atto 565 fluorescent molecule diffusing in 90 wt% glycerol. Right: spectral centroid (up panel) and intensity (bottom panel) as a function of time. (c) Left: 3D moving trajectory of a single SeTau 647 fluorescent molecule diffusing in 90 wt% glycerol. Right: spectral centroid (up panel) and intensity (bottom panel) as a function of time. (d) Mean square displacement (MSD) of Atto 665 (top), Atto 565 (middle), and Setau 647 (bottom) as a function of lag time for trajectory (a-c).

Supplementary Figure 11. Single molecule demonstration with photoblinking. **(a)** 3D trajectory of a two-Atto 565-labeled BSA protein diffusing in a 90 wt.% glycerol solution, with the trajectory color-coded to represent time. **(b)** Fluorescence intensity as a function of time for the trajectory shown in (a), demonstrating alternation between single-dye and two-dye intensity levels. **(c)** Mean square displacement as a function of lag time for the trajectory in (a). **(d)** Intensity distribution histogram for the trajectory shown in (a). The histogram displays two distinct peaks: the higher peak corresponds to both dyes being in the emissive state, while the lower peak reflects one dye transitioning to a dark state

In the Supplementary Information file, **Supplementary Figure 25d** has been added:

Supplementary Figure 25. Spectral registration. (d) Comparison of spectra acquired with 3D-SpecDIM and a commercial spectrofluorometer. Four different fluorophores were analyzed: four-color fluorescent bead (Thermo Fisher Scientific, TetraSpeckTM, T7279, blue line and circle), Atto 565 dye (yellow line and circle), yellow fluorescent bead (FSSY002, Bangs Lab, green line and circle), and red fluorescent bead (FSFR002, Bangs Lab, red line and circle). The root mean square error (RMSE) between the spectral data from the two systems was calculated to be 0.23.

2. *How does the movement of the particle in the tracking spot influence the spectrum? Is motion blur accounted for?*

Response: Thank you for your insightful comment.

Motion blur is indeed a critical factor that can influence spectral measurements, particularly when the particle undergoes substantial movement within the excitation volume during the EMCCD exposure time.

In 3D-SpecDIM, the target is locked to the excitation volume center via active feedback tracking. The role of the stage here is continuously repositioning the moving target to the center of excitation volume. In this way, the target keeps relatively stationary to the excitation volume and reduce the motion blur of the spectral image on EMCCD. The 3D positional tracking precision determines how well the target is locked to the excitation volume, which can affect the spectral localization precision. We have conducted an experiment to evaluate influence of the axial tracking precision on spectral localization precision (**Supplementary Figure 1a**). The result shows that the spectral localization precision decreases with the axial tracking precision decrease. However, compared with spectral imaging without 3D target-locking tracking, the blur caused by the movement of the particle during the spectral acquisition time can be effectively reduced (**Supplementary Figure 1b**).

We also evaluated the defocusing caused spectral localization precision deterioration (**Supplementary Figure 1c**), further highlight the advantage of 3D target-locking tracking on improve spectral localization precision deterioration.

Revision: Supplementary Figure 1 has been added to the Supplementary Information file:

Supplementary Figure 1. The impact of defocusing on spectral precision. (a) Z-axis localization precision as a function of spectral localization precision. Fixed fluorescent beads were tracked while systematically adjusting the feedback control parameters of the axial piezo stage to modify localization precision. Localization precisions were calculated with 1 second window. **(b)** Upper panel: kymograms of spectral images of a freely diffusing fluorescent bead in an aqueous solution under conditions without target-locking tracking. Lower panel: the spectral centroid as a function of frame number, with only successfully fitted data points shown. At certain time points, the signal-to-noise ratio was too low to obtain a valid spectral centroid estimation. **(c)** Defocusing distance as a function of spectral localization precision. A fixed fluorescent particle was imaged at various axial positions. As the Z defocusing distance increased from 0 to $\pm 1.2 \mu\text{m}$, the spectral localization precision decreased from 0.6 nm to approximately 2.6 nm at a Z defocusing distance of $-1.2 \mu\text{m}$ and 2.1 nm at $1.2 \mu\text{m}$. **(d)** Impact of motion blur on subpixel positional precision of

EMCCD image with target-locking tracking. Data represents the trajectories shown in Fig. 2a-c. Errors were calculated by subtracting the average x- and y-pixel localization values across all frames from the x- and y-localization values in each individual EMCCD frame.

3. Need details of how the prism's nonlinear dispersive characteristics were calibrated / spectrum was calibrated. A mapping function between pixels and wavelength should be shown in the SI. Is the EMCCD detection efficiency accounted for?

-Validation of spectral measurements by comparison with published spectra for different emitters should be shown in the SI.

Response: Thank you for your comment. In the revised Supplementary Information, we have included a detailed description of the spectral calibration method. Specifically, we utilized several narrowband filters and lasers to calibrate the spectral mapping between wavelength and pixel position. A second-order polynomial interpolation was used to establish the pixel-to-wavelength calibration curve, which is presented in **Supplementary Figure 25c**. We didn't deliberately account for the EMCCD detection efficiency in the spectral calibration procedure. However, when using the fluorescence of fluorescent beads to calibrate the spectral position, the EMCCD detection efficiency was inherently considered to some extent.

As shown in **Supplementary Figure 25d**, a comparison between spectra measured using 3D-SpecDIM and a commercial spectrofluorometer (FS5, Edinburgh Instruments) demonstrates good agreement, further validating the calibration approach. For the commercial spectrometer measurements, emission spectra were recorded for four different fluorophores: four-color fluorescent beads (Thermo Fisher Scientific, TetraSpeck™, T7279), Atto 565 dyes, yellow fluorescent beads (FSSY002, Bangs Laboratories), and red

fluorescent beads (FSFR002, Bangs Lab). As shown in Supplementary Figure 25d, the spectral data obtained with the 3D-SpecDIM system (circles) closely align with those measured by the commercial spectrometer (solid lines), demonstrating strong agreement. This consistency confirms that the 3D-SpecDIM system is able to accurately measure fluorescence spectra while providing the added advantage of high spatiotemporal resolution for single-molecule dynamics studies.

Revision: in Supplementary Information file, a **Supplementary Notes 3** has been added, in which the pixel-wavelength relationship calibration procedure was provided:

Supplementary Notes 3. Workflow and details of 3D-SpecDIM tracking and spectral data acquisition

1. Pixel-wavelength relationship calibration

Fixed four-color fluorescent beads (Thermo Fisher Scientific, TetraSpeck™, T7279) were placed on the microscope, and the 3D-SpecDIM system was utilized to track a fluorescent bead immobilized on the coverslip, capturing its spectral information using the EMCCD. The resulting image was stitched into a 16×80 matrix, with the 16×16 region representing the non-dispersed spatial channel, where the bead center was located at pixel coordinates (8, 8), and the remaining 16×64 region corresponding to the dispersed spectral channel for intensity analysis along pixel coordinates (**Supplementary Fig. 25a**). Using different filters (514/30 nm, 577/20 nm, 591/6 nm) and measuring the spectral channel positions of lasers (490.2 nm, 561.7 nm, 636.8 nm), six paired datasets of pixel shifts and spectral wavelengths were obtained (**Supplementary Fig 16b**). A quadratic polynomial fitting was then applied to derive the calibration

function for converting pixel shifts into spectral wavelengths (**Supplementary Fig 16c**), allowing accurate conversion of intensity variations along pixel shifts into the true wavelength distribution.

To demonstrate the accuracy of spectral detection, we compared the spectra acquired using the 3D-SpecDIM system with those obtained from a commercial spectrofluorometer (FS5, Edinburgh Instruments). For the commercial spectrometer measurements, emission spectra were recorded for four different fluorophores: four-color fluorescent beads (Thermo Fisher Scientific, TetraSpeckTM, T7279), Atto 565 dyes, yellow fluorescent beads (FSSY002, Bangs Laboratories), and red fluorescent beads (FSFR002, Bangs Lab). As shown in **Supplementary Figure 16d**, the spectral data obtained with the 3D-SpecDIM system (circles) closely align with those measured by the commercial spectrometer (solid lines), demonstrating strong agreement. This consistency confirms that the 3D-SpecDIM system is able to accurately measure fluorescence spectra while providing the added advantage of high spatiotemporal resolution for single-molecule dynamics studies.

In the Supplementary Information file, **Supplementary Figure 25** has been revised, in which **Supplementary Figure 25c** shows the pixel-to-wavelength calibration curve and **Supplementary Figure 25d** shows comparison result of spectra acquired with 3D-SpecDIM and a commercial spectrofluorometer:

Supplementary Figure 25. Spectral registration. (a) Top panel: the reference image (left) and spectral image (right) of four-colors fluorescent bead (Thermo Fisher Scientific, TetraSpeck™, T7279) on EMCCD. The spectral image was generated by horizontally flipping the raw spectral image. Bottom panel: the fluorescence intensity distribution versus pixel shift for spectral image. (b) Spectral calibration with different narrow bandpass filters and lasers. (c) Convert the pixel shift between the reference image and the spectral image into spectral wavelengths using quadratic polynomial fitting. (d) Comparison of spectra acquired with 3D-SpecDIM and a commercial spectrofluorometer. Four different fluorophores were analyzed: four-color fluorescent bead (Thermo Fisher Scientific, TetraSpeck™, T7279, blue line and circle), Atto 565 dye (yellow line and circle), yellow fluorescent bead (FSSY002, Bangs Lab, green

line and circle), and red fluorescent bead (FSFR002, Bangs Lab, red line and circle). The root mean square error (RMSE) between the spectral data from the two systems was calculated to be 0.23.

4. In the main paper, no references or explanation are given for the vision transformer / domain adaptation method used to fit spectra. (A couple of references are given in the SI, however.) The comparison shown in Fig. 2c does not convincingly show that the “ViT” method is more accurate, simply that it produces a less noisy result. This analytical choice, and the accuracy (rather than precision) of the method, need to be more clearly and quantitatively justified.

Response: Thank you for your comment. The choice of the Vision Transformer (ViT) model was motivated by its incorporation of positional encoding, which enhances its capacity to capture spatial relationships between pixels in the spectral image. This capability is especially advantageous for spectral identification tasks, as the positional information among pixels is crucial for accurately determining the spectral centroid.

We have conducted a comparative analysis of the performance of the ViT model with domain adaptation strategy and convolutional neural networks (CNNs) in the revised manuscript. The results demonstrate that ViT outperforms CNNs in spectral localization precision, particularly under high signal-to-noise ratio conditions (**Fig. 2d**).

To assess the accuracy of the ViT_d model, we compared the spectra of green fluorescent beads acquired using 3D-SpecDIM with those obtained from a commercial spectrofluorometer (Lumina, Thermo Scientific). The spectral centroid was measured at 519.4 nm by the spectrofluorometer, while the ViT_d model and traditional Gaussian fitting methods reported values of 517.87 nm and 518.22 nm, respectively (**Supplementary Figure 26**). Although both

methods exhibit similar measurement accuracy, the ViT_d model achieved a 32% improvement in spectral localization precision, reducing the error from 1.63 nm to 1.11 nm.

We agree that this point requires further clarification. We have revised the relevant description in the manuscript accordingly.

Revision: In Page 6 line 4, the following sentences have been added:

“To further improve the spectral localization precision, we developed an optimized spectral detection method based on the Vision Transformer (ViT) model with a domain adaptation strategy (ViT_d)^{48, 49}. Unlike conventional fitting methods, which rely on curve fitting for spectral centroid localization, the ViT_d model leverages positional encoding to capture spatial relationships within spectral images, thereby facilitating precise spectral centroid identification. Compared to the conventional normal Gaussian fitting method, the ViT_d-based approach improved spectral localization precision from 1.63 nm to 1.11 nm, representing a 32% enhancement (**Fig. 2c**, **Supplementary Fig. 5** and **Supplementary Fig. 6**). Additionally, the performance of the ViT_d model was compared with that of a convolutional neural network (CNN) in terms of spectral localization precision (**Fig. 2d**). The ViT_d model shows improved performance over both the CNN and the normal fitting method. It should be noted that the ViT_d-based spectral detection method is more effective in high photon counts situations than in low photon counts conditions due to the influence of the signal-to-noise ratio. This improvement highlights the advantage of the positional encoding capability of ViT_d in spectral feature recognition.”

In the Supplementary Information file, the caption of **Supplementary Figure 6d** has been revised:

“Supplementary Figure 6. Comparison of centroid recognized methods. (d) Comparison of spectral localization precision across different methods using fluorescent bead spectra data. The Vision Transformer with domain adaption strategy shows improved spectral precisions.

In the section of References, the following references have been added:

48. Alexey Dosovitskiy et al. An Image is Worth 16x16 Words: Transformers for Image Recognition at Scale. *9th International Conference on Learning Representations (ICLR)* (2021).
49. Zhang, Y., Liu, T., Long, M. & Jordan, M.I. Bridging Theory and Algorithm for Domain Adaptation. *Proceedings of the 36th International Conference on Machine Learning (ICML)* (2019).

In the Supplementary Information file, **Supplementary Figure 26** has been added:

Supplementary Figure 26. Comparison of green fluorescent dead spectra acquired using 3D-SpecDIM (circle) and a commercial spectrometer (solid line).

5. I am unclear as to what is meant by “normal fitting method” for spectra. Does this refer to fitting spectra as a sum of “normal distributions”, which is indeed a common method?

Response: Thank you for your comment.

The "normal fitting method" refers to fitting spectral curves with a Gaussian distribution, where the peak of the fitted Gaussian is defined as the spectral centroid. This approach is widely utilized in spectral analysis.

For instance, in the Supplementary Figure 2 of the reference (Nat Commun 7, 13544, 2016, <https://doi.org/10.1038/ncomms13544>), Gaussian fitting was applied to identify emission peaks in fluorescence spectra, as described: “Supplementary Figure 2. Spectrum fitting. Example of a peak extracted (black line) from the spectral part of the image (Tetraspeck™ beads), which was fit to a Gaussian distribution (red line) to determine the centre position”.

Supplementary Figure 2. Spectrum fitting. Example of a peak extracted (black line) from the spectral part of the image (Tetraspeck™ beads), which was fit to a Gaussian distribution (red line) to determine the centre position.

6. *Figure 1 seems unnecessarily large and detailed given that this system is basically 3D-SMART with a prism / camera added to the detection path. I suggest re-focusing on panels d) and e), which are unique to this work. In particular, the detail in panel e) is the main point of the paper, and therefore should be shown more prominently.*

Response: Thank you for your kind suggestion. We have revised Figure 1 in accordance with your feedback.

Revision: Fig. 1 has been revised:

Fig. 1. Schematic of 3D-SpecDIM. (a) The focused laser spot is rapidly scanned within a small volume after the objective lens. The molecule's deviation from the center of the illumination volume is estimated in real time using an FPGA, which processes photon arrival time information. Based on this deviation, a feedback control voltage is applied to the piezo stage to reposition the molecule at the center of the illumination volume. This process continuously locks the target molecule within the small excitation volume, enabling high-spatiotemporal-precision 3D single-molecule dynamics observation. (b) To simultaneously acquire the single-molecule fluorescence spectrum, a prism-based spectral imaging system was integrated into the detection path. The fluorescence was dispersed by a prism and projected onto an EMCCD to capture the spectral profile. The target-locking imaging strategy enables spectral acquisition with high imaging speed, high spectral precision, and large imaging depth. (c) The synchronization of 3D positional dynamics and spectral dynamics allows for multiparameter dynamic data acquisition.

7. Figure 2a repeats the data from Figure 1e; this should only be shown once.

Response: Thank you for pointing this out. We have addressed this issue by replacing the schematic in Figure 1e.

Revision: The trajectory in Fig. 1 has been revised accordingly.

8. *Fig. 3 – presentation of g) and h) needs to be re-thought. Very difficult to interpret.*

Response: Thank you for your comment.

In **Fig. 3g**, the spectrally encoded 3D trajectory of the mitochondrion is overlaid on the 3D volumetric image of the lysosome. The blue-to-yellow color-coded trajectory represents the 3D movement of the mitochondrion, with the color encoding the pH ratio. Meanwhile, the 3D image of the lysosome is color-coded from violet to red, indicating fluorescence intensity. The 3D lysosome volume image was reconstructed by registering the fluorescence photons to laser focus positions in a separated detection channel. The detailed reconstructed information can be found in **Supplementary Notes 4**. The checkerboard pattern represents transparency, while the alternating color blocks indicate different intensity levels. This visualization improves clarity and the intensity variations within the 3D image.

Fig. 3h shows the sum intensity projection images of **Fig. 3g** in xy plane (left), xz plane (middle), and yz plane (right).

We agree that **Fig. 3g-h** requires further clarification. We have revised the relevant description in the caption of **Fig. 3**.

Revision: In caption of Fig. 3, the following sentence has been revised and added:

“The blue-to-yellow color-coded trajectory represents the 3D movement of the mitochondrion, with the color encoding the pH ratio. Meanwhile, the 3D image of the lysosome is color-coded from violet to red, indicating fluorescence intensity. The 3D lysosome volume image was reconstructed by registering the fluorescence photons to laser focus positions in a separated detection channel.

Detailed reconstruction information is provided in **Supplementary Notes 4**. The checkerboard pattern represents transparency, while the alternating color blocks indicate different intensity levels. **(h)** The sum intensity projection images of **(g)** in xy plane (left), xz plane (middle), and yz plane (right).”.

9. *What data are shown in 2c? Is this a “Spectral centroid”? Is it the peak of the spectrum?*

Response: Thank you for your question. The data presented in **Fig. 2c** represents the spectral centroid, defined as the wavelength corresponding to the maximum emission peak, as determined through Gaussian fitting or the ViT method. Compared to directly using the spectral peak value, the fitting approach helps reduce the influence of noise and provides a more robust estimation of the spectral position.

To clarify this, we have updated the caption of **Fig. 2c** to explicitly state that **Fig. 2c** displays spectral centroid data.

Revision: In **Fig. 2** of the main text, the following caption was added:

“Spec. cent.: Spectral centroid.”.

10. *SI Fig. 6: Why is the image of the bead oblong? Astigmatism in the system would presumably influence the measured spectra in a focus-dependent manner. Scale bars should be shown on the images. The images also appear to have position-dependent illumination; beads are dim at right and no beads are observed at far right.*

Response: Thank you for pointing out these issues. The elongation observed in the previous **Supplementary Figure 6** resulted from non-uniform stretching

along the x- and y-directions during figure formatting. We apologize for this oversight. We have corrected this issue in the revised figure.

We agree that the astigmatism could affect the spectral localization precision. However, we didn't notice obvious precision deterioration caused by the astigmatism during our experiment.

There is indeed non-uniform illumination in the previous **Supplementary Figure 6**. However, this wide-field illumination was not used for spectral tracking. Instead, during spectral tracking, the excitation light remains focused on the target molecule, ensuring that only the fluorescence spectral image of the target molecule is captured by the EMCCD.

To eliminate potential ambiguities, we have replaced the previous wide-field image with the spectral image acquired during 3D single-particle tracking.

Revision: The Supplementary Figure 25 has been revised:

Supplementary Figure 25. Spectral registration. (a) Top panel: the reference image (left) and spectral image (right) of four-colors fluorescent bead (Thermo Fisher Scientific, TetraSpeck™, T7279) on EMCCD. The spectral image was generated by horizontally flipping the raw spectral image. Bottom panel: the fluorescence intensity distribution versus pixel shift for spectral image. (b) Spectral calibration with different narrow bandpass filters and lasers. (c) Convert the pixel shift between the reference image and the spectral image into spectral wavelengths using quadratic polynomial fitting. (d) Comparison of spectra acquired with 3D-SpecDIM and a commercial spectrofluorometer. Four different fluorophores were analyzed: four-color fluorescent bead (Thermo Fisher Scientific, TetraSpeck™, T7279, blue line and circle), Atto 565 dye (yellow line and circle), yellow fluorescent bead (FSSY002, Bangs Lab, green

line and circle), and red fluorescent bead (FSFR002, Bangs Lab, red line and circle). The root mean square error (RMSE) between the spectral data from the two systems was calculated to be 0.23.

11. Throughout: None of the heat maps (spectra) have colorbars or color scaling indicated

Response: Thank you for pointing this out. We have added the color bar of the heat maps.

Revision: The Fig. 2 and its caption have been revised:

In page 25, Line 5, the following sentence has been added:

“All heat maps use the same color coding, representing normalized intensity.”.

12. Use of acronyms seems excessive; why is a new acronym needed for the technique when this is just 3D-SMART with spectra?

Response: Thank you for your insightful comment. While the proposed method is built upon the 3D-SMART framework, it introduces a key advancement: the ability to simultaneously capture 3D positional dynamics and fluorescence spectral dynamics.

To reflect this significant enhancement, we have adopted the acronym 3D-SpecDIM, emphasizing the integration of spectral imaging capabilities into the 3D-SMART framework. This designation not only highlights the novel aspect of **spectral dynamics imaging** but also helps distinguish the method from the original 3D-SMART technique. We believe this terminology will improve clarity in communicating the unique features of our approach to the broader scientific community.

13. Manuscript should be thoroughly edited for typos and grammar

Response: Thank you for your comment. We have thoroughly reviewed the manuscript and have corrected the typos and grammatical errors to improve the clarity and readability of the text.

REVIEWER COMMENTS

Reviewer #1 (Remarks to the Author):

The authors have addressed all of our concerns very well. The revised version includes new materials comparing different AI models (CNN vs transformer), effectively demonstrating the strength of their approach. However, although they mention that the code is published, the provided address (<https://github.com/houlab/3D-SpecDIM>) appears to be incorrect or unavailable. Could the authors please supply the correct address for accessing the SpecViT code? Overall, I recommend publication pending this minor correction.

Reviewer #1 (Remarks on code availability):

The code is not available for download.

Response: We sincerely appreciate your positive feedback and recommendation. We apologize for the oversight regarding the code availability. The code is now publicly accessible at the following GitHub repository: <https://github.com/houlab/3D-SpecDIM>.

Reviewer #2 (Remarks to the Author):

In the revised manuscript, Sha et al. have carefully and effectively addressed nearly all the comments and concerns raised in the previous review round. The authors have included additional experimental results to substantiate their system's capability for single-molecule analysis, which was one of the primary concerns noted in earlier feedback. They have also clarified and expanded upon sections that were previously unclear or logically difficult to follow. For example, they detailed how their 3D-SpecDIM operates to achieve tracking of particles in 3D and obtain ratiometric fluorescence imaging. I now can support the publication of this work in Nature Communications. However, there are still a few comments that the authors need to address before the manuscript can be fully accepted. These points are detailed below.

Response: We appreciate your support for the publication of our work.

1. In their response to the previous review round, the authors noted that "In single-molecule spectral tracking experiments, the EMCCD exposure time is set to 100 ms to ensure the collection of sufficient photons in each spectral image frame." While this detail can be inferred from Figure 2 and its caption, it should be explicitly stated in the main manuscript to prevent any potential misunderstandings about their single-molecule operation by future readers.

Response: Thank you for your suggestion. We have now added the statement regarding the EMCCD exposure time (100 ms) for single-molecule spectral tracking in the main text and the Methods section of the revised manuscript.

Revision: In page 7 Line 26 the following sentence has been added:

“To enhance the signal-to-background ratio, the EMCCD exposure time was set to 100 ms.”.

In page 15 Line 14, the following sentences have been added to Methods section:

“For single-molecule spectral tracking experiments, the EMCCD exposure time was set to 100 ms, and the excitation laser power at the objective was adjusted to 2 μ W to ensure the collection of sufficient photons per frame while minimizing photobleaching.”.

2. The authors are advised to specify the methodology used for the dual-channel ratiometric detection discussed in the new Supplementary Figure 18. Although they provided several paragraphs explaining how their new method is better than the conventional one in response to the 10th comment of Reviewer #2, there is no significant addition to the main text apart from Supplementary Figure 18 itself. The authors should offer more detailed guidance on interpreting their improved ratiometric characterization and clarify the rationale behind their method's enhanced accuracy. Specifically, they need to explain why the pH value of 4.5, extracted using their method, represents an improvement over the conventional method's estimated pH of 6.

Response: Thank you for pointing this out. We have added more detailed information on the comparison of dual-channel detection method and our method.

Firstly, the calibrated pH is derived from the ratio of the emission intensity between mGold and JF549. Our method can eliminate the ratio deviation induced by fluorescence spectral crosstalk, thereby provides a more accurate intensity ratio and calibrated pH value.

Secondly, the pH value of 4.5 derived from our method is more consistent with previously reported pH value in lysosome, which typically range between 4.5 and 5.0 (e.g., Mindell, J. A., Annu. Rev. Physiol. 2012). In contrast, the dual-channel detection method overestimates the pH. Our method overcomes these

limitations by leveraging full-spectrum acquisition and linear unmixing, resulting in more accurate quantification.

Revision: In page 10 Line 19, the following sentence has been added:

“3D-SpecDIM yielded a calibrated lysosomal pH of 4.5, consistent with previously reported values, whereas the wavelength-split detection method produced a pH of 6, showing a substantial deviation.”.

In page 16 Line 12, the following sentences have been added to the Methods section:

“While conventional wavelength-split detection methods are susceptible to spectral crosstalk, the full spectral profile acquisition capability of 3D-SpecDIM enables spectral unmixing, thereby enhancing the sensitivity and accuracy of ratiometric fluorescence imaging.”.

“The performance of 3D-SpecDIM with spectral unmixing was demonstrated through ratiometric fluorescence imaging of mGold and JF549 (Supplementary Figure 18). The calculated intensity ratio was converted to pH values using a calibration curve. Spectral unmixing yielded a calibrated lysosomal pH of 4.5, consistent with previously reported values, whereas the conventional wavelength-split detection method produced a pH of 6, indicating a large deviation. These results highlight the improved accuracy of 3D-SpecDIM with spectral unmixing compared to traditional wavelength-split detection.”.

3. In response to Comment 12 of Reviewer #2, the authors mentioned that their Nile red measurement in their cellular blebbing experiment "did not involve a single Nile Red molecule but rather an ensemble of molecules." However, this clarification is not explicitly stated in the revised manuscript. The authors should make this point clear.

Response: Thank you for your comment. We have now explicitly clarified this point in the revised manuscript.

Revision: In page 11 Line 18, the following sentence has been revised:

“To monitor this process, we employed 3D-SpecDIM to track the 3D positional dynamics of 100 nm silver nanoparticles (AgNPs) attached to the cell surface⁵⁵, while simultaneously capturing the fluorescence spectral dynamics of surrounding Nile Red molecules (**Supplementary Fig. 20**).”.

4. The caption of Supplementary Figure 20 states that "The solid lines represent Gaussian fits...", yet no such lines appear in the figure. The authors need to review this figure carefully and correct any discrepancies.

Response: Thank you for pointing this out. We apologize for our oversight. The caption of Supplementary Figure 20 has been corrected in the revised Supplementary Information file.

Revision: In Supplementary Figure 20, the following caption has been modified:

“**Supplementary Figure 20.** Histogram distribution of laser irradiation time for blebbing (blue, n = 15) and no blebbing (orange, n = 8) events. The mean irradiation times of $\mu = 539.74$ s ($\sigma = 486.65$ s) for blebbing events and $\mu = 489.01$ s ($\sigma = 449.72$ s) for no blebbing events.”.

Reviewer #3 (Remarks to the Author):

1. I appreciate the many careful revisions the authors have completed, especially the addition of methodological detail, the revisions to main paper figures and text, and the new SI figures to help the reader understand this interesting work. My major concern remains the issue of spectral accuracy in the data presented. It is very clear that the authors have achieved excellent precision in their chosen output parameter, the spectral centroid, but throughout the work I note many instances where it seems that the accuracy of the centroid – and by extension, the measured spectrum itself – is very much in question. These concerns appear as a repeated theme in the detailed notes and questions below; I strongly encourage the authors to work to address this issue prior to publication if the editorial team concurs.

Response: We greatly thank you for your supportive comments and your recognition of our revisions. We also thank you for highlighting the importance of spectral accuracy.

In this revised version, we have conducted additional analyses to quantify the accuracy of spectral centroid measurements obtained with 3D-SpecDIM.

- We confirmed that the peak emission wavelengths of Atto565, Setau647, and Atto665 measured in our single-molecule experiments are consistent with the emission spectra reported by the manufacturers, demonstrating the reliability of our spectral measurements.
- We systematically compared the spectral centroids of multiple fluorophores and fluorescent beads (including TetraSpeck™ beads, Atto565, and orange/red-emitting fluorescent beads) measured by 3D-SpecDIM and a commercial spectrofluorometer (FS5, Edinburgh

Instruments). To provide a clearer assessment of spectral accuracy, we now explicitly list the centroid values obtained from both systems, along with the absolute errors, which range from 0.38 ± 3.48 nm.

- We evaluated the influence of the EMCCD's wavelength-dependent detection efficiency on the measured spectra. Both simulations and experimental data indicate that while QE correction slightly improves accuracy under high photon count conditions, it introduces substantial errors under low photon count conditions.

Further details of these comparisons are provided in our point-by-point response below.

2. The other major concern may be easily fixed as I believe it to be due to an oversight: The authors have not, in fact, made the code available on their GitHub (this may be due to a misunderstanding of privacy settings on the repository, or similar). In any case, at the time of this review, it is not available at the cited link (<https://github.com/houlab/3D-SpecDIM>), nor is there any indication that it exists on the publicly available version of the houlab.github.io page. It is also not included in the .zip file for reviewers. While the improved description of the code and the comparisons to other methods are decent improvements from the previous version of the manuscript, I was not able to evaluate the code.

Response: Thank you for pointing this out. We apologize for our oversight. The code is now publicly accessible at the following GitHub repository: <https://github.com/houlab/3D-SpecDIM>.

3. Finally, I note that the other reviewers shared my concerns that the example applications do not fully highlight the future utility of this technique. I do appreciate the expanded single-molecule data and analysis; I think this has helped somewhat. I also appreciate the wording changes that more carefully

contextualize the present demonstrations relative to claims of single-molecule level applications that are not demonstrated here; in short, here I leave the determination of suitability to the editorial team.

Response: Thank you for your comment.

4. Detailed comments / questions: Most importantly, the characterization of the system's accuracy remains in my opinion inadequate. The new SI Fig. 25 does not inspire confidence that their calibrations correctly reproduce spectra; systematic issues are clearly evident there (see specific comments below). Moreover, it is imperative that their reported parameter of choice, the "spectral centroid" be accurate, and not just precise. It seems to me that the centroids reported in this work are not accurate even though they clearly achieve excellent precision. This can be easily checked by comparing to predicted and measured centroids of bulk spectra; again, I give some comparisons and examples in the specific comments below.

Response: Thank you for your comments and suggestions. To better quantify the spectral accuracy of our system, we have directly compared the spectral centroids measured by 3D-SpecDIM and those from a commercial spectrofluorometer in Supplementary Figure 25. Detailed comparison of centroid values is provided in Supplementary Table 7. The mean absolute error across all tested dyes and beads is 0.38 ± 3.48 nm, indicating good agreement between the two systems. It should be noted that the error reflects the combined accuracy of both 3D-SpecDIM and the commercial spectrofluorometer.

Furthermore, we would like to clarify why this work places greater emphasis on precision rather than accuracy. 3D-SpecDIM is primarily designed to capture spectral changing dynamics that arise from environmental or biological fluctuations. In this context, the precision of measuring relative spectral shift—rather than the absolute centroid position—provides the most meaningful

information. We therefore prioritize spectral precision to ensure reliable detection of temporal and spatial changes, which is central to the system’s value in biological sensing. Even so, we recognize the importance of absolute spectral accuracy and will continue to refine the optical calibration pipeline in future iterations.

Revision: In page 8, Line 7 the following sentences have been added:

“To assess spectral accuracy, we compared the spectral centroids measured by 3D-SpecDIM with those obtained from a commercial spectrofluorometer (Supplementary Figure 25 and Supplementary Table 7). It is worth noting that 3D-SpecDIM is primarily designed to capture spectral changing dynamics that arise from environmental or biological fluctuations. In this context, the precision of measuring relative spectral shift—rather than the absolute centroid position—provides the most meaningful information.”.

In supplementary file, the following table has been added:

Supplementary Table 7. Comparison of measured centroid values between 3D-SpecDIM and Commercial spectrometer.

	atto565	SY540	FR640	Tdeepred	TGreen	TOrange
3D-SpecDIM (nm)	593.0	604.8	685.3	668.8	508.6	578.0
Commercial Spectrometer (nm)	592.2	603.1	680.7	674.8	507.7	577.7
Error (nm)	+0.8	+1.7	+4.6	-6.0	+0.9	+0.3

5. Reviewer 1 asked whether targets moving out of focus could compromise the accuracy of spectral estimation: Based on the new data in SI Fig 1b, it is clear that the accuracy is indeed compromised; systematic drifts in the centroid by more than 15 nm are evident in the first third of the trace alone. While some of this drift might be attributable to X-Y movement within the detection region, most of it should be calibrated out, right? Or, if it is due to X-Y movement, then this calibration needs to be revisited to eliminate spatial dependence. Also, the authors' responses here really only discuss the "precision" with Z-defocus, and barely mention accuracy.

Response: Thank you for your comments.

Supplementary Figure 1b presents spectral imaging data of a freely diffusing fluorescent bead in aqueous solution, acquired without target-locked 3D tracking. In this case, the bead exhibited movement in both the X-Y plane and along the Z-axis.

We agree that positional shifts—especially in the absence of active feedback control—can compromise the accuracy of spectral centroid estimation. In Supplementary Figure 1b, the systematic drift observed at the beginning of the trace is primarily attributed to gradual lateral (X-Y) movement of the fluorescent bead. As illustrated by the spectra at time points t_1 and t_2 in the figure below, the positional imaging channel (reference image) shows similar spot sizes for t_1 and t_2 , but with clear displacement along the X-axis, confirming the presence of lateral drift.

This movement not only shifts the molecular image within the reference channel but also introduces asymmetry and misalignment in the spectral channel due to the system's point excitation configuration. When compounded by axial (Z-axis) defocusing, these effects further degrade image quality and amplify localization errors. In our system, the spectral centroid is calculated

based on the pixel shift between the reference and spectral channels. As misalignment between the two channels increases due to drift and defocus, the resulting pixel shift can be either overestimated or underestimated, thereby reducing the spectral accuracy, as shown in the figure below:

To further validate this effect, we conducted additional experiments using a fixed fluorescent bead positioned at the center of the excitation volume. After disabling the target-locked 3D tracking, we manually shifted the sample stage to laterally displace the bead along the X-axis from -600 nm to +600 nm, corresponding to approximately a 6-pixel shift. This lateral displacement resulted in a spectral centroid shift of approximately 5.5 nm (from 521.6 nm to 516.3 nm), confirming that X-Y movement indeed compromises spectral centroid accuracy.

Fortunately, both lateral and axial drifts can be effectively greatly mitigated by 3D-SpecDIM through target-locked 3D tracking. During active tracking, the target is maintained within ± 100 nm of the center of the excitation volume (Supplementary Fig. S1d), thereby ensuring high spectral accuracy. This capability highlights the strength of our system in preserving spectral fidelity under dynamic conditions.

Additionally, in response to your thoughtful suggestion, we conducted further experiments to assess the impact of Z-defocus on the accuracy of spectral centroid estimation. As shown in the revised Supplementary Figure S1c, we used fluorescent beads with a known emission peak at 520 nm (as reported by the manufacturer's datasheet, <https://bangslabs.com/technical-library/visible-dye-color-palette/#Flow%20Cytometry%20Fluorochromes>) as the ground truth reference. Our measurements indicate that within a defocus range of ± 1.2 μm , the spectral centroid remains stable when the particle is centered within the scanning region. The average deviation from the reported 520 nm peak was 0.82 ± 0.64 nm.

Revision: In supplementary file, Supplementary Figure 1 has been modified:

(c) Spectral centroids as a function of defocusing distance. A fixed fluorescent particle was imaged at various axial positions to assess the effect of Z-defocus. As the defocus distance increased from 0 to $\pm 1.2 \mu\text{m}$, the spectral localization precision declined from 0.6 nm to approximately 2.6 nm. The red dashed line represents the ground truth emission wavelength, as reported in the manufacturer's datasheet. The average deviation between the measured spectral centroids and the reported peak was $+0.82 \pm 0.64 \text{ nm}$.

In Supplementary Note 3, the following sentence was added:

“It should be noted that although careful calibration, the accuracy of spectral centroid measurements can be affected by multiple factors, including optical path alignment, calibration parameters, and the specific filter sets used.”.

6. Notably, the new SI Fig. 1 also raises some new questions about the analysis software; it is surprising that in much of 1b “the signal-to-noise ratio was too low to obtain a valid spectral centroid estimation”... yet this is for a bright bead; surely a single-molecule signal would have far worse SNR. Moreover, the SNR looks fine by eye and a centroid could be computed analytically (without a NN).

Response: Thank you for the valuable comment. Supplementary Figure 1b presents spectral imaging data of a freely diffusing fluorescent bead in aqueous solution, acquired without target-locked 3D tracking. In this case, the bead

frequently moved outside the excitation volume, leading to a substantial reduction in excitation efficiency and fluorescence signal. As a result, many individual frames exhibited significantly reduced instantaneous signal-to-noise ratios.

Beyond the impact on SNR, the absence of spectral centroid output in some frames is also attributable to the assumptions and design of our spectral fitting algorithm. Specifically, the algorithm is optimized for data acquired under target-locked 3D tracking, where the image of the target molecule remains near the center of the imaging window. To estimate SNR, the algorithm calculates the average intensity within a 5×5 pixel region at the center of the spectral image and compares it to the average intensity in four corner regions, which are treated as background. If the calculated SNR falls below a predefined threshold, the frame is excluded from spectral centroid estimation. This filtering strategy is particularly effective for rejecting frames in which the tracked molecule has drifted significantly from the image center, thereby preventing large errors due to poor localization or background artifacts.

When target-locked 3D tracking is enabled in 3D-SpecDIM, the molecule remains centered, and such filtering rarely impacts data analysis. However, in the absence of target-locked tracking (as in Supplementary Figure 1b), lateral (X-Y) drift may cause the particle to move away from the center. In these cases, the predefined central region may capture a weaker signal than surrounding areas, leading to an underestimation of the true SNR and the exclusion of frames that visually appear acceptable. For example, as shown below, frame 720 and its corresponding spectral data reveal a visible spectral signal; however, the signal intensity within the center region is lower than in adjacent areas, triggering the exclusion criteria.

We emphasize that this behavior is expected given the design of our algorithm and does not impact the intended application of 3D-SpecDIM, which operates with target-locked 3D tracking to maintain the target near the center of the imaging window. Under these conditions, spatial deviations are minimized, and the spectral estimation algorithm performs reliably.

Additionally, we note that the previous version of the kymogram was plotted using frame-wise colormap normalization to enhance visual clarity. However, this may have inadvertently given the misleading impression of consistently high signal levels. In the revised Supplementary Figure, we present raw (non-normalized) data and apply a fixed color scale across all frames, providing a more accurate representation of the actual signal variations.

Revision: In supplementary file, Supplementary Figure 1 has been modified:

In Page 14 Line 25, the following sentences have been added:

“To improve the robustness of centroid estimation under low-photon conditions, we implemented additional data quality control criteria. Specifically, frames were excluded if the average signal intensity within the central 5×5 region of the position image did not exceed a predefined threshold relative to the background, as estimated from the peripheral corner regions. Additionally, frames exhibiting fitting errors greater than 0.3 in either the positional or spectral channels were also discarded. These strategies ensure that only frames with sufficient signal quality and reliable localization are included in the final spectral centroid analysis.”.

7. *What is the accuracy of the calculated spectral centroids for each of the single dyes? Specifically: In Fig. 2k, the average spectral centroids of the individual dyes are reported as 593 nm (ATTO 565), 682 nm (ATTO 665), and 688 nm (Setau 647). A quick inspection of the published spectra for these dyes illustrates that none of the measured centroids reported in this paper are accurate: ATTO 565’s centroid should be around 612 nm (see emission spectrum at <https://www.atto-tec.com/ATTO-565.html>); ATTO 665’s centroid should be around 702 nm (spectrum: <https://www.atto-tec.com/ATTO-665.html>), and Setau647’s centroid should be around 693 nm (<https://app.fluorofinder.com/dyes/1531-setau-647-nhs-ex-max-649-nm-em-max-695-nm>). The centroids reported in the paper disagree with the published*

spectra by up to ~20 nm; this absolutely must be addressed. An additional note on this: If, as is often the case, the emission filters are set to eliminate the bluer side of the emission spectrum, then one would expect the experimentally determined centroids to skew red, not blue. Again, this highlights the major issue of accuracy vs. precision in this work.

Response: Thank you for raising this important point. We have carefully re-examined the reference spectra for ATTO 565, ATTO 665, and SeTau 647, as well as the centroid values reported in this study.

For ATTO 565, the maximum emission reported on the manufacturer's official website (ATTO-TEC; <https://www.atto-tec.com/ATTO-565.html>) is approximately **590 nm**, which aligns well with our measured centroid of 593 ± 1.4 nm, as shown in the figure below:

For ATTO 665, the official spectrum shows a emission peak around **680 nm** (<https://www.atto-tec.com/ATTO-655.html>), while our measured centroid is 682 ± 2.4 nm, also within a reasonable margin, as shown in the figures below:

ATTO 665

N
680.5 am

AT10 685 in nn never Farbstof I von, Typ AT10 647N. Zu den charakterististren Egenschaften des Farbsiofs zahlen stake Absorption. It, den

Table 1: Properties of available ATTO-dye NHS-esters

Dye-NHS	Solvent	MW	Am	Sq	Lm	11..4'	CFno	CS=
ATTO 390	DMSO	440	441	325.4	0	390 476	24000	0.46 0.09
ATTO 425	DMSO	498	499	383.4	0	439 485	45000	0.19 0.17
ATTO 430L5	DMSO	686	664	547.7	-1	436 545	32000	0 32 022
ATTO 465	DMSO	493	393	278.4	1	453 506	75000	1.09 0.48
ATTO 488	DMSO	981	687	570.6	-1	500 520	90000	0.22 0.09
ATTO 495	DMSO	549	449	334.4	+1	498 526	80000	0.45 0.37
ATTO 490LS	DMSO	793	771	654.8	-1	498 658	40000	0.39 021
ATTO Reel 10	DMSO	627	527	412.5	+1	507 531	100000	0.21 0.14
ATTO 514	DMSO	1111	851	734.6	1	511 532	115000	0.21 0.07
ATTO 520	DMSO	564	461	349.5	+1	517 538	100000	0.16 0.20
ATTO 532	DMSO	1081	743	626.7	-1	532 552	115000	0.20 0.09
ATTO Rho6G	DMSO	711	611	496.6	+1	533 557	115000	0.19 0.16
ATTO 5400	DMSO	756	656	541.6	+1	543 562	105000	0.27 0.26
ATTO 542	DMSO	1125	1011	893.0	-3	542 562	120000	0.18 0.08
ATTO 550	DMSO	791	691	576.8	+1	554 576	120000	0.23 0.10
MAN				4126		P006 Mg	Mar	Pall
Al III IIFTHOSIS	UPLISU	812	839	628.1	1	572 595	120000	0.26 0.10
ATTO Rho11	DMSO	763	664	548.7	+1	577 600	120000	0.26 0.09
ATTO Rho12	DMSO	699	600	484.6	+1	582 607	110000	0.11 0.37
ATTO Rho101	DMSO	787	687	572.7	+1	587 609	120000	0.18 0.17
ATTO 5750	DMSO	808	708	591.7	+1	582 609	120000	0.29 0.12
ATTO 5800	DMSO	892	792	677.9	+1	587 609	110000	0.32 0.11
ATTO 590	DMSO	788	688	572.7	0	593 622	120000	0.39 0.43
ATTO R.41013	DMSO	843	743	628.8	« 1	603 626	120000	0.28 0.43
ATTO 594	DMSO	1389	903	786.9	-1	603 626	120000	0.22 0.50
ATTO 610	ACN	588	488	373.5	+1	616 633	150000	0.03 0.06
ATTO 6120	DMSO	888	788	673.8	+1	615 642	115000	0.35 0.60
ATTO 620	DMSO	709	609	494.7	1	620 642	120000	0.04 0.06
ATTO R.9914	DMSO	981	881	766.6	+1	626 648	140000	0.26 0.47
ATTO 633	DMSO	749	649	534.7	+1	630 651	130000	0.04 0.05
ATTO 643	DMSO	955	933	817.1	-1	643 665	150000	0.05 0.04
ATTO 647	ACN	811	690	574.8	0	647 667	120000	0.08 0.04
ATTO 647N	DMSO	843	743	628.9	+1	646 664	150000	0.06 0.05
ATTO 655	DMSO	887	625	509.6	0	663 680	125000	0.24 0.08
ATTO Oxa12	DMSO	835	736	621.9	+1	662 681	125000	0.32 0.12
ATTO 665	DMSO	820	720	605.7	+1	662 680	160000	0.07 0.06
ATTO 680	DMSO	828	623	507.6	0	681 698	125000	0.30 0.17
ATTO 700	DMSO	831	663	547.7	0	700 716	120000	0.26 0.41
ATTO 725	ACN	813	513	398.5	+1	728 751	120000	0.08 0.06
ATTO 740	ACM	665	565	450.6	+1	743 763	120000	0.07 0.07
Art 1182	ACN	553	153	338.4	+1	668 110000	0.08 0.24	

IAW molecular weicht S the eye mainline countess, n oireof molecular metallic, dye cams IHPLC MS acetendretwater o 1 soh%

Regarding SeTau 647, Fluorofinder reports a peak emission near **693** nm, our measured centroid of 688 ± 2.1 nm is slightly lower.

Overall, the emission centroids reported in this study are generally consistent with manufacturer-provided values and do not exhibit the ~20 nm deviation. We also emphasize that our measurements were conducted under single-molecule conditions. Minor deviations in emission centroids may arise from differences in the local environment and solvent conditions. Factors known to influence fluorescence emission spectra include polarity, viscosity, pH, probe-probe interactions, and the rigidity of the local environment (Joseph R. Lakowicz, *Principles of Fluorescence Spectroscopy*, 3rd Edition, Chapter 6: Solvent and Environmental Effects, 2006).

It is also worth noting that the spectral accuracy of 3D-SpecDIM depends on the quality of spectral calibration, with the precision of calibration filters playing a critical role. Using higher-precision filters could further improve the accuracy of spectral measurements.

8. I am not clear on what SI Fig. 8 is showing. Is the blue line simply an inverse relationship between time and # Setau molecules to reach a 5nm tracking precision (seems most likely, but unclear)? Is it some sort of a fit to all of the data? Is this figure meant to illustrate that most experimental conditions were more favorable for tracking than the theoretical minimum? (If so, do they each produce better than 5 nm precision, as predicted?)

Response: Thank you for your comment. We apologize for the lack of clarity in the previous description.

In **Supplementary Figure 8**, the blue solid line represents a theoretical benchmark calculated by determining the minimum photon counts required to achieve 5 nm spectral precision at selected exposure times (50 ms, 100 ms, 200 ms, and 500 ms), and fitting these points with an inverse function.

We used an inverse function model based on the following rationale: to maintain a constant spectral precision (e.g., 5 nm), a minimum number of

photons N_{photon} must be collected during each frame. Assuming a constant photon emission rate R for each fluorophore, and an exposure time T , a single molecule contributes $R \cdot T$ photons. Therefore, to reach the required photon count, the number of molecules $n_{molecule}$ follows:

$$n_{molecule} \times R \times T = N_{photon}$$

$$N_{photon} \propto T$$

$$\Rightarrow n_{molecule} = R \cdot T$$

The scattered data points in the figure correspond to various experimental configurations presented in this study, including different dyes and exposure times. All data were normalized to represent the equivalent number of SeTau 647 molecules. The fact that most experimental points lie above and to the right of the theoretical curve indicates that the majority of our conditions exceeded the minimum photon requirement, thereby enabling or surpassing 5 nm spectral precision in practice.

Revision: In supplementary file, the caption of Supplementary Figure 8 has been modified:

“The blue solid line represents a theoretical benchmark, obtained by calculating the photon counts required to reach 5 nm spectral precision at selected exposure times (50 ms, 100 ms, 200 ms, and 500 ms), and fitting these points with an inverse function. Data points represent experimental configurations used in this paper, including different dyes and exposure times. All fluorescence intensities were normalized and converted to the equivalent number of SeTau 647 molecules.”.

9. SI Fig. 10 should show the kymograms of the spectra / localization for each example molecule.

Response: Thank you for your kind suggestion. We have now added the corresponding kymograms diagram for each example trajectory in the revised **Supplementary Figure 10**.

Revision: In Supplementary Information file, Supplementary Figure 10 has been modified:

10. SI Fig. 11 is a nice demonstration of single-fluorophore blinking, but shows surprisingly low brightness per fluorophore. According to Fig. 2m, Atto 565 should have $\sim 17\text{kHz}$ brightness (assuming the same excitation parameters). Yet in SI Fig 11 its brightness is more like 10 kHz. In the response to Reviewer #2 the authors state that the power was reduced; this should be documented properly in the SI / methods to explain this discrepancy.

Response: Thank you for pointing this out. We have added this clarification to the Supplementary Information to avoid any confusion.

Revision: In Supplementary Information file, the caption of Supplementary Figure 11 has been modified:

“(b) Fluorescence intensity as a function of time for the trajectory shown in (a), demonstrating alternation between single-dye and two-dye intensity levels. A lower laser power was used in this experiment to reduce photobleaching (1.56 μ W after objective), resulting in decreased single-molecule fluorescence intensity compared to Fig. 2m.”.

11. SI Fig. 11 should also show the raw kymogram of the spectrum, and a trace of the centroid, to illustrate that it does not change when one molecule blinks.

Response: Thank you for your suggestion. This photoblinking experiment was designed to demonstrate the single-molecule sensitivity of 3D-SpecDIM. In this experiment, we focused on monitoring the fluorescence intensity fluctuations of the target molecules; therefore, spectral images were not recorded during data acquisition. However, if you believe spectral images are necessary for this experiment, we are happy to perform additional experiments to acquire the corresponding spectral data under photoblinking conditions.

12. SI Fig. 25: It is still not clear whether the authors account for bin stretching (more wavelengths are contained in the redder bins, translating to apparent higher counts in red bins in the raw data. Based on the consistent errors with higher measured red shoulder counts evident in Fig. SI 25d, it appears to me that the authors are not accounting for this stretch when transforming their spectra to wavelength space. This issue exacerbates inaccuracies in spectral measurement.

Response: Thank you for raising this important point. We agree that when converting pixel positions to wavelengths, it is essential to account for the non-linear dispersion introduced by the prism, which causes a denser distribution of wavelengths in the redder region of the spectrum.

To address this, our calibration procedure employs a second-order polynomial mapping from pixel space to wavelength space. This wavelength calibration inherently compensates for the non-uniform bin widths caused by the prism's dispersion. To address this, our calibration procedure includes a second-order polynomial mapping from pixel space to wavelength space. This wavelength calibration inherently compensates for the uneven bin widths introduced by the prism. This approach is consistent with that used in previous studies employing prism-based spectroscopy, such as Zhang et al., *Nature Methods*, 2015 (doi: 10.1038/nmeth.3528), as shown in the figure below. Therefore, although we did not explicitly mention bin stretching correction, it is already incorporated into our calibration method.

[FIGURE REDACTED]

13. On a related note, I continue to be concerned that not accounting for spectrally-dependent camera detection efficiencies harms the accuracy of the measured spectra.

Response: Thank you for your comment. We have carefully evaluated the influence of wavelength-dependent EMCCD detection efficiency on spectral accuracy. Although a common correction method involves dividing the measured intensity by the EMCCD's quantum efficiency at each wavelength, we found this approach to be unsuitable under low-photon-count conditions due to the presence of noise, such as in single-molecule spectral imaging.

To clarify this, we describe the forward imaging process of the EMCCD system as follows. The fluorescence spectrum of a molecule is represented by the theoretical photon's distribution $Phot(x)$. The detected signal is given by:

$$I_{detected}(x) = Phot(x) \cdot QE(x)$$

This signal is further subject to Poisson noise and background (Artur S. Nat Methods 18, 1082-1090 (2021)):

$$I_{measured}(x) \sim \text{Poisson}(I_{detected}(x) + B)$$

where B denotes background noise, assumed to be wavelength independent. Directly correcting $I_{measured}(x)$ by dividing it by $QE(x)$ can amplify noise and introduce bias, particularly in regions with low detection efficiency. This issue becomes especially problematic under low-photon-count conditions, where the signal is weak and comparable to the background level.

To validate this, we conducted both simulations and experimental comparisons. In our simulations, we extracted the QE curve of our EMCCD camera from the manufacturer's website and incorporated it into our simulation pipeline.:

Input: ground truth spectrum with 600nm centroid: $\lambda(x)$

average background level: $B=10$

quantum efficiency curve: $QE(x)$

Step 1: Modulate $\lambda(x)$ by $QE(x)$: $I_{det}(x) = \lambda(x) * QE(x)$

Step 2: Add Poisson-distributed noise: $I_m(x) = Poisson(I_{det}(x) + B)$

Step 3A: Estimate centroid from $I_m(x)$ via Gaussian fitting

Step 3B: Apply pixel-wise QE correction: $I_{cor}(x) = I_m(x)/QE(x)$

Estimate centroid from $I_{cor}(x)$ via Gaussian fitting

Step 4: Repeat Steps 1-3 over multiple trials

Step 5: Repeat Steps 1-4 over multiple trials to compute centroid error

We compared two centroid estimation strategies: (1) fitting a Gaussian function directly to the raw measured spectrum $I_m(x)$, and (2) first applying a pixel-wise QE correction $I_m(x)/QE(x)$, followed by Gaussian fitting. As shown in Supplementary Figure 33, under conditions of low signal-to-background ratio, such as when both the signal and background photon counts are 10 (similar with those in single-molecule spectral tracking experiment), applying QE correction significantly increases the spectral centroid error: the standard deviation rises from 5.16 nm (uncorrected) to 9.86 nm (corrected), and the estimation bias shifts from -0.23 nm to +10.72 nm. In contrast, at a higher SNR (e.g., 1000 signal photons and 10 background photons), QE correction slightly improves centroid accuracy, reducing the bias from -0.21 nm (uncorrected) to +0.042 nm (corrected), while the precision remains nearly identical (SD = 0.1160 nm vs. 0.1163 nm). However, such marginal improvement under ideal conditions does not justify the loss of robustness under realistic, low-photon-count scenarios commonly encountered in single-molecule experiments.

In addition to our simulation results, we experimentally confirmed that direct QE correction under low-photon-count conditions can introduce significant spectral bias. In our single-molecule tracking experiments, we consistently

observed a red shift of approximately 4-5 nm following QE correction (e.g., from 593 nm to 596 nm, 683 nm to 687 nm, and 688 nm to 693 nm). This is consistent with our simulation result that dividing noisy measurements by $QE(x)$ amplifies background fluctuations and distorts centroid estimation (Supplementary Fig. 33). In contrast, for high-signal experiments such as that in Fig. 2e, where the emission rate exceeded 5 MHz, the difference in spectral centroid with and without QE correction was only 0.0744 nm (519.8357 nm vs. 519.9101 nm). These results confirm our conclusion that while QE correction may yield marginal improvements under ideal high-SNR conditions, it introduces notable errors under realistic, photon-limited conditions and thus is not suitable for single-molecule applications in our system.

Revision: In Supplementary Information file, the following figure has been added

Supplementary Figure 33. Evaluation of QE correction in spectral centroid estimation. (a) Quantum efficiency (QE) curve of the EMCCD sensor. The shaded region (500-740 nm) indicates the working spectral range of the 3D-SpecDIM system. (b) Monte Carlo simulation of spectral centroid estimation under different photon counts. Blue boxes: no QE correction; red boxes: with QE correction. QE correction leads to increased error under low-photon conditions (e.g., 10 photons and 10 background), while the difference becomes negligible at high photon counts (≥ 500). (c) Comparison of spectral profiles with and without QE correction for a high-intensity fluorescent bead (corresponding to Fig. 2e, emission rate > 5 MHz). The centroid difference is minimal, demonstrating that QE correction has limited impact under high signal levels. (d) Comparison of spectral profiles for a single Atto565 molecule with and without QE correction. The QE-corrected spectrum shows amplified noise, consistent with the simulation results.

In supplementary file, Supplementary Note 6 was added:

Supplementary Notes 6. Influence of wavelength-dependent EMCCD detection efficiency on spectral accuracy

To evaluate the effect of wavelength-dependent quantum efficiency (QE) on spectral accuracy, we modeled the forward imaging process of the EMCCD system. The detected signal is given by⁵:

$$I_{measured}(x) \sim \text{Poisson}(Phot(x) \cdot QE(x) + B), \quad (11)$$

where $Phot(x)$ is the ground-truth photon distribution along the spectral axis, $QE(x)$ is the wavelength-dependent detector sensitivity, and B is a constant background.

We compared two centroid estimation methods (**Supplementary Figure 33**): (1) direct Gaussian fitting of $I_{measured}(x)$; (2) pixel-wise QE correction prior to fitting. Simulations showed that at low photon counts (e.g., 10 photons), QE

correction increased centroid bias from -0.23 nm to $+10.72$ nm and raised the standard deviation from 5.16 nm to 9.86 nm. At high photon counts (e.g., 1000), correction slightly improved bias (~ 0.042 nm), but precision remained similar (~ 0.116 nm). Experimental validation also confirmed these trends: in single-molecule conditions, QE correction introduced red shifts of 4–5 nm, while in high-intensity bead data, the centroid difference was negligible (~ 0.074 nm). These results confirm that while QE correction may yield marginal improvements under ideal high-SNR conditions, it introduces notable errors under realistic, photon-limited conditions and thus is not suitable for single-molecule applications in our system.

14. In SI Fig. 25d, the yellow fluorescent bead's measured spectrum does not match the bulk measurement, showing two distinct peaks rather than a shoulder. Why?

Response: Thank you for your comment. The discrepancy observed in the yellow fluorescent bead spectrum is primarily due to the cutoff characteristics of the quad-band dichroic mirror used in the detection path of 3D-SpecDIM (denoted as DC in Supplementary Figure 2). This dichroic mirror blocks light around 641 nm (<https://www.chroma.com/products/parts/zt405-488-561-640rpcv2>), as shown in the figure below. As a result, a portion of the yellow bead's emission is attenuated at this wavelength, producing an artificial dip in the spectrum and resulting in the appearance of two peak.

[FIGURE REDACTED]

15. SI Fig. 30: Why should the intensity fluctuations (often up to a factor of 2 or 3x) be attributed to changes in photon collection efficiency? Are they correlated to Z position? If so, this point needs to be thoroughly explained, especially in light of the “example” traces shown for tracking single dyes in SI Figs. 10 and 11, which do not fluctuate in this way despite changing Z position. *In particular, SI Fig. 30b is troubling, as this looks like blinking of a multi-dye cluster and does not appear to correlate to positional changes seen in the main figure localization trace.*

Response: Thank you for raising this important point. We agree that interpreting fluorescence intensity fluctuations is essential for understanding single-molecule trajectories. These fluctuations are indeed associated with changes in photon collection efficiency, which depend on the axial (Z) position of the particle. This effect is well explained by the theoretical framework of supercritical-angle fluorescence (SAF). According to this model, the fluorescence of fluorophores located near the glass-solution interface can be more efficiently collected due to enhanced SAF components. As the molecule moves axially away from the interface, the SAF contribution decreases, resulting in reduced photon collection efficiency (as shown in the figure below, see *Winterflood, C.M., et al. 2010. Physical Review Letters 105.*).

[FIGURE REDACTED]

This axial dependence of fluorescence collection is also strongly influenced by the refractive index contrast between the sample medium and the coverslip. A greater mismatch in refractive indices enhances the contribution of SAF, leading to more pronounced intensity fluctuations with axial displacement. This is supported by *Axelrod, D., 2013. Biophysical Journal 104, 1401-1409: “The higher the refractive index n_3 , the more evanescent components of the fluorophore near field can be captured, converted into propagating light, and detected”*.

In our experiments, during 3D tracking of fluorescent beads in aqueous solution, the substantial refractive index contrast between the glass coverslip ($n \approx 1.52$) and the aqueous medium ($n \approx 1.33$) facilitates pronounced SAF-based signal modulation. As shown in Supplementary Figure 30a, the fluorescence intensity exhibits a clear correlation with the Z-position, consistent with emission profiles dominated by supercritical-angle fluorescence.

In contrast, our single-molecule experiments were conducted in 90% glycerol solution ($n \approx 1.47$), where the refractive index closely matches that of glass. Under these conditions, the detected fluorescence is relatively insensitive to small axial displacements, and no strong Z-dependent modulation is expected.

Based on this theoretical framework, we re-evaluated the trajectory shown in the previous version of Fig. 2g-i. We found that the observed intensity fluctuations are best attributed to the blinking behavior of a two-dye cluster, consistent with your interpretation.

To clarify, we subsequently examined the single-molecule trajectories from the dataset used in **Fig. 2j-m**. As shown below, these data exhibit relatively stable fluorescence intensity over time with no significant blinking, confirming that the event shown in previous version of **Supplementary Figure 30b** is not representative of our broader dataset.

To better show the single-molecule tracking capability of our system, we have replaced the previous version of Fig. 2g-i with another single-molecule trajectory. The updated data exhibit relatively stable fluorescence intensity without signs of blinking and are more representative of our typical measurements. This replacement does not affect the overall conclusions of the study.

We sincerely appreciate your thoughtful observation, which significantly enhanced the clarity and precision of our analysis

Revision: **Fig. 2g-i** have been revised:

In the Supplementary Information file, **Supplementary Figure 30b** has been revised:

16. Additionally, SI Fig. 30b seems to show 6 frames per second rather than the stated 100ms (10 frames per second) exposure for single molecule experiments.

The 6 frames per second issue is also evident in SI movie #5.

Response: Thank you for pointing this out. Upon reviewing our experimental log and camera settings, we found that the exposure time used in this single-molecule experiment (previous version of **Fig. 2g-i**) was actually 150 ms, not 100 ms as previously stated. We apologize for this oversight. This longer exposure time, combined with the EMCCD readout time, resulted in an effective frame interval of approximately 163 ms. This timing structure is

illustrated in Supplementary Figure 7, which indicates both the exposure and readout durations.

We would like to emphasize that all other single-molecule experiments, including those shown in **Figure 2j-m, Supplementary Figure 10, and Supplementary Figure 11**, were all conducted with the exposure time of 100 ms, as originally reported. We have now corrected this detail in the revised manuscript and figure legends accordingly. Additionally, the replaced data in Figure 2g-i was collected using the 100 ms exposure time.

Revision: In the caption of Fig. 2, the following sentence was modified:

The inset (g) displays the fluorescence spectral image captured by the EMCCD and the corresponding spectral profile at the current time point. Exposure time: 100 ms. (m) of three different fluorophores: Atto 665 (N=33), SeTau 647 (N=35), and Atto 565 (N=26). The mean and standard deviation values are labeled on the histogram figure. Exposure time: 100 ms.

17. SI Fig. 31: These kymograms appear to be peak-normalized; they would be far better presented with the original scaling to allow the reader to more readily evaluate changes in signal-to-noise and brightness from the raw data.

Response: Thank you for the helpful suggestion. We have re-plotted the kymograms in Supplementary Figure 31 using the original (non-normalized) intensity to more accurately reflect variations in signal-to-noise ratio and fluorescence brightness.

Revision: Supplementary Figure 31 has been modified:

18. In particular, because of the peak-normalization, it is difficult to understand why certain frames of SI Fig. 31b fail while others that appear to have worse SNR produce a result.

Response: Thank you for pointing out this important issue. To clarify, we have reprocessed the data and re-plotted the kymograms without applying frame-by-frame normalization.

In the figure below, we overlaid the extracted spectral centroid values (white curve) onto the kymogram, with black arrows indicating frames where spectral reconstruction failed due to low signal-to-noise ratio. In the previous version, each frame was independently peak-normalized, which may have caused confusion by visually exaggerating the apparent SNR and making it difficult to distinguish frames with poor data quality.

We would also like to clarify why certain frames with seemingly low SNR were still successfully assigned spectral centroid values. This outcome stems from

the nature of our Gaussian fitting algorithm, which evaluates the full pixel distribution across the spectral image. In some cases, despite a weak or noisy visual appearance, the data retain sufficient structural features—such as peak shape and spatial continuity—to enable a stable fit. Conversely, some frames (e.g., Frame #3 and Frame #22) may appear visually bright but exhibit local asymmetry or noise at the spectral center, leading the algorithm to reject them based on internal quality criteria such as fit failure, high residuals, or failure to meet center-weighted SNR thresholds. For instance, although Frames #4, #25, and #32 appear visually dim, they display consistent Gaussian-like profiles that were successfully fitted by the algorithm.

Thank you again for your valuable suggestion. Enhancing the robustness of spectral centroid extraction under low-SNR conditions remains an important focus for our future work.

19. Can the authors comment on why the pH ratio appears to change / fluctuate between high and low so rapidly during the transition? In the kymogram shown in SI Fig. 31c, these “switch” events are even more evident than they originally appear in Fig. 3f. It almost looks like blinking of one or both channels, but this doesn’t make sense because this is not close to single-molecule level...

Response: Thank you for this insightful comment. We have carefully re-examined the presentation of the kymograms. In the previous version, the

apparent rapid switching behavior was partly due to frame-by-frame normalization applied during kymogram generation, which artificially amplified visual fluctuations. In the revised version, we have replotted the kymograms using the original intensity scaling to more accurately reflect signal changes.

Additionally, it is important to note that in some biological processes such as mitophagy, lysosomes do not maintain continuous, stable contact with damaged mitochondria. Instead, they often exhibit dynamic transient interactions before achieving stable fusion or engulfment. This dynamic nature has been reported in previous studies. For example, Wong et al., *Nature*, 2018 (page 382, doi:10.1038/nature25486), observed that “*LAMP1* vesicles approached mitochondria to form stable contacts (yellow arrows), but eventually left mitochondria (white arrow) without engulfing them”, suggesting that lysosomes may transiently interact and then disengage from mitochondria without completing full fusion. Such behaviors likely contribute to the transient pH ratio fluctuations observed during the mitophagy process.

Revision: Supplementary Figure 31c has been modified:

20. SI movies should include the kymograms (not peak-normalized).

Response: Thank you for your suggestion. We have now added the corresponding kymograms to the Supplementary Movies.

Revision: The kymograms have been added to the Supplementary Movies.

21. Code was not available at the stated link.

Response: We apologize for the access issue. Now the following GitHub repository is publicly accessible: <https://github.com/houlab/3D-SpecDIM>.

REVIEWER COMMENTS

Reviewer #3 (Remarks to the Author):

I appreciate the authors' publication of the code, as well as many of their manuscript modifications.

However: I continue to be very concerned that this approach yields inaccurate centroid values. In my previous comments, I provided a few centroid values that were calculated directly from reference spectra from the websites listed. The authors' response instead compared their measured centroid values to the PEAK values from these reference spectra. As I'm sure the authors are aware, the centroid of a distribution is not necessarily the same as the location of its peak, and therefore it is not a valid comparison to compare centroid values to peak values. In the case of these reference emission spectra, the centroids are as much as 20 nm away from the peak values. Therefore, the new stated error of "+/- 3.48 nm" and the new SI Table 7 are completely incorrect and need to be revised to compare to reference spectra centroids, NOT peak values.

So, my previous major concern stands: the reported centroids are not close to matching reference spectra centroids, so this method appears to be rather inaccurate despite its admirable precision.

Response: We greatly appreciate your valuable comment and sincerely apologize for any confusion caused by our use of the term "spectral centroid."

*We would like to clarify that, throughout our manuscript, the term "spectral centroid" was used operationally to refer to **the peak position of a fitted Gaussian function**, rather than the intensity-weighted average wavelength (i.e., the true mathematical centroid). This definition was stated in the **Methods** section of the previous version of the manuscript:*

377 **Spectral centroid calculations**

378 The "normal fitting method" refers to fitting spectral curves with a Gaussian distribution.
379 Specifically, this method involves using all recorded intensity-wavelength pairs and applying a
380 curve fitting algorithm to extract the parameters of the Gaussian profile. The spectral centroid
381 corresponds to the position of the peak of the fitted Gaussian curve, representing the most likely
382 wavelength of maximum emission.

Likewise, in our earlier **Response Letter** to Reviewer #2, we explicitly clarified that the spectral centroid refers to the wavelength corresponding to the maximum intensity of the fluorescence emission spectrum, as shown in the screen shot below:

Minor comments

1. The manuscript does not define 'spec. cent.' despite its use. A description of what constitutes the centroid of the spectrum is necessary for understanding its relevance and how it is calculated within the context of the study.

Response: Thank you for pointing out this oversight. "Spec. cent." refers to the spectral centroid, which is the wavelength corresponding to the maximum value of the fluorescence emission spectrum. We have now included a detailed explanation of this term in the manuscript.

Revision: In the caption of Fig. 2a, the following explanation has been added:

The use of Gaussian fitting to extract the **spectral peak emission wavelength** has been previously reported in the literature (Nat Commun 7, 13544, 2016, <https://doi.org/10.1038/ncomms13544>):

[FIGURE REDACTED]

This definition was applied consistently in our spectral calibration, spectral analysis, and comparisons with reference spectra. Accordingly, the values presented in Supplementary Table 7 reflect comparisons between our measured peak positions and the peak wavelengths reported in the reference spectra. These comparisons are thus consistent with our intended definition.

However, we now recognize that the term “centroid” was not appropriately used and may have led to confusion, as it can be interpreted differently within the context of spectral analysis. To address this issue, we have revised all mentions of “centroid position” to “spectral peak emission wavelength” throughout the manuscript to ensure clarity and consistency.

We believe that these revisions have significantly improved the clarity and rigor of the manuscript, and we are sincerely grateful to you for highlighting this important distinction.

Revision: The Fig2, Fig 4, Fig S1, Fig S6, FigS10, Fig S12, Fig S21, Fig S22, Fig S24 and Fig S31 have been modified by replacing the “spec cent.” with “wavelength”. Also, all the expressions of “spectral centroid” is replaced by “spectral peak emission wavelength” in both main text and supplementary file.